# CACTI: Leveraging Copy Masking and Contextual Information to Improve Tabular Data Imputation

Aditya Gorla [1 2]   Ryan Wang [3]   Zhengtong Liu [3]   Ulzee An [1 3]   Sriram Sankararaman [1 3 4]

## Abstract

We present CACTI, a masked autoencoding approach for imputing tabular data that leverages the structure in missingness patterns and contextual information. Our approach employs a novel median truncated copy masking training strategy that encourages the model to learn from empirical patterns of missingness while incorporating semantic relationships between features – captured by column names and text descriptions – to better represent feature dependence. These dual sources of inductive bias enable CACTI to outperform state-of-the-art methods – an average $R^2$ gain of 7.8% over the next best method (13.4%, 6.1%, and 5.3% under missing not at random, at random and completely at random, respectively) – across a diverse range of datasets and missingness conditions. Our results highlight the value of leveraging dataset-specific contextual information and missingness patterns to enhance imputation performance. Code is publicly available at github.com/sriramlab/CACTI

## 1. Introduction

Missingness is a pervasive problem in real-world tabular datasets with the potential to adversely affect downstream inferential tasks (Rubin, 1987; Schafer & Graham, 2002). While many techniques to estimate or *impute* missing entries have been proposed (see Section 2.1), missing data imputation remains a challenging problem.

A primary reason underlying this challenge is that missing-

ness can arise due to a variety of mechanisms. Existing methods either explicitly or implicitly make simplifying assumptions about these mechanisms motivated by inferential tractability (see Section 2.1 and Jarrett et al. (2022)). These assumptions rarely hold in real-world settings and practitioners often lack prior knowledge on the underlying missingness mechanism. Consider a medical survey where questions are hierarchically structured, with more specific inquiries contingent upon affirmative responses to general ones so that a patient is only asked about specific symptoms if they report a broader health issue. In this example, entries pertaining to more specific health issues will be missing depending on the observed values or missingness status of entries relevant to broader health status. We hypothesize that the missingness patterns in the data could potentially be leveraged to improve imputation accuracy.

Additionally, existing methods underutilize the rich contextual information in the data. While they allow for the inclusion of fully observed covariates, they lack a straightforward mechanism to effectively incorporate unstructured knowledge about the relatedness between or the context of the features being imputed. In the medical surveys example, the answer to the broader question can constrain the answers to more specific questions. We hypothesize that imputation models can use this prior information to inform their imputation.

**Contributions**   In this work, we present **C**ontext **A**ware **C**opy masked **T**abular **I**mputation (**CACTI**), a transformer-based architecture that leverages inductive biases from observed missingness patterns and textual information about features to address existing gaps in tabular data imputation. CACTI makes several novel contributions to tabular imputation. First, we introduce median truncated copy masking (MT-CM), a novel training strategy that enables the effective application of copy masking (An et al., 2023) to transformer-based Masked Autoencoders (MAE) (He et al., 2021). Unlike existing approaches which use complete data or random masks (Du et al., 2024), MT-CM uses empirical missingness patterns to guide the learning process. Our results demonstrate that a naive application of copy masking to transformer-based MAE architectures leads to suboptimal performance while MT-CM addresses this gap. Second,

[1]Department of Computational Medicine, David Geffen School of Medicine, UCLA, Los Angeles, CA, USA [2]Bioinformatics Interdepartmental Program, UCLA, Los Angeles, CA, USA [3]Department of Computer Science, UCLA, Los Angeles, CA, USA [4]Department of Human Genetics, UCLA, Los Angeles, CA, USA. Correspondence to: Aditya Gorla <adityagorla@ucla.edu>, Sriram Sankararaman <sriram@cs.ucla.edu>.

*Proceedings of the 42nd International Conference on Machine Learning*, Vancouver, Canada. PMLR 267, 2025. Copyright 2025 by the author(s).

we provide theoretical motivation for MAE training without fully observed data which motivates the need for copy masking. Third, we leverage contextual information from feature names and descriptions as a source of inductive bias. This context-aware approach enhances learning efficiency by minimizing reliance on learning features' relationships solely from limited observed data and provides a direct way to incorporate unstructured information or prior knowledge. Fourth, our comprehensive evaluation establishes CACTI as a state-of-the-art tabular imputation approach across various missingness settings. Finally, both MT-CM and context awareness frameworks are simple and modular, allowing them to be used in conjunction with any deep learning framework beyond the tabular imputation domain.

## 2. Background

We begin by introducing the tabular imputation task adopting notation similar to previous works (Jarrett et al., 2022; Ipsen et al., 2021; Yoon et al., 2018) to ensure consistency. The *complete* data for a single sample with $K$ features, $\mathbf{X}_n := (x_{n1}, \cdots, x_{nK}) \in \mathcal{X} = \mathcal{X}_1 \times \cdots \times \mathcal{X}_K$ with $k \in [K]$ features and $n \in [N]$ observations, is drawn i.i.d from an arbitrary data generating process $\mathbf{X}_n \sim \mathcal{D}_K$.

We do not have access to the complete data but only the *incomplete*, observed data: $\tilde{\mathbf{X}}_n := (\tilde{x}_{n1}, \ldots, \tilde{x}_{nK})$, Equation (1). The incomplete data can be viewed as a *corrupted* version of the complete data mediated by the missingness mask $\mathbf{M}_n = (m_{n1}, \ldots, m_{nK}) \in \{0,1\}^K$, where $x_{nk}$ is observed if $m_{nk} = 1$ and $x_{nk}$ is missing (denoted as $*$) if $m_{nk} = 0$:

$$\tilde{x}_{nk} = \begin{cases} x_{nk}, & \text{if } m_{nk} = 1 \\ * , & \text{if } m_{nk} = 0 \end{cases} \in \tilde{\mathcal{X}}_k := \mathcal{X}_k \cup \{*\} \quad (1)$$

Across $N$ observations, this process results in an observed data matrix $\tilde{\mathbf{X}} = (\tilde{\mathbf{X}}_1; \ldots; \tilde{\mathbf{X}}_N)$ and the associated mask $\mathbf{M} = (\mathbf{M}_1; \ldots; \mathbf{M}_N)$.

The imputation task can be formalized as a learning a function $f : \tilde{\mathcal{X}} \to \mathcal{X}$ resulting in an *uncorrupted* version of the incomplete data $\bar{\mathbf{X}}_n := (\bar{x}_{n1}, \ldots, \bar{x}_{nK})$ resulting in the final imputed dataset ($\hat{\mathbf{X}}_n = (\hat{x}_{n1}, \ldots, \hat{x}_{nK})$, Equation (2)):

$$\hat{x}_{nk} = \begin{cases} x_{nk}, & \text{if } m_{nk} = 1 \\ \bar{x}_{nk}, & \text{if } m_{nk} = 0 \end{cases} \quad (2)$$

Additionally, we might have access to additional information that can be leveraged to aid imputation. Specifically, we assume we have access to external information (shared across all $N$ samples) such as the semantic *context* and relatedness between features which can be represented as $\mathbf{C} := (\mathbf{C}_1, \ldots, \mathbf{C}_K) \in \mathbb{R}^{C \times K}$, a $C$-dimensional embedding representation of context information for each feature.

**Missingness Mechanisms** Let us define a selector function $s_{\mathbf{M}_n} : \mathcal{X} \to \prod_{k \in \{k:m_{nk}=1\}} \mathcal{X}_k$ that selects all the observed features in the complete data. $\mathbf{X}_n^o := s_{\mathbf{M}_n}(\mathbf{X}_n)$ defines the *observed* part and $\mathbf{X}_n^m := s_{1-\mathbf{M}_n}(\mathbf{X}_n)$ defines the *missing* part. The framework laid out by Rubin (1976) (also (Little & Rubin, 1987)) prescribes the following underlying missingness mechanisms, from the most to the least restrictive assumption: **MCAR** ($p(\mathbf{M}_n|\mathbf{X}_n) = p(\mathbf{M}_n)$, i.e. $\mathbf{M}_n \perp \mathbf{X}_n$; missingness is independent of the data), **MAR** ($p(\mathbf{M}_n|\mathbf{X}_n) = p(\mathbf{M}_n|\mathbf{X_n}^o)$; missingness only depends on the fully observed data), and **MNAR** when the mechanism is neither MCAR nor MAR.

### 2.1. Related work

There are two main classes of tabular imputation methods: *iterative* and *generative*. Iterative methods iteratively impute the missing values in each feature by estimating the conditional distribution given all other features' observed data (van Buuren & Groothuis-Oudshoorn, 2011; Stekhoven & Bühlmann, 2011; Jarrett et al., 2022). While estimating the conditional distribution is a simpler problem, these approaches are limited by challenges in selecting optimal conditional distributions and sometimes requiring complete observations for model fitting. In contrast, generative approaches attempt to estimate a joint distribution of all the features which is a considerably harder statistical task than estimating univariate conditional probabilities (Yoon et al., 2018; Dai et al., 2021; Yoon & Sull, 2020; Mattei & Frellsen, 2019; Ipsen et al., 2021; Nazabal et al., 2020; Zhang et al., 2024a; Zheng & Charoenphakdee, 2023; Muzellec et al., 2020). Many of these approaches require either complete data or restrictive assumptions on the missingness mechanisms (Nazabal et al., 2020; Richardson et al., 2020; Mattei & Frellsen, 2019). Other classical imputation approaches include: K-nearest neighbors, matrix completion and unconditional mean substitution (Hastie et al., 2014; Hawthorne & Elliott, 2005).

**Transformers** Several recent works have proposed transformer (or self-attention) based architectures to model tabular data (Huang et al., 2020; Arik & Pfister, 2020; Majmundar et al., 2022; Yoon et al., 2020; Hollmann et al., 2025; Gardner et al., 2024). However, these approaches primarily focus on self-supervised learning tasks by employing a masked reconstruction task or target direct downstream prediction and do not explicitly address the imputation problem. Recent works (Yin et al., 2020; Yang et al., 2024; Lin et al., 2024; An et al., 2025) have also leveraged unstructured (natural language) contextual awareness to improve representation learning, pre-training efficiency and the performance of generative tabular models; however, these approaches have not yet been effectively leveraged in tabular imputation.

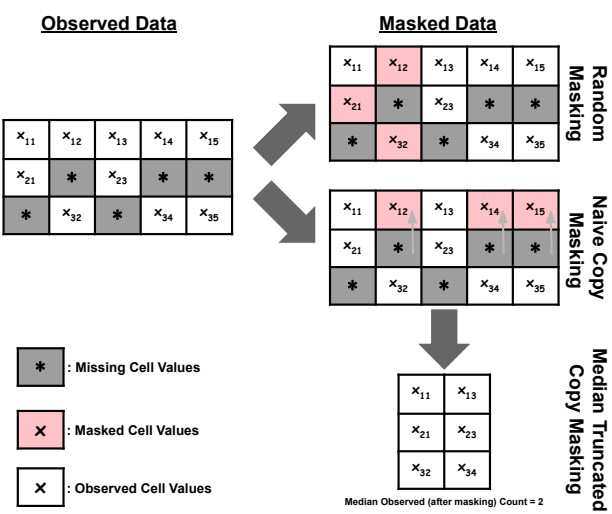

**Observed Data**     **Masked Data**

*Figure 1.* **Median Truncated Copy Masking overview**. In contrast to random masking, where some subset of features are masked uniformly at random, copy masking recycles missing value patterns actually present in the dataset. This approach simulates realistic missingness patterns that provide a source of useful inductive bias during training. Median Truncated Copy Masking extends this strategy for MAE training by truncating the number of features available to the encoder, ensuring it has access to at most the median number of fully observed features in each batch.

ReMasker (Du et al., 2024), a transformer-based approach for tabular imputation that builds on the MAE approach (He et al., 2021), learns to reconstruct randomly masked values based on the unmasked observed values (Figure 1). The model is highly expressive but is trained under a (completely) random masking strategy during training.

**Copy masking**    Recent work by An et al. (2023) proposed using the missingness patterns in the observed data to create masks to train an imputation model under a reconstruction loss function. Given an observed missingness mask $\mathbf{M}$, copy maksing involves shuffling the matrix row-wise to create a mask $\mathbf{M}^{perm} \in \{0, 1\}^{N \times K}$ where with probability $p_{cm}$ (masking ratio) we either apply the $\mathbf{M}^{perm}$ mask for a sample or leave it unchanged. We term the resulting mask matrix as the naive *copy mask* $\mathbf{M}^{cm}$ (Figure 1; See Algorithm 1 for details[1]). While Autocomplete (An et al., 2023) implements naive copy masking in conjunction with a shallow MLP to show strong downstream performance, this approach is limited in its expressivity to learn complex relational patterns between the features.

---

[1]Notice that this algorithm results in sampling $\mathbf{M}$ with replacement when number of epochs is $> 1$.

## 3. CACTI

CACTI employs an encoder-decoder Transformer architecture for tabular data imputation. This architecture needs to be trained on a reconstruction task. However, since the observed data is incomplete, a masking strategy that introduces additional missingness on which the quality of reconstruction can be assessed must be devised.

### 3.1. Median truncated copy masking

Previous works for tabular data imputation (Du et al., 2024) adopt the same approach used in MAEs: applying a *random mask* on the observed portions of the incomplete data during training. Our first contribution is in replacing random masking. We extend naive copy masking (An et al., 2023) to develop median truncated copy masking (MT-CM) which leverages the missingness structure in the observed data to create masks that better reflect true missingness patterns. We hypothesize that this approach provides a useful inductive bias for the model that can be particularly effective in cases where missingness is structured (Jackson et al., 2023), *e.g.*, consider the missingness pattern $p(m_{ni} = 0 | m_{nj} = 0) = 1$ where feature $i$ is missing any time $j$ is missing. While it is challenging to define a unified or well-defined generative model for the mask, the empirical patterns of missingness provide useful information to design such a mask.

Under MT-CM (and naive copy masking), we can segregate features in each sample into three *sets*: the (observed but) masked values $\mathbb{M}_n^{cm} = \{k : (m_{nk} = 1) \cap (m_{nk}^{cm} = 0)\}$, the unmasked values $\mathbb{O}_n^{cm} = \{k : (m_{nk} = 1) \cap (m_{nk}^{cm} = 1)\}$ and the true missing values $\mathbb{V}_n = \{k : m_{nk} = 0\}$. Consequently, we can define a training strategy by minimizing a reconstruction loss over the value sets $\mathbb{M}_n^{cm}$ and $\mathbb{O}_n^{cm}$.

A naive application of copy masking (Figure 1) to transformer-based MAE architectures, however, leads to inefficient learning due to the large variance in missingness proportions across samples (Mitra et al., 2023; Jackson et al., 2023; An et al., 2023) while uniform feature sizes (or sequence lengths) within a batch are critical for efficient learning with a transformer-based encoder (Krell et al., 2022). A possible approach to enforce uniformity when using naive copy masking is to replace all missing or masked features with a null token. This strategy, even at low copy masking rates, results in a significant proportion of null tokens in each batch, which provides no meaningful information for learning a robust latent representation. Furthermore, increasing the copy masking rate proportionally increases the fraction of null tokens that can further reduce learning efficiency and overall model performance. Empirical results confirm this trend, with higher rates of naive copy masking leading to reduced model performance (see Appendix A).

To tackle this issue, we propose the Median Truncation Copy

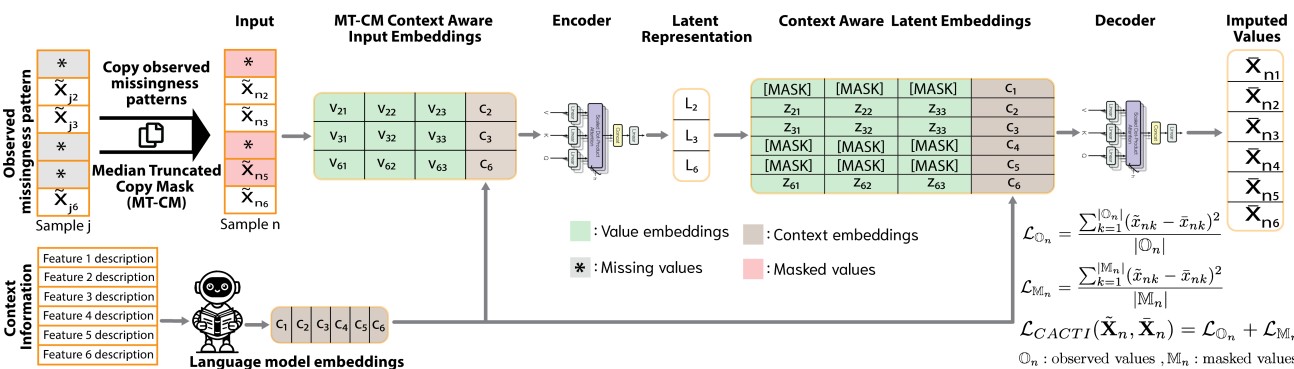

*Figure 2.* **CACTI model overview.** CACTI samples observed missingness patterns to generate masks via Median Truncated Copy Masking (MT-CM) to guide the learning. Features' context are also embedded with a language model. The MT-CM strategy masks out some portion of the observed features from sample $n$ using observed missingness patterns from other samples ($j$) in the same dataset. This is followed by concatenating context information to the remaining (unmasked) features. A transformer encoder processes this data. Then the model adds context information and [MASK] tokens for the missing/masked features before being processed by the decoding transformer which reconstructs the values. CACTI optimizes reconstruction loss ($\mathcal{L}_{CACTI}$) over observed and masked features to produce the final imputation estimates.

Masking (MT-CM) training strategy (Figure 1). Let $N_B$ be the number of samples in the $B$-th batch, and $o_n = |\mathbb{O}_n^{cm}|$ be the number of observed features in the $n$-th sample after the application of naive copy masking in this batch. The median number of observed values within the batch is defined as $o_B^{median} = \text{median}(o_1, \ldots, o_{N_B})$.

The MT-CM strategy truncates the sequence length of observed values for each sample to ensure it contains no more than $o_B^{median}$ observed values. Formally, for each sample $n$ in the batch, the truncated sequence length $o_n^{trunc}$ is computed as $o_n^{trunc} = \min(o_n, o_B^{median})^2$. This ensures that the proportion of null tokens in any batch is upper bounded by 50% regardless of the copy masking rate. Overall, MT-CM results in the final set of observed and masked features: $\mathbb{O}_n = \{k : (m_{nk} = 1) \cap (m_{nk}^{cm} = 1) \cap (k \leq o_n^{trunc})\}$ and $\mathbb{M}_n = \mathbb{M}_n^{cm} \cup (\mathbb{O}_n^{cm} \setminus \mathbb{O}_n)$, respectively. During training, the feature order of each sample in every batch is permuted to ensure that the first $o_n^{trunc}$ features are retained as observed features and is different every iteration. See Algorithm 2 for extended MT-CM details. We also empirically show that, unlike naive copy masking, our MT-CM strategy results in overall performance increasing as the copy masking rate increases (see Appendix A).

### 3.1.1. THEORETICAL MOTIVATION FOR COPY MASKING

In this section, we provide a brief theoretical motivation of the need for copy masking.

Assume the *complete* data for a single sample is drawn from an arbitrary data generating distribution $\mathbf{X} \overset{i.i.d}{\sim} P_X(\mathbf{x})$. This

complete data vector undergoes a corruption process mediated by a missingness mask which results in the partially observed data: $\tilde{\mathbf{X}} = \mathbf{X} \odot \mathbf{M}$ where the missingness mask process $\mathbf{M}|\mathbf{X} \sim P_{M|X}$.

Under a masked autoecoding model, we aim to learn an encoder-decoder ($f_\psi$ and $d_\theta$ respectively) that minimizes the risk:

$$R(\psi, \theta) =$$
$$\mathbb{E}_{\mathbf{X}, \mathbf{M}}\left[||\mathbf{X} \odot (1 - \mathbf{M}) - d_\theta(f_\psi(\mathbf{X} \odot \mathbf{M})) \odot (1 - \mathbf{M})||_2^2\right] \quad (3)$$

Here $\odot$ denotes entrywise product.

The risk (or its finite-sample approximation) defined in Equation 3 cannot be computed since we only observe $\tilde{\mathbf{X}}$. Instead, given the missing data $\tilde{\mathbf{X}}$, we generate a mask $\mathbf{M}'|\tilde{\mathbf{X}}, \mathbf{M} \sim Q_{M'|\tilde{\mathbf{X}}, M}$ and aim to minimize the alternate risk:

$$R_Q(\psi, \theta) = \mathbb{E}_{\mathbf{X}, \mathbf{M}}\Big[\mathbb{E}_{\mathbf{M}'|\tilde{\mathbf{X}}, \mathbf{M}}[||\mathbf{X} \odot \mathbf{M} \odot (1 - \mathbf{M}')$$
$$- d_\theta(f_\psi(\mathbf{X} \odot \mathbf{M} \odot \mathbf{M}')) \odot \mathbf{M} \odot (1 - \mathbf{M}')||_2^2]\Big] \quad (4)$$

Consider a sample that is completely observed so that $\mathbf{M} = \mathbf{1}$ so that $\tilde{\mathbf{X}} = \mathbf{X}$. On this sample, $R_Q$ becomes:

$$R_Q(\psi, \theta) = \mathbb{E}_{\mathbf{X}, \mathbf{M} = \mathbf{1}}\Big[\mathbb{E}_{\mathbf{M}'|\mathbf{X}, \mathbf{M} = \mathbf{1}}[||\mathbf{X} \odot (1 - \mathbf{M}')$$
$$- d_\theta(f_\psi(\mathbf{X} \odot \mathbf{M}')) \odot (1 - \mathbf{M}')||_2^2]\Big] \quad (5)$$

Equation 5 motivates choosing $Q$ to be the same distribution as $\mathbf{M}|\mathbf{X}$ so that $R_Q \approx R$. More broadly, this motivates choosing a masking distribution $Q$ that approximates the true distribution of missing entries $\mathbf{M}|\mathbf{X}$. For example, if the true missingness mechanism is MCAR where the

---

[2]In Algorithm 1, every sample retains $\geq 1$ feature; $\mathbf{M}_i^{perm}$ is not applied if it results in 0 remaining observed features.

probability of each feature being missing is independent and identically distributed, a random masking strategy where each entry masked independent of other features with a constant probability is expected to provide an appropriate inductive bias for the imputation model.

These observations drive the core rationale for copy masking. Copy masking tries to approximate the true masking distribution and missingness structure by sampling from the observed missingness mask. For example, in the MCAR setting where each feature has a constant probability of being missing, copy masking will reduce to random masking. On the other hand, when the missingness probability varies across features, copy masking will lead to features being masked with differential probabilities based on their empirical frequencies. The use of empirical masks can also capture correlations among features in the missingness mechanism. While copy masking is still a simplification and does not fully model the missingness mechanism, our empirical results suggest that it enables the imputation model to attend to realistic patterns of missingness.

### 3.2. Context Awareness

Our second key contribution is making the imputation backbone context aware by incorporating prior information about the semantic information associated with each feature by using language model embedding of feature description into the value embedding vector in both the encoder and decoder stage. We use the semantic similarity between column name and description information to make our imputation backbone context-aware, providing useful inductive bias to improve imputation performance.

Let $\tilde{\mathbf{X}}_n := (\tilde{\mathbf{X}}_{n1}, \ldots, \tilde{\mathbf{X}}_{nK})$ be the observed data for a single sample in $\tilde{\mathcal{X}}$. For the final model embedding dimension $E$, we would like to achieve a context-aware embedding of the data sample $\mathbf{E}_n = (\mathbf{E}_{n1}, \ldots, \mathbf{E}_{nK}) \in \mathbb{R}^{E \times K}$ (where $E = \dim(\mathbf{E}_{nk})$). To achieve this, we can create a partitioned embedding for each feature, which has a value component $\mathbf{U}_{nk}$ and context component $\mathbf{C}_{nk}$ such that $\mathbf{E}_{nk} = (\mathbf{U}_{nk}; \mathbf{C}_k)^3$, where $\mathbf{U}_{nk} \in \mathbb{R}^U, \mathbf{C}_k \in \mathbb{R}^C$ and $E = U + C$. As a design choice, we set $U = 0.75E$ and $C = 0.25E$, prioritizing value information as the primary object of relevance which warrants its overrepresentation relative to context information. We define a linear projection[4] $l : \tilde{\mathcal{X}} \to \mathcal{U}$ that maps each scalar feature value to a $U$-dimensional embedding vector representation, resulting in $\mathbf{U}_n = (\mathbf{U}_{n1}, \ldots, \mathbf{U}_{nK}) \in \mathbb{R}^{U \times K}$.

We propose using of language models to obtain representa-

tions (embeddings) of each column's semantic information. For each of the $K$ columns in the data $\tilde{\mathbf{X}}$, we process the column name and description (when available) through a language model (using default tokenizer) to obtain embeddings $\mathbf{C}_k^{ci}$. Given a set of tokenized descriptions $\mathcal{T}_k$, we obtain the last layer hidden state for each token and aggregate the information to obtain the column's semantic context $\mathbf{C}_k^{ci} = \frac{1}{|\mathcal{T}_k|} \sum_{i=1}^{|\mathcal{T}_k|} \text{Embd}(t_{ki})$. Since language models (Devlin et al., 2019; Lee et al., 2025) typically have hidden state dimensions in the range $[768, 4096]$, we perform a linear projection $r_e : \mathcal{C}^{ci} \to \mathcal{C}$ that maps each column information embedding to an $C$-dimensional context embedding, resulting in $\mathbf{C} = (\mathbf{C}_1, \ldots, \mathbf{C}_K) \in \mathbb{R}^{C \times K}$. Transformers also require fixed sin-cosine embeddings $\mathbf{P} = (\mathbf{P}_1, \ldots, \mathbf{P}_K) \in \mathbb{R}^{E \times K}$ to preserve positional information (Dufter et al., 2021). Thus final context-aware embeddings are achieved by concatenation of the value and context $\mathbf{E}_n = [\mathbf{U}_n || \mathbf{C}] + \mathbf{P}$, with positional information added.

Different base models can be used for generating context embeddings. In this study, we use the GTE-en-MLM-large, a new state-of-the-art text embedding model (Zhang et al., 2024b) as the default based on our empirical results comparing the effectiveness of these models. We note that the generation of column context embeddings has a one-time, fixed cost. These embeddings can be pre-computed and reused across multiple runs for the same dataset.

### 3.3. Transformer Backbone

Figure 2 provides a pictorial description of the CACTI autoencoder architecture backbone with a detailed description deferred to Appendix B. Briefly, the CACTI backbone consists of an encoder and decoder, both utilizing transformer architectures with (residual) self-attention blocks. The encoder processes context-aware embeddings ($\mathbf{E}_n$) of the observed data, dropping missing or masked features after applying the MT-CM strategy. The decoder combines context information embeddings and a latent representation of the MT-CM input features to estimate the uncorrupted version ($\bar{\mathbf{X}}_n$) of the incomplete data. The model is trained to minimize the reconstruction loss between the imputed and observed data, using a unified MSE loss over the masked ($\mathbb{M}_n$) and fully observed values ($\mathbb{O}_n$). See Algorithm 3 and Algorithm 4 for a sketch of the CACTI implementation.

## 4. Evaluation Results

We empirically evaluate CACTI's performance against state-of-the-art methods using 10 benchmarking datasets across all three missingness scenarios. Next, we conduct a thorough ablation analysis to quantify the contributions of our proposed MT-CM and context awareness strategies. Finally,

---

[3]$\mathbf{C}_k$ has no index subscript $n$ because we assume the feature context information is shared and consistent across all samples.

[4]We set all the true missing values to any special protected value for this step.

*Table 1.* **Overall benchmark results.** Average performance comparison of CACTI and CMAE (CACTI without context) against existing imputation methods on the train/test splits (separated by |) over 10 datasets. Metrics (arrows indicate direction of better performance) are evaluated under MAR, MCAR, and MNAR at 30% missingness. – indicates method which cannot perform out-of-samples (test split) imputation. Best metric in bold and second best underlined. Extended table with standard errors in Appendix.

| METHOD | $R^2$ ($\uparrow$) | | | RMSE ($\downarrow$) | | | WD ($\uparrow$) | | |
|---|---|---|---|---|---|---|---|---|---|
| | MCAR | MAR | MNAR | MCAR | MAR | MNAR | MCAR | MAR | MNAR |
| CACTI (OURS) | **0.46\|0.46** | **0.47\|0.47** | **0.46\|0.46** | **0.66\|0.64** | **0.67\|0.69** | **0.68\|0.67** | 4.35\|**4.46** | **1.87\|1.94** | 4.45\|**4.57** |
| CMAE (OURS) | 0.44\|0.45 | 0.46\|0.46 | 0.44\|0.44 | 0.67\|0.65 | 0.69\|0.70 | 0.70\|0.69 | 4.40\|4.50 | 1.94\|2.02 | 4.57\|4.69 |
| REMASKER | 0.44\|0.44 | 0.44\|0.44 | 0.40\|0.40 | 0.68\|0.67 | 0.69\|0.71 | 0.73\|0.71 | 4.62\|4.72 | 2.46\|2.52 | 4.79\|4.93 |
| DIFFPUTER | 0.40\|0.42 | 0.39\|0.43 | 0.36\|0.37 | 0.73\|0.70 | 0.77\|0.75 | 0.79\|0.77 | 4.53\|4.56 | 2.55\|2.38 | 4.79\|4.90 |
| HYPERIMPUTE | 0.41\|– | 0.44\|– | 0.39\|– | 0.72\|– | 0.73\|– | 0.76\|– | **4.26**\|– | 2.46\|– | **4.30**\|– |
| MISSFOREST | 0.35\|0.34 | 0.38\|0.36 | 0.34\|0.32 | 0.77\|0.75 | 0.79\|0.82 | 0.79\|0.78 | 6.78\|6.80 | 3.80\|3.83 | 7.01\|7.06 |
| notMIWAE | 0.35\|0.35 | 0.35\|0.35 | 0.29\|0.30 | 0.75\|0.74 | 0.80\|0.82 | 0.82\|0.80 | 5.56\|5.60 | 2.38\|2.39 | 6.26\|6.20 |
| SINKHORN | 0.28\|– | 0.29\|– | 0.26\|– | 0.84\|– | 0.89\|– | 0.88\|– | 7.02\|– | 3.96\|– | 7.51\|– |
| ICE | 0.28\|0.27 | 0.34\|0.33 | 0.26\|0.25 | 0.86\|0.87 | 0.78\|0.83 | 0.93\|0.93 | 4.82\|5.18 | 2.74\|2.81 | 5.32\|5.67 |
| AUTOCOMPLETE | 0.24\|0.24 | 0.29\|0.29 | 0.21\|0.21 | 0.88\|0.86 | 0.88\|0.89 | 0.94\|0.92 | 10.14\|10.18 | 5.04\|5.07 | 10.44\|10.42 |
| MICE | 0.19\|0.19 | 0.23\|0.23 | 0.18\|0.18 | 1.06\|1.04 | 1.04\|1.05 | 1.08\|1.07 | 8.25\|8.33 | 4.16\|4.23 | 8.34\|8.49 |
| GAIN | 0.19\|0.21 | 0.18\|0.22 | 0.17\|0.18 | 0.91\|0.86 | 0.95\|0.93 | 1.01\|0.96 | 7.73\|7.34 | 4.44\|4.10 | 9.53\|9.14 |
| SOFTIMPUTE | 0.09\|0.10 | 0.10\|0.11 | 0.09\|0.09 | 1.02\|0.96 | 1.06\|1.02 | 1.05\|0.99 | 8.35\|7.86 | 4.84\|4.46 | 8.73\|8.23 |
| MIWAE | 0.00\|0.00 | 0.00\|0.00 | 0.00\|0.00 | 1.00\|0.98 | 1.05\|1.07 | 1.03\|1.00 | 7.83\|7.90 | 4.53\|4.57 | 8.36\|8.37 |
| MEAN | 0.00\|0.00 | 0.00\|0.00 | 0.00\|0.00 | 0.95\|0.93 | 1.00\|1.02 | 0.98\|0.95 | 11.96\|12.00 | 6.35\|6.38 | 12.25\|12.26 |

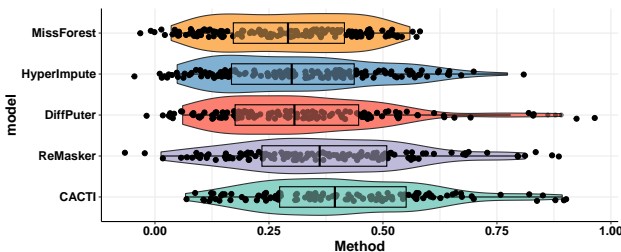

*Figure 3.* **Top-5 methods benchmark.** Violin plots display the distribution of $R^2$ metrics for each of the top five methods across all missing percentages and conditions over all datasets.

we conduct a comprehensive sensitivity analysis to identify key aspects and hyperparameter configurations that significantly impact the performance and usability of our method.

**Baseline methods** We benchmark CACTI against 13 top methods from the field. Detailed descriptions of the methods are defered to Appendix C.1. Briefly, we compare against ReMasker (Du et al., 2024) as the primary masked transformer-based autoencoing method. Diff-Puter (Zhang et al., 2024a) represents the recent state-of-the-art in diffusion-based imputation. AutoComplete (An et al., 2023) is a naive copy masking autoencoder model developed for biomedical data. Hyperimpute (Jarrett et al., 2022) is the current best iterative hybrid machine learning approach. We also compared to leading iterative methods: Missfor-

est (Stekhoven & Bühlmann, 2011), ICE (Royston & White, 2011) and MICE (van Buuren & Groothuis-Oudshoorn, 2011). and generative approaches: Sinkhorn (Muzellec et al., 2020), GAIN (Yoon et al., 2018), MIWAE (Mattei & Frellsen, 2019) and notMIWAE (Ipsen et al., 2021) (an extention of MIWAE for MNAR). Lastly, we also include widely-used approaches such as Softimpute (Hastie et al., 2014) and unconditional Mean (Hawthorne & Elliott, 2005). For all baselines, we use default (or recommended if available) settings for all models and CACTI default settings are outlined in Appendix C.2.

**Datasets** To allow for comparison with previous works (Section 2.1), we use ten real-world datasets (Kelly et al.), with details included in Appendix C.3. For each dataset, we create an 80-20 train-test split to test both in-sample and out-of-sample imputation. The data is fully observed, so we can simulate missingness under each of three missingness conditions: MCAR, MAR, and MNAR. For MCAR, each value is masked according to a Bernoulli random variable with fixed mean. In MAR, a random subset of features are fixed as fully observed while entries in the remaining features are masked based on a logistic model. For MNAR, we take the input features of the MAR mechanism and further mask them according to a Bernoulli random variable with fixed mean. In accordance with prior work, the primary benchmarking is performed under 30% simulated missingness proportion while extended results are included for 10%, 50% and 70% simulated missingness proportions. Simulations were performed using the HyperImpute package (Jarrett et al., 2022).

**Evaluations metrics** We evaluate imputation performance along three metrics: Pearson's $R^2$, root mean square error (RMSE) and Wasserstein distance (WD). We use $R^2$ as a measure of imputation concordance due to its invariance to mean or scale shifts and its direct applicability across continuous, binary, and ordinal features. For consistency with previous works (Zhang et al., 2024a; Jarrett et al., 2022), we report RMSE as an absolute measure of imputation accuracy and WD as a measure of alignment between the imputed and true values. Note that we perform evaluation on the original scale of each feature as opposed to the min-max transformed scale to reflect utility in real world applications. The main tables and figures present the mean of the metrics aggregated across the test split of the relevant datasets except for Table 1, which reports metrics on both the train and test splits. The Appendix contains figures reporting all the per-dataset metrics with 95% confidence intervals on both train and test splits.

Table 1 summarizes the average performance of each model across the 10 datasets under 30% simulated missingness. CACTI outperforms all existing baselines across all metrics and missingness conditions. We observe an average relative improvement over the next best method (with respect to $R^2$) of 13.4%, 6.1%, and 5.3% under MNAR, MAR, and MCAR, respectively. Notably in Appendix D we also observe that across the ten datasets under all three missingness mechanisms, CACTI dominates all methods in at least one of the three metrics and in a majority of the datasets outperform all other methods on all metrics. In our experiments, we also include median truncated Copy Masked Auto Encoder (CMAE; CACTI without context) as an additional baseline to demonstrate that CMAE alone consistently dominates ReMasker across R2 and RMSE metrics.

These results underscore the versatility of our approach in achieving effective imputation across diverse missingness scenarios without strong assumptions about the source of the missingness. The robust performance, particularly in the most challenging MNAR settings, highlights the advantage of leveraging inductive biases from observed data through the use of the MT-CM training strategy. Additionally, the improved accuracy of CACTI over ReMasker under MCAR, where MT-CM and random masking should be approximately equivalent, highlights benefits of context awareness.

In Appendix D, we extend our benchmarking to 10%, 50% and 70% simulated missingness proportions for all three mechanisms. Figure 3 displays the results of the top five methods for each dataset under each of the four missingness percentages and three mechanisms. The results show that, on average, CACTI is the most effective imputation approach across all settings. Lastly, we verify in Table A14 that the resource requirements while training CACTI are reasonable ($<$ 5.8 seconds per epoch on the largest dataset

*Table 2.* **Ablation analysis.** Comparison of models with RM, MT-CM, and/or CTX. ✓ indicates model has the feature and × if not. Metrics presented represent the average model performance at 30% missingness.

| MODEL | RM | CTX | MT-CM | $R^2$ (↑) | | RMSE (↓) | |
|---|---|---|---|---|---|---|---|
| | | | | MAR | MNAR | MAR | MNAR |
| RMAE | ✓ | × | × | 0.21 | 0.20 | 1.00 | 1.03 |
| RMAE+ CTX | ✓ | ✓ | × | 0.26 | 0.26 | 0.96 | 0.86 |
| CMAE | × | × | ✓ | **0.46** | 0.43 | 0.68 | 0.70 |
| CACTI | × | ✓ | ✓ | **0.46** | **0.45** | **0.67** | **0.68** |

and requiring $<$ 300MB of GPU memory both of which are comparable to that of ReMasker).

## 4.1. Ablation Analysis

We aim to investigate the relative contributions of key aspects of CACTI via a series of ablation analyses. First, we assess the relative contribution of MT-CM compared to random masking (RM). Second, we evaluate the impact of context awareness (CTX) when used in conjunction with random masking alone. Third, we analyze the additional gains achieved by incorporating context awareness on top of our MT-CM training strategy. Finally, we explore the value of each of the sources of inductive bias: the observed missingness patterns or the features' context information. To do this, we construct three additional models: 1) Random Masking Auto Encoder (RMAE), 2) Random Masking Auto Encoder with ConTeXt awareness (RMAE+CTX) and 3) CMAE. The RMAE model uses the same transformer backbone as CACTI while using the same random masking strategy as ReMasker, RMAE+CTX extends the RMAE model with the same context aware (CTX) embeddings used in CACTI, and CMAE is the CACTI model without the CTX embeddings. We conduct the ablation analysis over four different datasets (see Appendix C.3) under MAR and MNAR with masking ratio fixed at 90%.

The ablation results in Table 2 first indicate that both MT-CM and context awareness are essential for achieving good performance. Next, we observe that, under MNAR, MT-CM provides a 115% gain in $R^2$ over random masking while context awareness provides a 30% gain when used with random masking. We note since this is an internal ablation, all hyperparameters were held constant. This resulted in the performance of RMAE being lower than ReMasker due to differences in their masking rates. In Table A15, we conducted additional direct comparisons between CMAE and ReMasker on all ten datasets that demonstrate that CMAE (by replacing random masking with MT-CM) alone provides a statically significant improvement in performance compared to ReMasker (t-test p$<$0.05).

*Table 3.* **Context Contributions**. One sided paired T-test between CACTI and CMAE imputation $R^2$ to evaluate the statistical significance of contexts' contribution and out performance (win rate).

| Miss % | Avg. $R^2$ Gain (P-value $\times 10^{-2}$) | | | Win Rate % | | |
|---|---|---|---|---|---|---|
| | MCAR | MAR | MNAR | MCAR | MAR | MNAR |
| 10 | 0.014 (1.96) | 0.023 (0.79) | 0.017 (0.23) | 80 | 89 | 80 |
| 30 | 0.014 (0.35) | 0.007 (9.24) | 0.017 (0.09) | 80 | 70 | 100 |
| 50 | 0.011 (0.65) | 0.010 (3.66) | 0.016 (0.13) | 90 | 70 | 100 |
| 70 | 0.010 (2.01) | 0.012 (0.72) | 0.015 (0.05) | 89 | 90 | 100 |

Table 2 also shows that using context awareness in conjunction with MT-CM leads to a nearly 5% improvement. To quantify whether context provides a statistically significant contribution, we extended our analysis to directly contrast CACTI and CMAE (CACTI without context) under all missingness percentages, datasets and missingness settings (Table 3 and Figure A9). CACTI outperforms CMAE (win rate) in a majority of the datasets across all settings, with respect to $R^2$. We then performed one-sided paired t-tests to demonstrate that context provides a statistically significant improvement (p<0.05) in all settings except for MAR at 30%. These results confirm that context *can* improve imputation accuracy though its contribution can vary depending on the dataset and the missingness setting. Overall, while either one of our strategies could provide meaningful improvements in imputation accuracy, the use of empirical missingness patterns through copy masking tends to be more useful than the contextual information.

Lastly, we explored design choices involving the loss function (Table A16). Training with the loss over observed values alone ($\mathcal{L}_{\mathbb{O}}$) yields poor imputation performance. In contrast, training on reconstruction of masked values ($\mathcal{L}_{\mathbb{M}}$), by forcing the model to learn relationships between the observed and masked features, leads to significantly better performance. As expected, the combined (observed and masked value) reconstruction loss ($\mathcal{L}_{\mathbb{O}} + \mathcal{L}_{\mathbb{M}}$) consistently achieves the best performance, due to the constraint of maintaining a latent space that both preserves the relationship between observed while inferring missing features.

## 4.2. Model Sensitivity Analysis

### 4.2.1. Model architecture

First, we investigate CACTI's sensitivity to three core architectural configuration choices: encoder depth ($N_e$), decoder depth ($N_d$) and overall embedding dimension size ($E$). The aggregated results of this analysis over four different datasets (see Appendix C.3) under MAR and MNAR, with the masking ratio fixed at 90%, are summarized in Table 4. These results indicate that the encoder and decoder depths have a relatively minor impact (especially in the MNAR setting) although our results tend to slightly favor a deeper encoder ($N_e = 10$) and a shallower decoder ($N_d = 4$). In contrast, we observe higher sensitivity of our model with respect to the choice of embedding dimension size with highest accuracy attained at $E = 64$. Notably, we see a significant drop-off in performance at very large embedding sizes (near 512) likely due to over-fitting.

### 4.2.2. MT-CM masking rate

We next investigate the impact of the choice of MT-CM masking ratio ($p_{cm}$). This parameter can be interpreted as controlling the strength of the inductive bias during the learning process. A higher $p_{cm}$ encourages the model to place greater emphasis on the observed missingness patterns in the data, allowing the model to capture and extract additional information. Figure A12 summarizes the results of this analysis over 4 different datasets (see Appendix C.3) under MAR and MNAR. Our experiments indicate that, on average, $p_{cm} \geq 0.90$ results in the most accurate results, with slight differences based on the missingness mechanism ($p_{cm} = 0.99$ for MAR and $p_{cm} = 0.95$ for MNAR). We remark that this is a notable departure from existing random masking approaches (Du et al., 2024) which report the optimal choice of masking rate can differ significantly ($> 10\%$) based on the dataset.

### 4.2.3. Context embedding model

Since our ablation analysis indicates that context awareness does provide a meaningful improvement to performance, we would like to understand the sensitivity with respect to the choice of language model used to derive the contextual embeddings. To this end, we assess six open-source base models: BERT-base, BERT-large (Devlin et al., 2019), DeBERTa-v3-base, DeBERTa-v3-large (He et al., 2023), GTE-en-MLM-large (Zhang et al., 2024b) and NV-Embed-v2 (Lee et al., 2025). These base models usually have a dimension of 768 for their last layer while the large models have a dimension of 1024 and NV-Embed-v2 has a dimension of 4096. Table 5 summarizes the results of this analysis over four different datasets (see Appendix C.3) under MAR and MNAR. These results indicate that there is marginal sensitivity to the choice of embedding model with GTE-en-MLM-large leading to the highest accuracy while DeBERTa-v3-large obtains the lowest accuracy. There does not appear to be a clear relation between overall performance and embedding size. This indicates that the semantic context learned by each model is more important that the size of the model. This is also supported by the fact that NV-Embed-v2 (7B parameters) consistently under performs BERT-base (110M parameters). Overall GTE-en-MLM-large or BERT-

*Table 4.* **Model architecture sensitivity.** Average performance effect of (a) encoder depth, (b) decoder depth, and (c) embedding size. Metrics represent the average across four datasets at 30% missingness proportion.

| (a) Encoder Depth | | | | | (b) Decoder Depth | | | | | (c) Embedding Size | | | | |
|---|---|---|---|---|---|---|---|---|---|---|---|---|---|---|
| DEPTH | $R^2$ (↑) | | RMSE (↓) | | DEPTH | $R^2$ (↑) | | RMSE (↓) | | SIZE | $R^2$ (↑) | | RMSE (↓) | |
| | MAR | MNAR | MAR | MNAR | | MAR | MNAR | MAR | MNAR | | MAR | MNAR | MAR | MNAR |
| 4 | 0.46 | **0.44** | 0.68 | **0.69** | 4 | **0.47** | **0.44** | **0.67** | **0.69** | 32 | **0.46** | 0.42 | **0.67** | 0.70 |
| 6 | 0.46 | **0.44** | 0.68 | **0.69** | 6 | 0.46 | **0.44** | **0.67** | **0.69** | 64 | **0.46** | **0.44** | **0.67** | **0.69** |
| 8 | 0.45 | **0.44** | 0.68 | **0.69** | 8 | **0.47** | **0.44** | **0.67** | **0.69** | 128 | 0.40 | 0.43 | 0.73 | 0.70 |
| 10 | **0.47** | **0.44** | **0.67** | **0.69** | 10 | 0.44 | **0.44** | 0.69 | **0.69** | 256 | 0.41 | 0.39 | 0.72 | 0.73 |
| 12 | 0.46 | **0.44** | 0.68 | **0.69** | 12 | 0.43 | 0.43 | 0.70 | 0.71 | 512 | 0.38 | 0.35 | 0.85 | 0.97 |

large seems to be good default choices for generic English language tabular data. Additionally, the ratio of context dimension to total embedding dimension (CTX proportion $\frac{C}{E}$) directly influences the contirbution of context awareness. Table A17 shows that CACTI is fairly insensitive to the choice of CTX proportion with 50% or 25% of the embeddings ($E$) containing context information as optimal. Finally, context embeddings from domain-specific models like BioClinicalBERT (Alsentzer et al., 2019) may help improve imputation for specialized fields like biomedicine, where features have unique contextual relations (e.g., disease classifications). Prior work by Lehman et al. (2023) shows these models outperform general-purpose models on domain-specific tasks. But we leave this line of inquiry for future work.

*Table 5.* **Embedding model sensitivity.** Average performance effect of embedding models (30% missingness proportion).

| EMBEDDING MODEL | $R^2$ (↑) | | RMSE (↓) | |
|---|---|---|---|---|
| | MAR | MNAR | MAR | MNAR |
| BERT-BASE | **0.47** | **0.45** | **0.67** | 0.69 |
| BERT-LARGE | 0.46 | **0.45** | **0.67** | **0.68** |
| DEBERTA-V3-BASE | **0.47** | **0.45** | **0.67** | 0.69 |
| DEBERTA-V3-LARGE | 0.45 | 0.43 | 0.68 | 0.70 |
| GTE-EN-MLM-LARGE | **0.47** | **0.45** | **0.67** | **0.68** |
| NVEMBED-V2 | 0.46 | 0.44 | 0.68 | 0.69 |

### 4.2.4. TRAINING CONVERGENCE

We finally evaluate the training convergence behavior of our model in the ***letter*** dataset. The results in Figure 4 indicate that the convergence behavior differ based on the missingness setting. Under the more difficult MNAR imputation setting, increased training epochs results in a consistent increase in imputation accuracy that does not fully saturate even at 1500 epochs. In contrast, under the simpler MAR setting, the model quickly converges to its optimal performance around 300 epochs, with increased training causing overfitting as indicated by a reduction in test set accuracy. Given these results and assuming that we do not know the missingness regime a priori, we recommend users start with

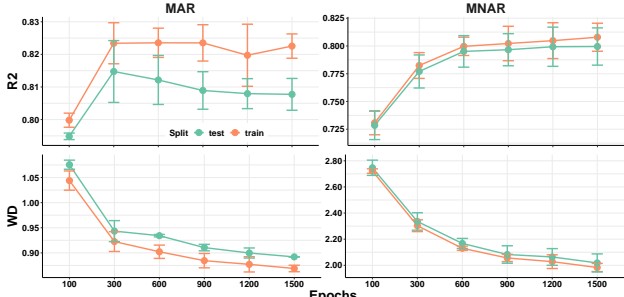

*Figure 4.* **CACTI Training profile.** Evaluated across training epochs under MAR and MNAR on the **letter** dataset with 30% missingness. Points are mean ± 95% CI.

300-600 epochs and monitor overfitting on validation data.

## 5. Conclusion

This work introduces a conceptual framework for leveraging information about context and the missingness patterns in the data to improve tabular data imputation. We posit that the observed missingness patterns and semantic information associated with the features serve as both crucial and valuable sources of inductive bias. These hypotheses led us to develop CACTI which integrates these dual sources of bias into a transformer-based imputation model. Our extensive benchmarking and ablation analysis demonstrate that information from each dataset's unique missingness patterns and column context significantly improves imputation accuracy, allowing CACTI to reach state-of-the-art performance. Our MT-CM masking strategy can be used with any masked learning model, while context awareness can be integrated into any deep learning-based imputation framework, demonstrating the broad applicability of our results. These results suggest that identifying additional sources and structures of useful bias is a worthwhile avenue for future tabular imputation research, particularly in fields with smaller datasets with high MNAR missingness such a biomedical data.

## Impact Statement

This paper presents CACTI whose goal is to advance the classical machine learning field of imputation for tabular data. There are many potential societal consequences of our work, none of which we feel must be specifically highlighted here. This is general approach that is compatible with any generic or field specific tabular dataset. CACTI allows users to more effectively learn an imputation function by leveraging the structure of missingness unique to each dataset and allows for straightforward integration of the unstructured textual information about the features being imputed.

## Acknowledgments

We thank Jonathan Flint for their valuable feedback and discussions throughout this project. S.S. was supported, in part, by NIH grant R35GM153406 and NSF grant CAREER-1943497. A.G. was supported, in part, by NIH grant R01MH130581 and R01MH122569.

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

# A. Copy Masking

---
**Algorithm 1** Naive copy masking.

---
1: **Input:** Observed mask $\mathbf{M} \in \{0,1\}^{N \times K}$, masking ratio $p_{cm} \in (0,1)$
2: $\pi : \{1, \ldots, N\} \to \{1, \ldots, N\}$  {permute indices}
3: $\mathbf{P}_{ij} = \begin{cases} 1, & \text{if } i = \pi(j) \\ 0, & \text{otherwise} \end{cases} \in \{0,1\}^{N \times N}$
4: $\mathbf{M}^{perm} = \mathbf{P}\mathbf{M}$  {Apply row-wise permutation}
5: $\mathbf{M}^{cm} = \mathbf{M}$  {init copy mask}
6: **for** $i \leftarrow 1, \ldots, N$ **do**
7: $\quad u \sim \mathcal{U}(0,1)$  {sample from uniform}
8: $\quad \mathrm{ct}_{\mathrm{feat}} = \mathbf{M}^{cm}_{i,:} \bullet \mathbf{M}^{perm}_{i,:}$  {count features left via dot product}
9: $\quad$ **if** $u < p_{cm}$ **and** $\mathrm{ct}_{\mathrm{feat}} \geq 1$ **then**
10: $\quad\quad \mathbf{M}^{cm}_{i,:} \leftarrow \mathbf{M}^{perm}_{i,:}$  {use copy mask}
11: $\quad$ **end if**
12: **end for**
13: **Output:** $\mathbf{M}^{cm}$

---

**Copy Masking ablation analysis**   We compare the performance characteristics of naive copy masking (CM) and MT-CM training strategy with respect to the masking rate ($p_{cm}$). To perform this analysis, we construct a copy masking auto encoder architecture (CMAE) which is our CACTI model without the context aware embeddings and use either the naive CM or MT-CM training.This analysis is performed on ***bike*** and ***obesity*** datasets and the average results across these two datasets under all three missingness scenarios are reported in  Table A6. The first three rows of Table A6 demonstrate a consistent decrease in performance across all three missingness settings as $p_{cm}$ is increased. The higher mask probability ($p_{cm}$) leads to more null tokens in each training batch which reduces training performance because the model must create meaningful latent representation (for the decoder) from positions that contain no information for the encoding layers to work with. As a result, a low $p_{cm}$ of about 10% producing the best performance. Strikingly, this trend is reversed by the MT-CM strategy (as seen in the last 5 rows of Table A6) where increasing the $p_{cm}$ results in increased performance with best performance attained at 90% masking. We also see that best performance (w.r.t $R^2$) under MT-CM is 6.7%, 5.8% and 2.1% higher than naive copy masking under MNAR, MAR and MCAR, respectively.

*Table A6.* Performance comparison of MT-CM vs naive CM and varying masking rate for CMAE. Metrics represent the average across four datasets.  Experiments were performed under three missingness scenarios at 30% missingness.  Best in bold and second best underlined.

| MASKING TYPE | MASKING RATE | $R^2$ ($\uparrow$) | | | RMSE ($\downarrow$) | | |
|---|---|---|---|---|---|---|---|
| | | MCAR | MAR | MNAR | MCAR | MAR | MNAR |
| NAIVE COPY MASKING | 10 | 0.378 | 0.396 | 0.358 | 0.644 | 0.620 | 0.643 |
| | 30 | 0.355 | 0.397 | 0.339 | 0.658 | 0.626 | 0.663 |
| | 50 | 0.348 | 0.384 | 0.324 | 0.667 | 0.638 | 0.673 |
| MEDIAN TRUNCATED COPY MASKING (MT-CM) | 10 | 0.366 | 0.346 | 0.341 | 0.651 | 0.654 | 0.656 |
| | 30 | 0.378 | 0.399 | 0.362 | 0.643 | 0.623 | 0.645 |
| | 50 | 0.375 | 0.408 | 0.371 | 0.647 | 0.617 | 0.637 |
| | 90 | **0.386** | 0.416 | **0.382** | **0.638** | **0.615** | **0.635** |
| | 95 | **0.386** | **0.420** | 0.376 | 0.641 | 0.622 | 0.640 |

---

**Algorithm 2** Median Truncated Copy Masking (MT-CM)

---

1: **Input:** Batch of embeddings $\mathbf{E} \in \mathbb{R}^{N_B \times K \times D}$, observed masks $\mathbf{M} \in \{0,1\}^{N_B \times K}$, naive copy masks $\mathbf{M}^{cm} \in \{0,1\}^{N_B \times K}$ (from Algorithm 1)

                                      {Step 1: Calculate observed feature counts after copy masking}

2: **for** $n \leftarrow 1, ..., N_B$ **do**

3:     $o_n \leftarrow \sum_{k=1}^{K} \mathbf{M}_{n,k}^{cm}$

4: **end for**

                                            {Step 2: Find median observed count}

5: $o_B^{median} \leftarrow \mathrm{median}(\{o_1, ..., o_{N_B}\})$

                            {Step 3: Apply median truncation with permutation}

6: **for** $n \leftarrow 1, ..., N_B$ **do**

7:     $o_n^{trunc} \leftarrow \min(o_n, o_B^{median})$                                   {Truncate to median at most}

8:     $\pi_n : \{1, ..., K\} \rightarrow \{1, ..., K\}$                           {Random permutation of features}

9:     $\mathbb{O}_n \leftarrow \emptyset, \mathbb{M}_n \leftarrow \emptyset$                            {Initialize observed and masked sets}

10:    $\mathbf{E}_n' \leftarrow []$                                          {Initialize truncated embeddings}

11:    $\mathrm{count} \leftarrow 0$

12:    **for** $k \leftarrow 1, ..., K$ **do**

13:       **if** $\mathbf{M}_{n,\pi_n(k)}^{cm} = 1$ **and** $\mathrm{count} < o_n^{trunc}$ **then**

14:          $\mathbb{O}_n \leftarrow \mathbb{O}_n \cup \{\pi_n(k)\}$                           {Add to observed set}

15:          $\mathbf{E}_n' \leftarrow [\mathbf{E}_n' || \mathbf{E}_{n,\pi_n(k)}]$                        {Append embedding}

16:          $\mathrm{count} \leftarrow \mathrm{count} + 1$

17:       **else if** $\mathbf{M}_{n,\pi_n(k)} = 1$ **then**

18:          $\mathbb{M}_n \leftarrow \mathbb{M}_n \cup \{\pi_n(k)\}$                           {Add to masked set}

19:       **end if**

20:    **end for**

21:    **if** $o_n^{trunc} < o_B^{median}$ **then**

22:       $pz = o_B^{median} - o_n^{trunc}$                                {Null token padding size}

23:       $\mathbf{NT}_n \leftarrow \mathbf{0}^{pz \times D}$

24:       $\mathbf{E}_n' \leftarrow [\mathbf{E}_n' || \mathbf{NT}_n]$                  {Concat null token padding when needed}

25:    **end if**

26: **end for**

27: **Output:** $[\mathbf{E}_1'; ...; \mathbf{E}_{N_B}'], \{\mathbb{O}_1, ..., \mathbb{O}_{N_B}\}, \{\mathbb{M}_1, ..., \mathbb{M}_{N_B}\}$

---

# B. CACTI method extended details

We now provide a detailed description on the transformer-based autonencoding backbone architecture of CACTI.

**Encoder** : The context-aware embeddings $\mathbf{E}_n$ are used for the subsequent steps in the transformer backbone. For the embedding $\mathbf{E}_n$ of the incomplete data $\tilde{\mathbf{X}}_n$, we define a selector function $se_{\mathbf{CP-MT}}$ that drops all missing or masked features based on the missingness mask $\mathbf{M}_n^{cm}$, median truncates each batch as previously described and any remaining missing/masked cells are replaced with fixed null padding. This process results in $\mathbf{E}'_n = se_{\mathbf{CP-MT}}(\mathbf{E}_n)$ where $\mathbf{E}'_n \in \mathbb{R}^{E \times K'}$ and $K'(= o_B^{median})$ is the number of features after median truncation. This matrix $\mathbf{E}'_n$ is processed by the encoder that consists of a series of $N_e$ self-attention blocks with residual connections, where the output is $\mathbf{L}_n$.

**Decoder** : We transform the context $\mathbf{C}^{ci}$ information embedding into decoder context embeddings using a linear projection $r_d : \mathcal{C}^{ci} \to \mathcal{C}'$ where $\mathcal{C}' \subseteq \mathbb{R}^{C \times K}$. The underlying rationale is that the encoder and decoder benefit from different kinds of context information. Next, define selector function $sd_{\mathbf{M}}$ that maps the latent representation $\mathbf{L}$ to match the shape and order of the original input features. The missing/masked features are filled with a fixed mask vector which is then passed through a linear projection to become decoder value information, resulting in $\mathbf{V}_n \in \mathbb{R}^{U \times K}$. The decoder context and value information is concated (denoted by $||$), with positional encoding, to get the context-aware decoder latent representational $\mathbf{Z}_n = [\mathbf{V}_n || \mathbf{C}'] + \mathbf{P}$. This latent representation is processed through $N_d$ layers of self-attention with residual connections. The final output is passed through a 2-layer MLP $g : \mathcal{Z} \to \bar{\mathcal{X}}$ to estimate the uncorrupted version of the incomplete data $\bar{\mathbf{X}}_n$.

**Optimization** : The model is trained to minimize the reconstruction loss $\mathcal{L}(\tilde{\mathbf{X}}_n, \bar{\mathbf{X}}_n)$ between the imputed and the observed data. We optimize our model against a loss function which is a sum of the loss over the observed value ($\mathbb{O}_n$) and masked (i.e., observed but hidden) values ($\mathbb{M}_n$). Note that we perform a min-max scaling of the input data before passing the cell values to the model. This allows us to use a unified MSE loss for all features and constrains the models search space. The model's internal output $\bar{\mathbf{X}}_n$ is therefor logits. The final loss formulation is:

$$
\begin{aligned}
\mathcal{L}_{\mathbb{O}_n} &= \frac{\sum_{k=1}^{|\mathbb{O}_n|}(\tilde{x}_{nk} - \bar{x}_{nk})^2}{|\mathbb{O}_n|} \\
\mathcal{L}_{\mathbb{M}_n} &= \frac{\sum_{k=1}^{|\mathbb{M}_n|}(\tilde{x}_{nk} - \bar{x}_{nk})^2}{|\mathbb{M}_n|} \\
\mathcal{L}(\tilde{\mathbf{X}}_n, \bar{\mathbf{X}}_n) &= \mathcal{L}_{\mathbb{O}_n} + \mathcal{L}_{\mathbb{M}_n}
\end{aligned}
\tag{6}
$$

We train our model through stochastic gradient descent using the AdamW optimizer with learning rate (lr) 0.001, default decay settings (0.90, 0.95) and Cosine Annealing with warmup lr scheduler. During inference, the model's internal output is transformed back to the original space for continuous features by inverting in the min-max scaling.

---

**Algorithm 3** CACTI Training Algorithm

---

1: **Input:** Training dataset $\mathcal{D} = \{(\tilde{\mathbf{X}}_i, \mathbf{M}_i)\}_{i=1}^{N}$, Context information $\mathbf{C}^{ci}$, Masking ratio $p_{cm}$, Training epochs $Epoch_{max}$, Batch size $B$

2: **Parameters:** Encoder weights $\psi$, Decoder weights $\theta$, (encoder, decoder) Value embedding weights $\phi_{e,d}$, (encoder, decoder) Context embedding weights $\omega_{e,d}$, Fixed sin-cos positional embeddings $\mathbf{P}$

3: **for** $epoch \leftarrow 1, \dots, Epoch_{max}$ **do**

4:     $\mathbf{M}^{cm} \leftarrow \texttt{NaiveCopyMask}([\mathbf{M}_1; ...; \mathbf{M}_N], p_{cm})$                 {Apply naive copy masking, Algorithm 1}

5:     **for** each batch $\mathcal{B} = \{(\tilde{\mathbf{X}}_n, \mathbf{M}_n, \mathbf{M}_n^{cm})\}_{n=1}^{B}$ sampled from $\mathcal{D}$ and $\mathbf{M}^{cm}$ **do**

6:         {Process batch of samples}

7:         **for** $n \leftarrow 1, ..., B$ **do**

8:             $\mathbf{U}_n \leftarrow l_{\phi_e}(\tilde{\mathbf{X}}_n)$                           {Project observed data to value embeddings}

9:             $\mathbf{C}_n \leftarrow r_{\omega_e}(\mathbf{C}^{ci})$                          {Project context info. to context embeddings}

10:            $\mathbf{E}_n \leftarrow [\mathbf{U}_n || \mathbf{C}_n] + \mathbf{P}$        {Concat. value and context embeddings + add positional embeddings}

11:         **end for**

12:         $\mathbf{E}', \mathbb{O}, \mathbb{M} \leftarrow \texttt{MT-CM}([\mathbf{E}_1; ...; \mathbf{E}_B], [\mathbf{M}_1; ...; \mathbf{M}_B], [\mathbf{M}_1^{cm}; ...; \mathbf{M}_B^{cm}])$         {Median truncated copy masking; Algorithm 2}

13:         $\mathcal{L}_{batch} \leftarrow 0$

14:         **for** $n \leftarrow 1, ..., B$ **do**

15:             $\mathbf{L}_n \leftarrow \texttt{Encode}_\psi(\mathbf{E}'_n)$                 {Apply encoder on remaining context aware embeddings}

16:             $\mathbf{V}_n \leftarrow \texttt{MaskReorder}_{\phi_d}(\mathbf{L}_n, \mathbb{O}_n, \mathbb{M}_n)$ {match original feat. order, set missing feats. to [MASK] and project}

17:             $\mathbf{C}' \leftarrow r_{\omega_d}(\mathbf{C}^{ci})$                    {Project context info. to decoder context embeddings}

18:             $\mathbf{Z}_n \leftarrow [\mathbf{V}_n || \mathbf{C}'] + \mathbf{P}$        {Concat. value and context embeddings and add positional embeddings}

19:             $\bar{\mathbf{X}}_n \leftarrow \texttt{Decode}_\theta(\mathbf{Z}_n)$                             {Decode a.k.a do imputation}

20:             $\mathcal{L}_n \leftarrow \mathcal{L}_{\mathbb{O}_n}(\tilde{\mathbf{X}}_n, \bar{\mathbf{X}}_n) + \mathcal{L}_{\mathbb{M}_n}(\tilde{\mathbf{X}}_n, \bar{\mathbf{X}}_n)$         {Calc. MSE loss over observed and masked values}

21:             $\mathcal{L}_{batch} \leftarrow \mathcal{L}_{batch} + \mathcal{L}_n$

22:         **end for**

23:         $\mathcal{L}_{batch} \leftarrow \mathcal{L}_{batch} / B$                                 {Average loss over batch}

24:         $\psi, \theta, \phi_{e,d}, \omega_{e,d} \leftarrow \text{UpdateWeights}(\nabla \mathcal{L}_{batch})$                       {Gradient update}

25:     **end for**

26: **end for**

27: **Output:** Trained parameters $\psi, \theta, \phi_{e,d}, \omega_{e,d}$

---

---

**Algorithm 4** CACTI Inference Algorithm

---

1: **Input:** Observed data $\mathcal{D} = \{(\tilde{\mathbf{X}}_i, \mathbf{M}_i)\}_{i=1}^{N}$, Context information $\mathbf{C}^{ci}$

2: **Parameters:** (From Algorithm 3) Trained encoder weights $\psi$, decoder weights $\theta$, value embedding weights $\phi_{e,d}$, context embedding weights $\omega_{e,d}$, positional embeddings $\mathbf{P}$

3: $\mathbf{C} \leftarrow r_{\omega_e}(\mathbf{C}^{ci})$

4: $\mathbf{C}' \leftarrow r_{\omega_d}(\mathbf{C}^{ci})$

5: **for** $n \leftarrow 1, \dots, N$ **do**

6:     $\mathbf{U}_n \leftarrow l_{\phi_e}(\tilde{\mathbf{X}}_n)$

7:     $\mathbf{E}_n \leftarrow [\mathbf{U}_n || \mathbf{C}] + \mathbf{P}$

8:     $\mathbf{E}'_n \leftarrow \texttt{TRUNC}(\mathbf{E}_n, \mathbf{M}_n)$                                     {Drop all true missing feats.}

9:     $\mathbf{L}_n \leftarrow \texttt{Encode}_\psi(\mathbf{E}'_n)$

10:     $\mathbf{V}_n \leftarrow \texttt{Mask}_{\phi_d}(\mathbf{L}_n)$                            {set missing feats. to [MASK] and project}

11:     $\mathbf{Z}_n \leftarrow [\mathbf{V}_n || \mathbf{C}'] + \mathbf{P}$

12:     $\bar{\mathbf{X}}_n \leftarrow \texttt{Decode}_\theta(\mathbf{Z}_n)$

13:     $\hat{\mathbf{X}}_n \leftarrow \tilde{\mathbf{X}}_n \odot \mathbf{M}_n + \bar{\mathbf{X}}_n \odot (1 - \mathbf{M}_n)$                     {set missing feats. to imputed values}

14: **end for**

15: **Output:** $\hat{\mathbf{X}}_n$

---

# C. Experimental Details

## C.1. Baseline methods overview

ReMasker (Du et al., 2024) applies the random masking (transformer) autoencoder framework to impute missing values. DiffPuter (Zhang et al., 2024a) is a method that leverages the Expectation-Maximization algorithm along with a diffusion model to iteratively learn the conditional probability of missing data. AutoComplete (An et al., 2023) is a naive copy masking autoencoder based model which learns to reconstruct missing values. Hyperimpute (Jarrett et al., 2022) is an iterative imputation framework that automatically selects and configures (classical machine learning) models for column-wise imputation. Missforest (Stekhoven & Bühlmann, 2011) iteratively trains random forest models on observed data and applies them to impute missing data. ICE (Royston & White, 2011) is a method which conditionally models and imputes missing data iteratively until convergence. MICE (van Buuren & Groothuis-Oudshoorn, 2011) is variation of ICE which utilizes Bayesian ridge regression. Softimpute (Hastie et al., 2014) uses iterative rank-restricted soft singular value decomposition to complete a matrix with missing values. Sinkhorn (Muzellec et al., 2020) is a generative method utilizing optimal transport distances as a loss criterion to impute missing values. GAIN (Yoon et al., 2018) is a generative-adversarial network with the generator trained to impute missing values conditioned on observed values, and the discriminator trained to identify which values were imputed. MIWAE (Mattei & Frellsen, 2019) is an (generative) approach which applies the importance weighted autoencoder framework and imputes missing data by optimizing the variational lower bound on log likelihood on observed data. notMIWAE (Ipsen et al., 2021) is an extension of MIWAE which incorporates prior information about the type of missingness, allowing modeling of the conditional distribution of the missingness pattern given the data, to try to effectively tackle the MNAR setting (particaully self-masking MNAR). Finally, Mean (Hawthorne & Elliott, 2005) imputes missing values with the column-wise unconditional mean.

*Table A7.* Default hyperparameter settings used for baseline methods.

| MODEL | HYPERPARAMETERS |
|---|---|
| HYPERIMPUTE | CLASS_THRESHOLD = 2, BASELINE_IMPUTER = 0, OPTIMIZER = "SIMPLE" |
| GAIN | BATCH_SIZE = 256, N_EPOCHS = 1000, HINT_RATE = 0.9, LOSS_ALPHA = 10 |
| ICE | MAX_ITER = 500 |
| MEAN | NONE |
| MICE | N_IMPUTATIONS = 1, MAX_ITER = 100, TOL = 0.001 |
| MISSFOREST | N_ESTIMATORS = 10, MAX_ITER = 500 |
| MIWAE | N_EPOCHS = 500, BATCH_SIZE = 256, LATENT_SIZE = 1, N_HIDDEN = 1, K = 20 |
| SINKHORN | EPS = 0.01, LR = 1E-3, OPT = TORCH.OPTIM.ADAM, N_EPOCHS = 500, BATCH_SIZE = 256, N_PAIRS = 1, NOISE = 1E-2, SCALING = 0.9 |
| SOFTIMPUTE | MAXIT = 1000, CONVERGENCE_THRESHOLD = 1E-5, MAX_RANK = 2, SHRINK_LAMBDA = 0, CV_LEN = 3, RANDOM_STATE = 0 |
| REMASKER | MAX_EPOCHS = 300, BATCH_SIZE = 64, MASK_RATIO = 0.5, EMBED_DIM = 32, DEPTH = 6, DECODER_DEPTH = 4, NUM_HEADS = 4, MLP_RATIO = 4, ENCODER_FUNC = 'LINEAR', WEIGHT_DECAY = 0.05, BASE_LR = 1E-3, MIN_LR = 1E-5, WARMUP_EPOCHS = 40 |
| DIFFPUTER | MAX_ITER = 10, RATIO = 30, HID_DIM = 1024, NUM_TRIALS = 10, NUM_STEPS = 50 |
| AUTOCOMPLETE | LR = 0.001, BATCH_SIZE = 1024, EPOCHS = 300, MOMENTUM = 0.9, ENCODING_RATIO = 1, DEPTH = 1, COPYMASK_AMOUNT = 0.5, NUM_TORCH_THREADS = 8, SIMULATE_MISSING = 0.01 |
| NOTMIWAE | N_HIDDEN=128, N_SAMPLES=20, BATCH_SIZE=16, EMBEDDING_SIZE=20, MISSING_PROCESS=SELFMASKING_KNOWN |

## C.2. CACTI hyperparameter configuration

*Table A8.* Default Parameters for CACTI

| MODEL | PARAMETER | SETTING |
|---|---|---|
| GLOBAL | OPTIMIZER | ADAMW |
| | INITIAL LEARNING RATE | 1E-3 |
| | LR SCHEDULER | STEP WISE WARMUP COSINE ANNEALING |
| | BETAS (GRAD MOMENTS DECAY) | (0.90, 0.95) |
| | WARMUP EPOCHS | 50 |
| | GRADIENT CLIPPING THRESHOLD | 5.0 |
| | TRAINING EPOCHS | 300 |
| | BATCH SIZE | 128 |
| | MASKING RATIO ($p_{cm}$) | 0.90 |
| ENCODER | DEPTH ($N_e$) | 10 |
| | EMBEDDING WIDTH ($E$) | 64 |
| | NUMBER OF HEADS | 8 |
| DECODER | DEPTH ($N_d$) | 4 |
| | EMBEDDING WIDTH ($E$) | 64 |
| | NUMBER OF HEADS | 8 |
| CONTEXT EMBEDDINGS | MODEL | GTEV1.5-EN-MLM-LARGE-8192 |
| | EMBEDDING SIZE ($dim(\mathbf{C}_k^{ci})$) | 1024 |
| | CONTEXT EMBEDDING RATIO ($\frac{C}{E}$) | 0.25 |

## C.3. Datasets details

To evaluate the performance of CACTI across a diverse set of data types, we chose datasets that contain only continuous features, as well as datasets that contain some combination of categorical, binary, and integer features, labeled as mixed. Also, to demonstrate robustness across datasets of different feature counts and dataset sizes, we benchmark across datasets ranging from 8 features to 57 features, as well as datasets ranging from 2,111 samples to 47,621 samples.

Baseline benchmarking studies are conducted on all 10 datasets which are fully observed. The ablation and sensitivity analysis are conducted on the four following datasets: *bike*, *default*, *spam* and *students*.

*Table A9.* Dataset summary

| NAME | FEATURE COUNT | TRAIN SPLIT SIZE | TEST SPLIT SIZE | TOTAL SIZE | FEATURE TYPE | FEAT. DESC. |
|---|---|---|---|---|---|---|
| *California* HOUSING | 8 | 16,512 | 4,128 | 20,640 | CONTINUOUS ONLY | YES |
| *Magic* GAMMA TELESCOPE | 10 | 15,216 | 3,804 | 19,020 | CONTINUOUS ONLY | YES |
| *Spam* BASE | 57 | 3,680 | 921 | 4,601 | CONTINUOUS ONLY | NO |
| *Letter* RECOGNITION | 16 | 16,000 | 4,000 | 20,000 | CONTINUOUS ONLY | YES |
| ESTIMATION OF *Obesity* LEVELS | 16 | 1,688 | 423 | 2,111 | MIXED | YES |
| SEOUL *Bike* SHARING DEMAND | 12 | 7,008 | 1,752 | 8,760 | MIXED | NO |
| *Default* OF CREDIT CARD CLIENTS | 23 | 24,000 | 6,000 | 30,000 | MIXED | NO |
| ADULT *Income* | 14 | 38,096 | 9,525 | 47,621 | MIXED | YES (SOMETIMES) |
| ONLINE *Shoppers* PURCHASING INTENTION | 17 | 9,864 | 2,466 | 12,330 | MIXED | NO |
| PREDICT *Students'* DROPOUT AND ACADEMIC SUCCESS | 36 | 3,539 | 885 | 4,424 | MIXED | YES |

## D. Extended Benchmarking Results

This section provides additional results to demonstrate CACTI's performance against the 13 baseline methods as measured by $R^2$, RMSE, and WD across 10 datasets, 3 missingness scenarios (MCAR, MAR, and MNAR), and 4 missingness ratios (0.1,0.3,0.5,0.7). Furthermore, we show performance on both the train split and the test split, demonstrating performance in both in sample and out of sample scenarios.

Here we point out that a method that is missing from the $R^2$ plots (not to be confused with $R^2 \approx 0$) implies lack of convergence due to the loss function taking on NaN values. Notably, at missingness rates $\geq 30\%$ ReMasker, DiffPuter, notMIWAE and AutoComplete show convergence difficulties for one or more datasets using their recommend parameter settings. In particular, ReMasker failed in all MCAR and MNAR datasets at 70% simulated missingness and in ***shoppers*** in MCAR and MNAR at 30%. DiffPuter also fails in ***shoppers*** at 30% simulated missingness under all 3 missingness settings. notMIWAE fails on ***income*** at MCAR 30% and ***spam*** in almost all settings except for MCAR 70%. Finally, AutoComplete fails on the ***income*** dataset under MNAR and MCAR at 70% simulated missingness. We made sure to re-run these failed runs at least twice to rule out random chance or a hardware issue. We report these failed runs to ensure transparency and did not adjust the recommend/default parameters to try to force these methods to not converge to NaN to ensure a fair comparison with all other methods which successfully ran and converged on all datasets.

Table A10. Average performance comparison with standard errors (in parenthesis) of 15 imputation methods on the train/test splits (separated by |) over 10 datasets at 10% missingness. Metrics (arrows indicate direction of better performance) evaluated under the MAR, MCAR, and MNAR conditions. – indicates method cannot perform out-of-samples imputation. Best in bold and second best underlined.

| METHOD | R² (↑) MCAR | R² (↑) MAR | R² (↑) MNAR | RMSE (↓) MCAR | RMSE (↓) MAR | RMSE (↓) MNAR | WD (↓) MCAR | WD (↓) MAR | WD (↓) MNAR |
|---|---|---|---|---|---|---|---|---|---|
| CACTI | **0.528\|0.538** (0.002)\|(0.004) | **0.520\|0.528** (0.011)\|(0.012) | **0.533\|0.543** (0.004)\|(0.005) | **0.590\|0.548** (0.005)\|(0.006) | **0.684\|0.674** (0.018)\|(0.008) | **0.641\|0.623** (0.018)\|(0.022) | **4.020\|4.098** (0.018)\|(0.048) | **2.172\|2.287** (0.050)\|(0.062) | **4.128\|4.307** (0.031)\|(0.028) |
| CMAE | 0.513\|0.524 (0.005)\|(0.004) | 0.495\|0.516 (0.020)\|(0.004) | 0.513\|0.525 (0.004)\|(0.004) | 0.604\|0.561 (0.007)\|(0.011) | 0.713\|0.698 (0.036)\|(0.008) | 0.661\|0.638 (0.021)\|(0.021) | 4.185\|4.246 (0.033)\|(0.046) | 2.259\|2.389 (0.055)\|(0.064) | 4.346\|4.532 (0.040)\|(0.037) |
| ReMasker | 0.492\|0.502 (0.002)\|(0.003) | 0.483\|0.490 (0.011)\|(0.017) | 0.480\|0.491 (0.004)\|(0.004) | 0.637\|0.599 (0.005)\|(0.007) | 0.730\|0.715 (0.011)\|(0.018) | 0.698\|0.678 (0.008)\|(0.021) | 5.242\|5.289 (0.040)\|(0.032) | 2.854\|2.934 (0.071)\|(0.063) | 5.507\|5.681 (0.124)\|(0.115) |
| DiffPuter | 0.453\|0.475 (0.002)\|(0.003) | 0.409\|0.417 (0.016)\|(0.003) | 0.428\|0.449 (0.006)\|(0.004) | 2.912\|2.637 (1.814)\|(0.006) | 0.847\|0.834 (0.019)\|(0.027) | 0.831\|0.746 (0.041)\|(0.020) | 5.086\|4.584 (0.449)\|(0.026) | 2.793\|2.918 (0.089)\|(0.079) | 4.893\|5.022 (0.038)\|(0.048) |
| HyperImpute | 0.525\|– (0.004)\|– | 0.502\|– (0.012)\|– | 0.505\|– (0.003)\|– | 0.697\|– (0.011)\|– | 0.798\|– (0.023)\|– | 0.752\|– (0.013)\|– | 6.154\|– (0.192)\|– | 3.374\|– (0.193)\|– | 6.032\|– (0.213)\|– |
| MissForest | 0.436\|0.429 (0.003)\|(0.005) | 0.421\|0.408 (0.010)\|(0.015) | 0.414\|0.414 (0.004)\|(0.004) | 0.805\|0.684 (0.009)\|(0.005) | 0.901\|0.839 (0.017)\|(0.010) | 0.851\|0.786 (0.010)\|(0.021) | 8.591\|7.062 (0.161)\|(0.044) | 4.616\|4.182 (0.125)\|(0.069) | 8.701\|7.699 (0.189)\|(0.029) |
| NotMIWAE | 0.385\|0.393 (0.004)\|(0.004) | 0.319\|0.331 (0.013)\|(0.013) | 0.333\|0.348 (0.003)\|(0.003) | 0.715\|0.681 (0.006)\|(0.005) | 0.881\|0.891 (0.011)\|(0.019) | 0.827\|0.815 (0.006)\|(0.021) | 5.260\|5.158 (0.039)\|(0.055) | 3.209\|3.334 (0.045)\|(0.056) | 6.050\|6.071 (0.060)\|(0.063) |
| Sinkhorn | 0.299\|– (0.001)\|– | 0.302\|– (0.014)\|– | 0.297\|– (0.002)\|– | 0.904\|– (0.008)\|– | 0.961\|– (0.019)\|– | 0.938\|– (0.010)\|– | 6.065\|– (0.155)\|– | 3.405\|– (0.145)\|– | 6.188\|– (0.213)\|– |
| ICE | 0.386\|0.383 (0.002)\|(0.003) | 0.374\|0.360 (0.016)\|(0.003) | 0.364\|0.370 (0.004)\|(0.004) | 0.804\|0.687 (0.010)\|(0.006) | 0.890\|0.826 (0.019)\|(0.011) | 0.848\|0.783 (0.021)\|(0.021) | 7.282\|5.714 (0.159)\|(0.043) | 3.742\|3.162 (0.137)\|(0.068) | 7.063\|5.918 (0.214)\|(0.045) |
| AutoComplete | 0.290\|0.274 (0.004)\|(0.007) | 0.281\|0.278 (0.013)\|(0.016) | 0.264\|0.271 (0.003)\|(0.006) | 0.882\|0.939 (0.004)\|(0.067) | 1.007\|1.007 (0.020)\|(0.006) | 0.976\|0.968 (0.021)\|(0.006) | 8.762\|10.348 (0.046)\|(0.051) | 4.708\|6.020 (0.081)\|(0.108) | 8.581\|11.224 (0.064)\|(0.067) |
| MICE | 0.247\|0.251 (0.002)\|(0.003) | 0.240\|0.237 (0.012)\|(0.012) | 0.242\|0.246 (0.002)\|(0.002) | 1.026\|1.059 (0.009)\|(0.013) | 1.163\|1.364 (0.026)\|(0.054) | 1.066\|1.176 (0.013)\|(0.028) | 9.284\|7.880 (0.169)\|(0.242) | 5.219\|4.621 (0.128)\|(0.271) | 9.568\|8.296 (0.271)\|(0.477) |
| GAIN | 0.213\|0.248 (0.003)\|(0.004) | 0.206\|0.247 (0.010)\|(0.014) | 0.238\|0.266 (0.003)\|(0.002) | 0.942\|0.818 (0.011)\|(0.008) | 1.067\|0.990 (0.019)\|(0.019) | 1.002\|0.923 (0.020)\|(0.013) | 8.795\|8.013 (0.170)\|(0.104) | 5.049\|4.930 (0.122)\|(0.122) | 9.255\|8.977 (0.189)\|(0.095) |
| SoftImpute | 0.114\|0.132 (0.003)\|(0.002) | 0.104\|0.112 (0.003)\|(0.007) | 0.114\|0.127 (0.003)\|(0.002) | 0.990\|0.892 (0.009)\|(0.006) | 1.147\|1.096 (0.019)\|(0.019) | 1.062\|1.010 (0.008)\|(0.020) | 8.282\|8.278 (0.132)\|(0.158) | 4.955\|5.087 (0.160)\|(0.109) | 8.767\|8.914 (0.088)\|(0.095) |
| MIWAE | 0.042\|0.042 (0.001)\|(0.002) | 0.037\|0.042 (0.007)\|(0.002) | 0.036\|0.039 (0.002)\|(0.002) | 1.158\|1.126 (0.013)\|(0.034) | 1.319\|1.328 (0.019)\|(0.020) | 1.226\|1.201 (0.008)\|(0.023) | 12.619\|12.458 (0.132)\|(0.144) | 6.930\|7.046 (0.155)\|(0.160) | 13.109\|13.135 (0.155)\|(0.163) |
| MEAN | 0.000\|0.002 (0.000)\|(0.001) | 0.000\|0.000 (0.000)\|(0.000) | 0.000\|0.003 (0.000)\|(0.001) | 0.991\|0.948 (0.010)\|(0.016) | 1.122\|1.120 (0.023)\|(0.009) | 1.060\|1.044 (0.021)\|(0.023) | (0.167)\|(0.170) | (0.112)\|(0.136) | (0.145)\|(0.163) |

*Table A11.* Average performance comparison with standard errors (in parenthesis) of 15 imputation methods on the train/test splits (separated by |) over 10 datasets at 30% missingness. Metrics (arrows indicate direction of better performance) evaluated under the MAR, MCAR, and MNAR conditions. – indicates method cannot perform out-of-samples imputation. Best in bold and second best underlined.

| METHOD | $R^2$ (↑) | | | RMSE (↓) | | | WD (↓) | | |
|---|---|---|---|---|---|---|---|---|---|
| | MCAR | MAR | MNAR | MCAR | MAR | MNAR | MCAR | MAR | MNAR |
| CACTI | **0.456\|0.461** (0.002\|0.003) | **0.468\|0.470** (0.003\|0.010) | **0.461\|0.456** (0.004\|0.004) | **0.662\|0.641** (0.008\|0.005) | **0.670\|0.686** (0.004\|0.006) | **0.680\|0.666** (0.006\|0.006) | 4.346\|**4.459** (0.018\|0.027) | **1.870\|1.942** (0.033\|0.030) | **4.449\|4.568** (0.029\|0.027) |
| CMAE | 0.441\|0.447 (0.002\|0.002) | 0.459\|0.460 (0.010\|0.011) | 0.440\|0.439 (0.010\|0.003) | 0.673\|0.653 (0.008\|0.007) | 0.685\|0.696 (0.016\|0.016) | 0.699\|0.691 (0.004\|0.006) | 4.399\|4.500 (0.032\|0.033) | 1.941\|2.017 (0.033\|0.030) | 4.568\|4.688 (0.033\|0.030) |
| reMasker | 0.437\|0.438 (0.002\|0.002) | 0.445\|0.443 (0.009\|0.010) | 0.402\|0.402 (0.003\|0.004) | 0.681\|0.665 (0.008\|0.007) | 0.691\|0.712 (0.017\|0.016) | 0.729\|0.709 (0.005\|0.008) | 4.620\|4.719 (0.023\|0.032) | 2.461\|2.519 (0.033\|0.033) | 4.788\|4.928 (0.033\|0.033) |
| DiffPuter | 0.400\|0.415 (0.003\|0.004) | 0.386\|0.430 (0.010\|0.010) | 0.363\|0.372 (0.003\|0.004) | 0.731\|0.704 (0.008\|0.006) | 0.770\|0.752 (0.014\|0.005) | 0.794\|0.767 (0.006\|0.006) | 4.528\|4.559 (0.028\|0.031) | 2.552\|2.385 (0.064\|0.059) | 4.785\|4.900 (0.063\|0.066) |
| HyperImpute | 0.406\|– (0.003\|–) | 0.439\|– (0.008\|–) | 0.393\|– (0.005\|–) | 0.722\|– (0.007\|–) | 0.727\|– (0.017\|–) | 0.757\|– (0.006\|–) | **4.261\|–** (0.083\|–) | 2.459\|– (0.092\|–) | **4.302\|4.866** (0.075\|0.075) |
| MissForest | 0.346\|0.338 (0.003\|0.002) | 0.381\|0.362 (0.009\|0.010) | 0.337\|0.322 (0.003\|0.003) | 0.766\|0.753 (0.009\|0.006) | 0.789\|0.817 (0.012\|0.004) | 0.790\|0.782 (0.006\|0.006) | 6.776\|6.798 (0.011\|0.029) | 3.804\|3.827 (0.039\|0.050) | 7.010\|7.057 (0.024\|0.040) |
| notMIWAE | 0.347\|0.347 (0.005\|0.005) | 0.347\|0.348 (0.010\|0.011) | 0.289\|0.296 (0.008\|0.008) | 0.754\|0.737 (0.011\|0.006) | 0.801\|0.824 (0.012\|0.004) | 0.824\|0.795 (0.010\|0.010) | 5.559\|5.600 (0.060\|0.057) | 2.380\|2.391 (0.071\|0.068) | 6.262\|6.204 (0.167\|0.151) |
| Sinkhorn | 0.275\|– (0.001\|–) | 0.287\|– (0.008\|–) | 0.259\|– (0.002\|–) | 0.844\|– (0.008\|–) | 0.888\|– (0.015\|–) | 0.877\|– (0.005\|–) | 7.016\|– (0.010\|–) | 3.958\|– (0.045\|–) | 7.508\|– (0.039\|–) |
| ICE | 0.281\|0.275 (0.002\|0.002) | 0.341\|0.327 (0.009\|0.009) | 0.259\|0.251 (0.004\|0.003) | 0.856\|0.869 (0.007\|0.009) | 0.783\|0.834 (0.014\|0.021) | 0.932\|0.930 (0.018\|0.018) | 4.818\|5.179 (0.024\|0.076) | 2.743\|2.813 (0.041\|0.059) | 5.325\|5.667 (0.085\|0.114) |
| AutoComplete | 0.239\|0.240 (0.003\|0.002) | 0.288\|0.291 (0.009\|0.004) | 0.213\|0.210 (0.002\|0.002) | 0.882\|0.862 (0.008\|0.006) | 0.876\|0.895 (0.017\|0.013) | 0.940\|0.921 (0.011\|0.013) | 10.143\|10.181 (0.044\|0.063) | 5.035\|5.073 (0.057\|0.060) | 10.443\|10.421 (0.051\|0.043) |
| MICE | 0.188\|0.190 (0.002\|0.002) | 0.229\|0.232 (0.009\|0.009) | 0.182\|0.178 (0.002\|0.002) | 1.057\|1.044 (0.006\|0.004) | 1.044\|1.051 (0.010\|0.013) | 1.085\|1.074 (0.003\|0.013) | 8.248\|8.333 (0.041\|0.031) | 4.161\|4.229 (0.059\|0.056) | 8.344\|8.494 (0.035\|0.048) |
| GAIN | 0.186\|0.206 (0.003\|0.002) | 0.179\|0.215 (0.002\|0.002) | 0.166\|0.177 (0.002\|0.003) | 0.914\|0.856 (0.004\|0.007) | 0.954\|0.933 (0.011\|0.011) | 1.008\|0.962 (0.006\|0.009) | 7.726\|7.345 (0.064\|0.091) | 4.437\|4.105 (0.061\|0.063) | 9.527\|9.137 (0.107\|0.135) |
| SoftImpute | 0.093\|0.102 (0.000\|0.001) | 0.096\|0.105 (0.001\|0.006) | 0.091\|0.091 (0.002\|0.002) | 1.021\|0.957 (0.009\|0.007) | 1.063\|1.024 (0.011\|0.011) | 1.054\|0.987 (0.006\|0.009) | 8.345\|7.865 (0.150\|0.026) | 4.835\|4.457 (0.100\|0.038) | 8.728\|8.235 (0.078\|0.057) |
| MIWAE | 0.001\|0.002 (0.000\|0.000) | 0.000\|0.003 (0.000\|0.000) | 0.000\|0.002 (0.000\|0.000) | 0.998\|0.979 (0.008\|0.006) | 1.054\|1.073 (0.021\|0.013) | 1.026\|1.003 (0.008\|0.007) | 7.825\|7.899 (0.012\|0.032) | 4.531\|4.565 (0.043\|0.044) | 8.358\|8.374 (0.078\|0.057) |
| Mean | 0.000\|0.000 (0.000\|0.000) | 0.000\|0.000 (0.000\|0.000) | 0.000\|0.000 (0.000\|0.000) | 0.949\|0.930 (0.008\|0.006) | 1.005\|1.023 (0.014\|0.010) | 0.977\|0.954 (0.005\|0.006) | 11.956\|12.000 (0.012\|0.026) | 6.351\|6.380 (0.043\|0.046) | 12.248\|12.257 (0.053\|0.031) |

*Table A12.* Average performance comparison with standard errors (in parenthesis) of 15 imputation methods on the train/test splits (separated by |) over 10 datasets at 50% missingness. Metrics (arrows indicate direction of better performance) evaluated under the MAR, MCAR, and MNAR conditions. – indicates method cannot perform out-of-samples imputation. Best in bold and second best underlined.

| METHOD | $R^2$ (↑) | | | RMSE (↓) | | | WD (↓) | | |
|---|---|---|---|---|---|---|---|---|---|
| | MCAR | MAR | MNAR | MCAR | MAR | MNAR | MCAR | MAR | MNAR |
| CACTI | **0.354\|0.358** (0.001)\|(0.002) | **0.431\|0.435** (0.001)\|(0.002) | **0.354\|0.355** (0.001)\|(0.002) | **0.730\|0.723** (0.002)\|(0.005) | **0.683\|0.694** (0.002)\|(0.003) | **0.771\|0.755** (0.003)\|(0.009) | **5.128\|5.152** (0.042)\|(0.051) | **1.835\|1.905** (0.043)\|(0.042) | **5.209\|5.289** (0.029)\|(0.040) |
| CMAE | 0.341\|0.344 (0.002)\|(0.002) | 0.420\|0.423 (0.002)\|(0.003) | 0.338\|0.339 (0.002)\|(0.003) | 0.747\|0.743 (0.007)\|(0.015) | 0.692\|0.702 (0.005)\|(0.023) | 0.782\|0.766 (0.003)\|(0.009) | 5.331\|5.358 (0.056)\|(0.059) | 1.899\|1.980 (0.044)\|(0.046) | 5.359\|5.436 (0.032)\|(0.032) |
| ReMasker | 0.307\|0.311 (0.001)\|(0.007) | 0.397\|0.398 (0.007)\|(0.007) | 0.263\|0.265 (0.002)\|(0.003) | 0.761\|0.754 (0.008)\|(0.022) | 0.694\|0.710 (0.016)\|(0.025) | 0.838\|0.821 (0.003)\|(0.008) | 5.605\|5.610 (0.026)\|(0.043) | 2.770\|2.853 (0.089)\|(0.085) | 6.071\|6.124 (0.062)\|(0.053) |
| DiffPuter | 0.292\|0.296 (0.003)\|(0.003) | 0.332\|0.333 (0.007)\|(0.007) | 0.261\|0.266 (0.003)\|(0.004) | 0.844\|0.853 (0.002)\|(0.006) | 0.806\|0.813 (0.016)\|(0.025) | 0.879\|0.864 (0.003)\|(0.009) | 5.593\|5.550 (0.055)\|(0.059) | 2.924\|2.946 (0.068)\|(0.063) | 5.984\|5.972 (0.034)\|(0.038) |
| HyperImpute | 0.290\|– (0.003)\|– | 0.397\|– (0.002)\|– | 0.270\|– (0.015)\|– | 0.908\|– (0.009)\|– | 0.792\|– (0.018)\|– | 0.935\|– (0.013)\|– | 5.618\|– (0.089)\|– | 3.033\|– (0.100)\|– | 5.644\|– (0.199)\|– |
| MissForest | 0.261\|0.249 (0.001)\|(0.003) | 0.330\|0.318 (0.006)\|(0.002) | 0.247\|0.239 (0.006)\|(0.004) | 0.873\|0.812 (0.006)\|(0.003) | 0.836\|0.805 (0.018)\|(0.020) | 0.904\|0.850 (0.010)\|(0.010) | 7.715\|6.176 (0.133)\|(0.030) | 4.143\|3.611 (0.094)\|(0.057) | 7.656\|6.417 (0.173)\|(0.037) |
| NotMIWAE | 0.251\|0.253 (0.003)\|(0.003) | 0.302\|0.300 (0.006)\|(0.006) | 0.226\|0.229 (0.003)\|(0.004) | 0.809\|0.806 (0.007)\|(0.005) | 0.795\|0.810 (0.022)\|(0.027) | 0.876\|0.864 (0.006)\|(0.010) | 6.252\|6.242 (0.046)\|(0.050) | 3.055\|3.076 (0.081)\|(0.080) | 6.449\|6.445 (0.064)\|(0.074) |
| Sinkhorn | 0.154\|– (0.001)\|– | 0.209\|– (0.005)\|– | 0.139\|– (0.001)\|– | 1.007\|– (0.015)\|– | 0.961\|– (0.035)\|– | 1.023\|– (0.013)\|– | 6.827\|– (0.141)\|– | 3.552\|– (0.100)\|– | 6.822\|– (0.182)\|– |
| ICE | 0.232\|0.229 (0.002)\|(0.008) | 0.288\|0.279 (0.008)\|(0.008) | 0.207\|0.206 (0.002)\|(0.003) | 0.931\|0.870 (0.007)\|(0.004) | 0.863\|0.833 (0.018)\|(0.022) | 0.975\|0.923 (0.011)\|(0.009) | 6.535\|**4.858** (0.139)\|(0.041) | 3.522\|2.927 (0.081)\|(0.042) | 6.391\|**5.046** (0.173)\|(0.034) |
| AutoComplete | 0.193\|0.195 (0.003)\|(0.006) | 0.261\|0.259 (0.007)\|(0.002) | 0.173\|0.177 (0.003)\|(0.003) | 0.891\|0.890 (0.002)\|(0.007) | 0.847\|0.873 (0.012)\|(0.023) | 0.942\|0.930 (0.003)\|(0.010) | 10.552\|10.550 (0.177)\|(0.261) | 4.865\|4.941 (0.119)\|(0.100) | 10.733\|10.781 (0.266)\|(0.211) |
| MICE | 0.132\|0.130 (0.002)\|(0.007) | 0.191\|0.192 (0.008)\|(0.001) | 0.120\|0.121 (0.003)\|(0.003) | 1.132\|1.191 (0.009)\|(0.012) | 1.071\|1.164 (0.018)\|(0.029) | 1.155\|1.199 (0.012)\|(0.028) | 9.016\|9.803 (0.141)\|(0.176) | 4.457\|5.132 (0.099)\|(0.145) | 8.658\|9.544 (0.200)\|(0.405) |
| GAIN | 0.125\|0.133 (0.003)\|(0.007) | 0.136\|0.170 (0.010)\|(0.002) | 0.071\|0.065 (0.003)\|(0.003) | 1.147\|1.073 (0.012)\|(0.021) | 1.013\|0.961 (0.021)\|(0.013) | 1.386\|1.376 (0.013)\|(0.013) | 12.572\|10.876 (0.142)\|(0.183) | 4.998\|4.579 (0.111)\|(0.110) | 16.199\|16.287 (0.153)\|(0.136) |
| SoftImpute | 0.076\|0.076 (0.001)\|(0.005) | 0.079\|0.088 (0.005)\|(0.002) | 0.063\|0.061 (0.002)\|(0.001) | 1.070\|1.017 (0.007)\|(0.005) | 1.037\|0.996 (0.021)\|(0.024) | 1.116\|1.053 (0.008)\|(0.008) | 8.413\|7.772 (0.143)\|(0.102) | 4.703\|4.520 (0.062)\|(0.163) | 8.398\|7.947 (0.266)\|(0.052) |
| MIWAE | 0.023\|0.018 (0.000)\|(0.003) | 0.013\|0.013 (0.003)\|(0.002) | 0.019\|0.019 (0.002)\|(0.001) | 1.138\|1.172 (0.010)\|(0.027) | 1.166\|1.195 (0.030)\|(0.040) | 1.167\|1.167 (0.017)\|(0.020) | 8.381\|8.469 (0.142)\|(0.183) | 4.230\|4.305 (0.111)\|(0.110) | 8.368\|8.454 (0.153)\|(0.136) |
| Mean | 0.000\|0.000 (0.000)\|(0.000) | 0.000\|0.000 (0.000)\|(0.000) | 0.000\|0.000 (0.000)\|(0.000) | 0.989\|0.986 (0.006)\|(0.009) | 0.997\|1.011 (0.024)\|(0.017) | 1.023\|1.007 (0.020)\|(0.009) | 12.640\|12.638 (0.116)\|(0.122) | 6.340\|6.393 (0.110)\|(0.108) | 12.573\|12.583 (0.156)\|(0.154) |

*Table A13.* Average performance comparison with standard errors (in parenthesis) of 15 imputation methods on the train/test splits (separated by |) over 10 datasets at 70% missingness. Metrics (arrows indicate direction of better performance) evaluated under the MAR, MCAR, and MNAR conditions. – indicates method cannot perform out-of-samples imputation and NA indicates method failed. Best in bold and second best underlined.

| METHOD | $R^2$ (↑) MCAR | $R^2$ (↑) MAR | $R^2$ (↑) MNAR | RMSE (↓) MCAR | RMSE (↓) MAR | RMSE (↓) MNAR | WD (↓) MCAR | WD (↓) MAR | WD (↓) MNAR |
|---|---|---|---|---|---|---|---|---|---|
| CACTI | **0.227\|0.228**
(0.001)\|(0.001) | **0.386\|0.383**
(0.001)\|(0.009) | **0.227\|0.231**
(0.002)\|(0.002) | **0.827\|0.826**
(0.002)\|(0.009) | 0.746\|0.721
(0.002)\|(0.008) | **0.839\|0.825**
(0.004)\|(0.006) | 6.215\|6.262
(0.031)\|(0.048) | **1.740\|1.765**
(0.041)\|(0.045) | 5.971\|6.002
(0.037)\|(0.031) |
| CMAE | 0.222\|0.224
(0.001)\|(0.002) | 0.374\|0.374
(0.008)\|(0.002) | 0.214\|0.217
(0.002)\|(0.002) | 0.836\|0.836
(0.009)\|(0.008) | 0.755\|0.724
(0.010)\|(0.010) | 0.853\|0.841
(0.006)\|(0.008) | 6.440\|6.519
(0.041)\|(0.055) | 1.791\|1.801
(0.049)\|(0.051) | 6.114\|6.145
(0.037)\|(0.048) |
| REMASKER | (NA)(NA)
NA NA | 0.385\|0.385
(0.008)\|(0.008) | (NA)(NA)
NA NA | (NA)(NA)
NA NA | **0.723\|0.690**
(0.008)\|(0.010) | (NA)(NA)
NA NA | (NA)(NA)
NA NA | 2.792\|2.771
(0.056)\|(0.055) | (NA)(NA)
NA NA |
| DIFFPUTER | 0.176\|0.186
(0.008)\|(0.008) | 0.283\|0.289
(0.009)\|(0.010) | 0.158\|0.163
(0.009)\|(0.003) | 0.910\|0.912
(0.008)\|(0.010) | 0.850\|0.817
(0.010)\|(0.011) | 0.924\|0.926
(0.010)\|(0.007) | 6.798\|6.763
(0.056)\|(0.055) | 3.290\|3.229
(0.049)\|(0.135) | 7.191\|7.169
(0.033) |
| HYPERIMPUTE | 0.166\|–
(0.002)\|– | 0.336\|–
(0.010)\|– | 0.149\|–
(0.003)\|– | 1.041\|–
(0.018)\|– | 0.872\|–
(0.014)\|– | 1.055\|–
(0.014)\|– | **5.372\|–**
(0.109)\|– | 2.859\|–
(0.079)\|– | 5.790\|–
(0.168)\|– |
| MISSFOREST | 0.164\|0.157
(0.001)\|(0.002) | 0.301\|0.291
(0.008)\|(0.007) | 0.153\|0.150
(0.002)\|(0.002) | 0.955\|0.925
(0.008)\|(0.004) | 0.862\|0.798
(0.012)\|(0.011) | 0.977\|0.943
(0.011)\|(0.007) | 7.088\|5.394
(0.163)\|(0.076) | 3.869\|3.252
(0.084)\|(0.050) | 7.050\|5.743
(0.149)\|(0.045) |
| NOTMIWAE | 0.148\|0.153
(0.003)\|(0.003) | 0.251\|0.253
(0.009)\|(0.010) | 0.152\|0.154
(0.003)\|(0.004) | 0.885\|0.880
(0.009)\|(0.007) | 0.995\|0.966
(0.025)\|(0.033) | 0.924\|0.912
(0.011)\|(0.012) | 7.856\|7.835
(0.108)\|(0.137) | 3.981\|3.970
(0.149)\|(0.130) | 7.573\|7.579
(0.135) |
| SINKHORN | 0.078\|–
(0.001)\|– | 0.164\|–
(0.006)\|– | 0.067\|–
(0.001)\|– | 1.057\|–
(0.013)\|– | 0.959\|–
(0.010)\|– | 1.049\|–
(0.019)\|– | 7.502\|–
(0.108)\|– | 3.815\|–
(0.082)\|– | 7.408\|–
(0.161)\|– |
| ICE | 0.146\|0.135
(0.001)\|(0.002) | 0.265\|0.254
(0.008)\|(0.008) | 0.123\|0.120
(0.002)\|(0.003) | 0.996\|0.990
(0.008)\|(0.008) | 0.886\|0.832
(0.011)\|(0.012) | 1.028\|1.004
(0.009)\|(0.007) | 6.790\|**5.103**
(0.081)\|(0.163) | 3.218\|2.653
(0.076)\|(0.035) | 6.671\|**5.293**
(0.141)\|(0.056) |
| AUTO COMPLETE | 0.124\|0.154
(0.002)\|(0.005) | 0.286\|0.286
(0.008)\|(0.009) | 0.106\|0.104
(0.002)\|(0.004) | 0.923\|0.926
(0.005)\|(0.005) | 0.833\|0.805
(0.008)\|(0.010) | 0.954\|0.958
(0.027)\|(0.007) | 10.989\|10.122
(0.040)\|(0.062) | 4.582\|4.562
(0.022)\|(0.059) | 11.075\|11.242
(0.059) |
| MICE | 0.070\|0.071
(0.001)\|(0.001) | 0.203\|0.199
(0.008)\|(0.009) | 0.059\|0.063
(0.002)\|(0.002) | 1.198\|1.240
(0.021)\|(0.016) | 1.095\|1.187
(0.016)\|(0.038) | 1.206\|1.230
(0.012)\|(0.016) | 9.168\|9.583
(0.171)\|(0.237) | 4.153\|5.258
(0.104)\|(0.241) | 8.980\|9.247
(0.140)\|(0.191) |
| GAIN | 0.037\|0.038
(0.001)\|(0.001) | 0.108\|0.135
(0.008)\|(0.002) | 0.025\|0.025
(0.001)\|(0.002) | 1.425\|1.426
(0.009)\|(0.002) | 1.050\|0.992
(0.014)\|(0.018) | 1.472\|1.481
(0.020)\|(0.022) | 15.924\|15.685
(0.412)\|(0.127) | 5.204\|4.824
(0.132)\|(0.325) | 16.965\|17.240
(0.325)\|(0.365) |
| SOFTIMPUTE | 0.015\|0.013
(0.002)\|(0.001) | 0.087\|0.085
(0.008)\|(0.001) | 0.028\|0.029
(0.002)\|(0.002) | 1.185\|1.139
(0.014)\|(0.023) | 1.040\|0.993
(0.012)\|(0.015) | 1.221\|1.165
(0.007)\|(0.006) | 8.561\|8.010
(0.114)\|(0.103) | 4.643\|4.686
(0.076)\|(0.062) | 8.669\|8.296
(0.171)\|(0.062) |
| MIWAE | 0.020\|0.020
(0.006)\|(0.006) | 0.012\|0.011
(0.001)\|(0.002) | 0.012\|0.011
(0.001)\|(0.002) | 1.137\|1.147
(0.012)\|(0.016) | 1.148\|1.120
(0.016)\|(0.019) | 1.161\|1.201
(0.007)\|(0.015) | 12.625\|12.615
(0.097)\|(0.095) | 6.423\|6.379
(0.103)\|(0.164) | 12.47\|12.480
(0.150) |
| MEAN | 0.000\|0.000
(0.000)\|(0.000) | 0.000\|0.000
(0.000)\|(0.000) | 0.000\|0.000
(0.000)\|(0.000) | 0.987\|0.983
(0.012)\|(0.016) | 1.013\|0.978
(0.016)\|(0.015) | 1.002\|0.992
(0.008)\|(0.010) | 8.457\|8.408
(0.081)\|(0.097) | 4.209\|4.145
(0.095)\|(0.106) | 8.574\|8.666
(0.127)\|(0.130) |

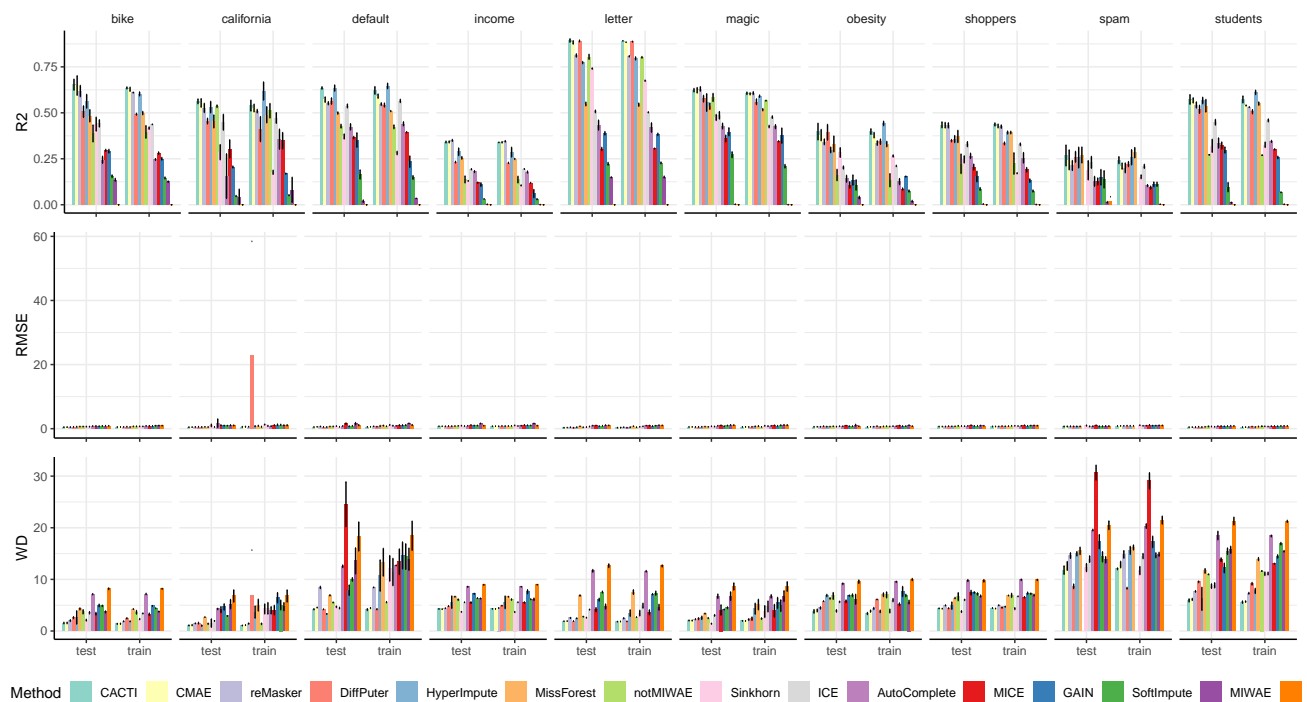

(a) MCAR at 10% missingness

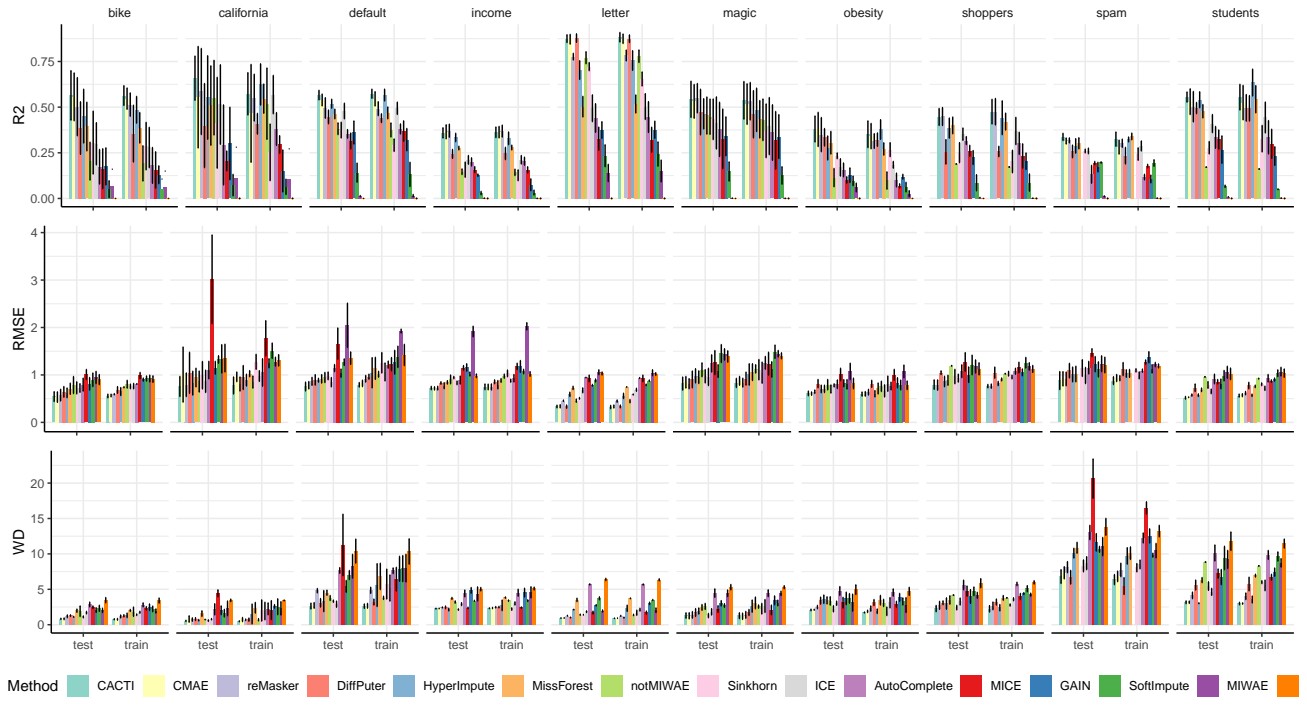

(b) MAR at 10% missingness

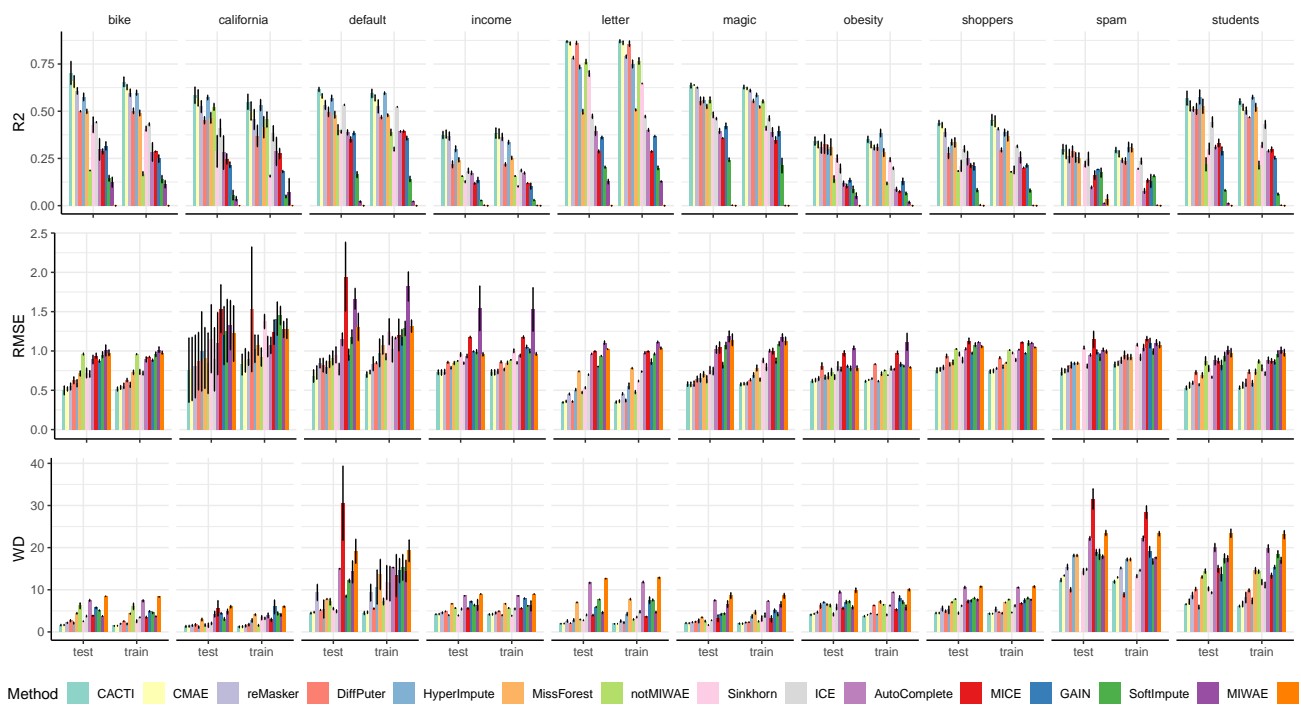

(c) MNAR at 10% missingness

*Figure A5.* Performance comparison of CACTI against 13 baseline methods. Experiments were performed on 10 datasets, under MCAR, MAR, and MNAR, at 10% missingness. Results shown as mean ± 95% CI.

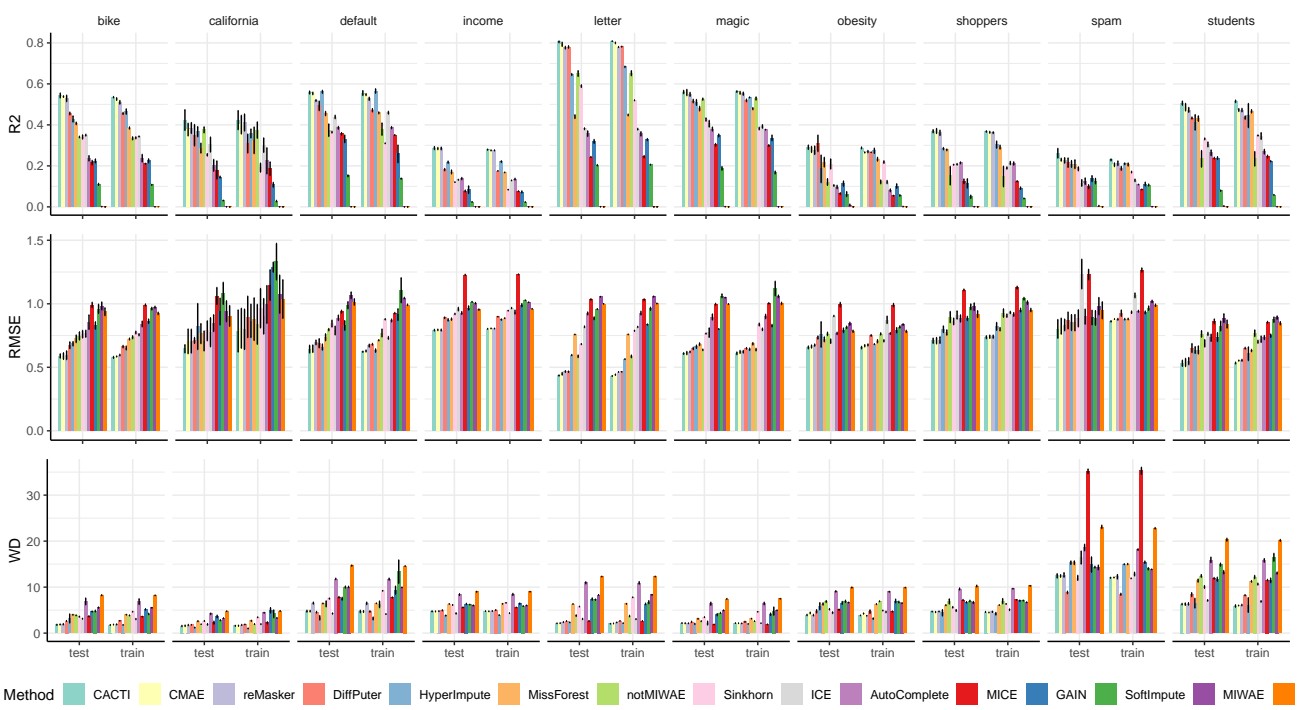

(a) MCAR at 30% missingness

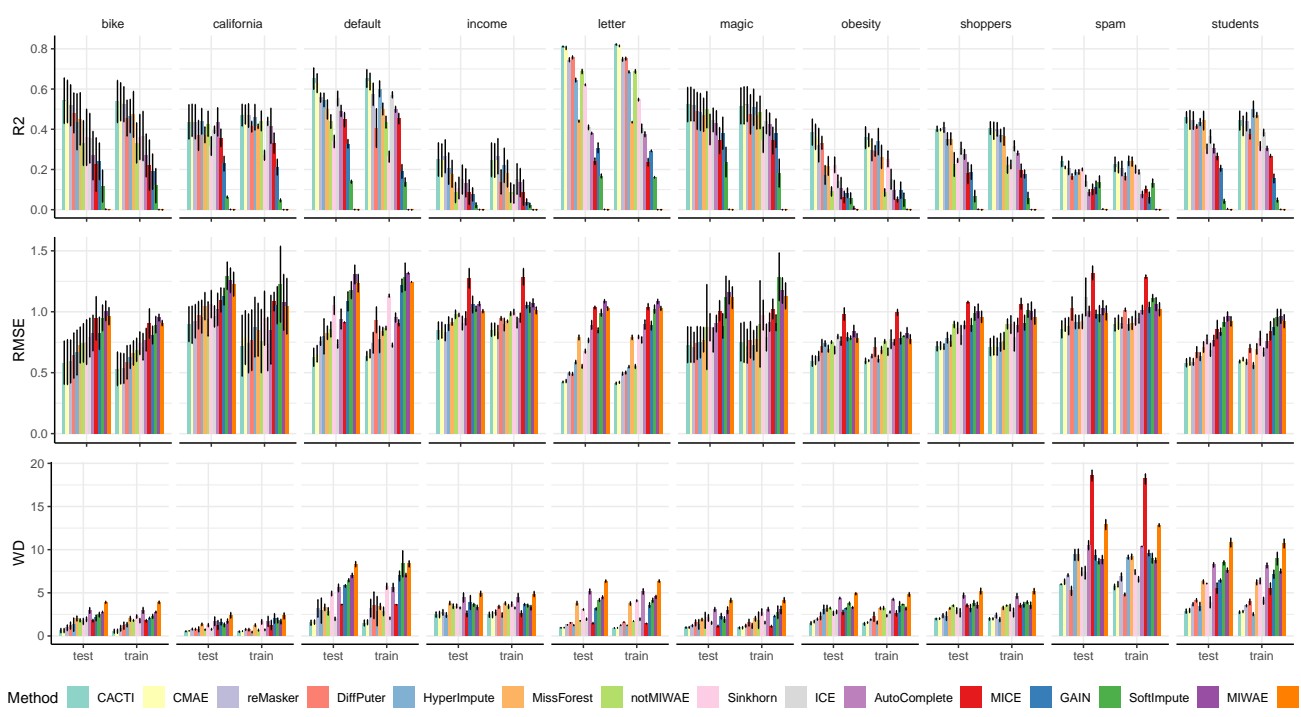

(b) MAR at 30% missingness

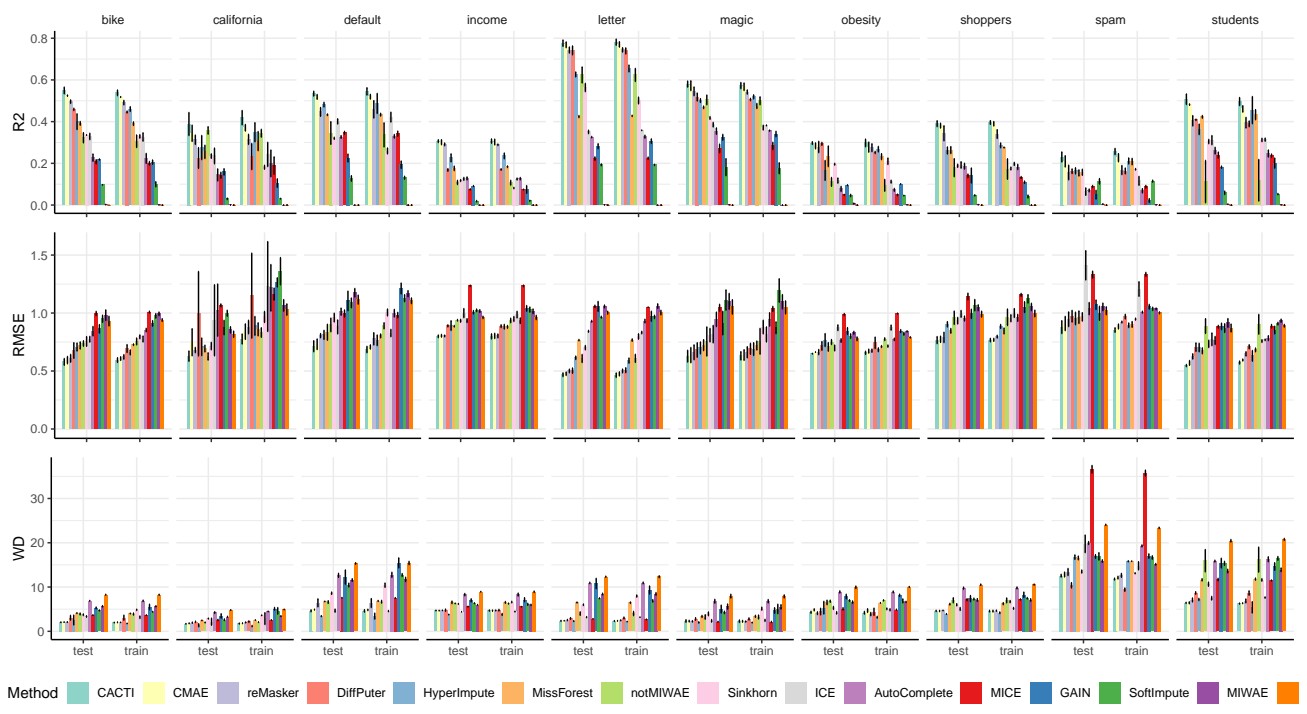

(c) MNAR at 30% missingness

*Figure A6.* Performance comparison of CACTI against 13 baseline methods. Experiments performed on 10 datasets, under MCAR, MAR, and MNAR, at 30% missingness. Results shown as mean ± 95% CI.

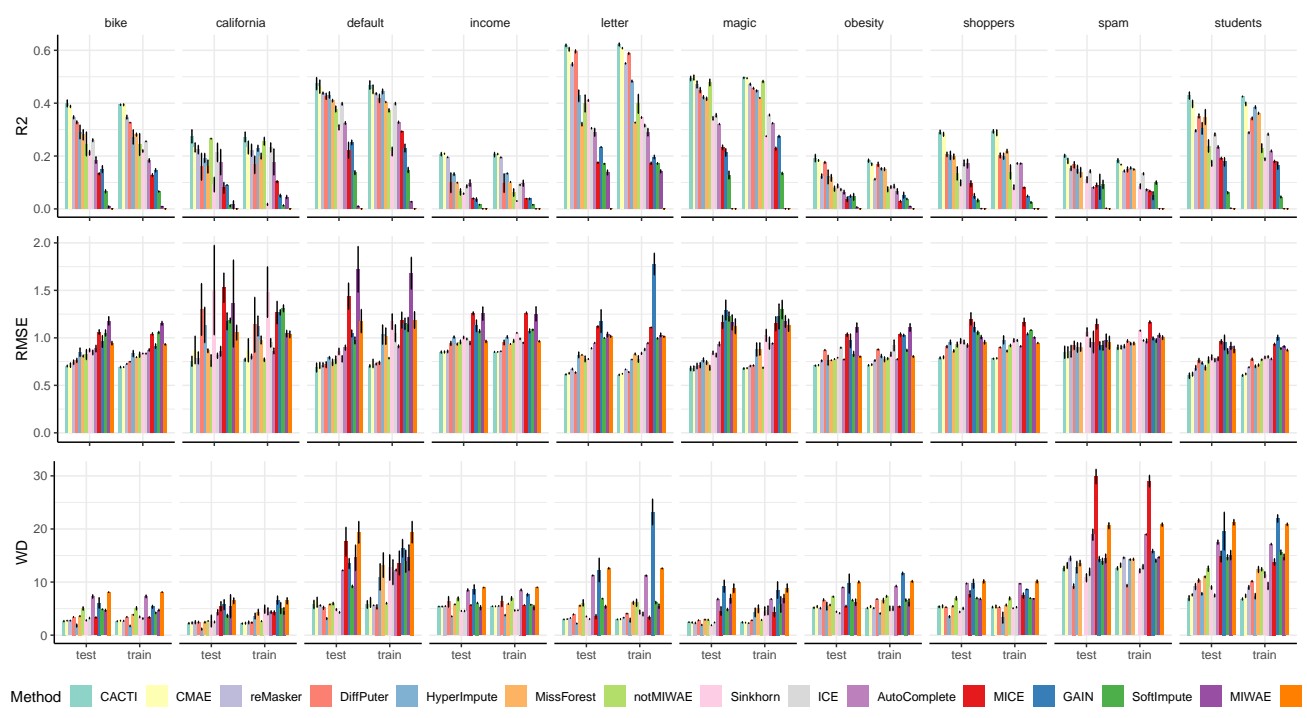

(a) MCAR at 50% missingness

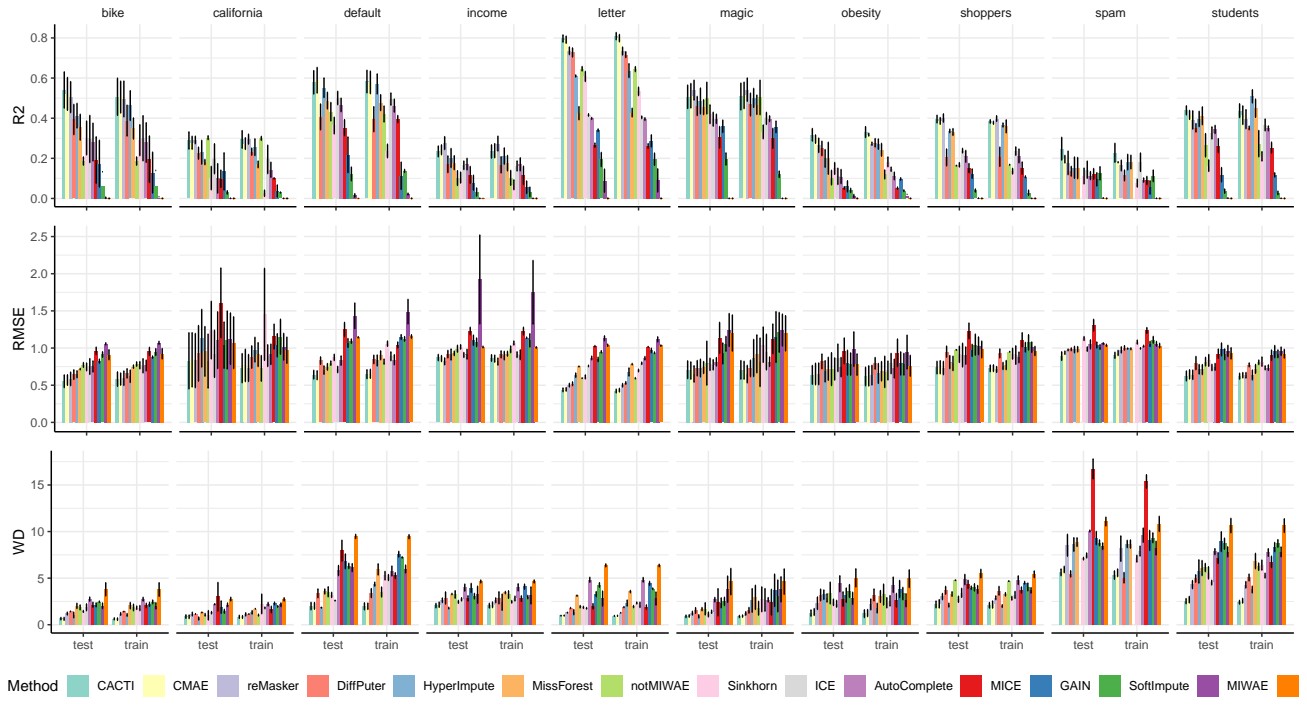

(b) MAR at 50% missingness

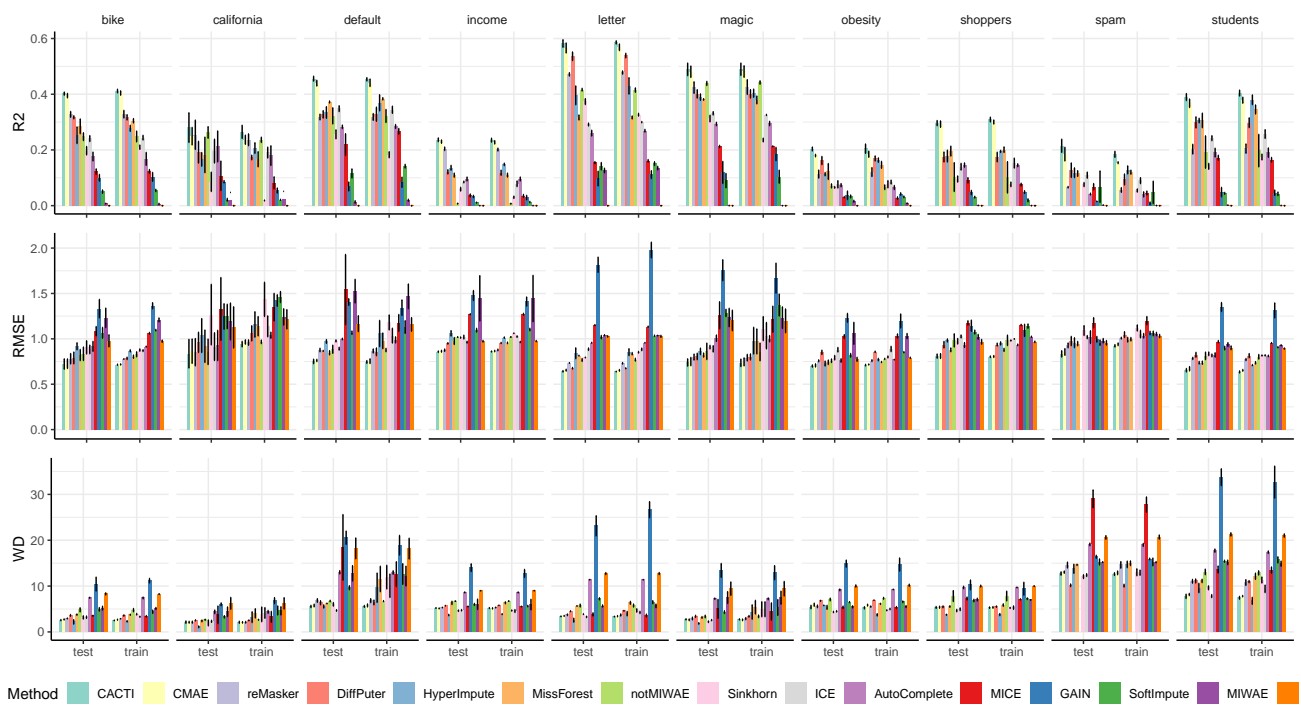

(c) MNAR at 50% missingness

*Figure A7.* Performance comparison of CACTI against 13 baseline methods. Experiments were performed on 10 datasets under MCAR, MAR, and MNAR at 50% missingness. Results shown as mean ± 95% CI.

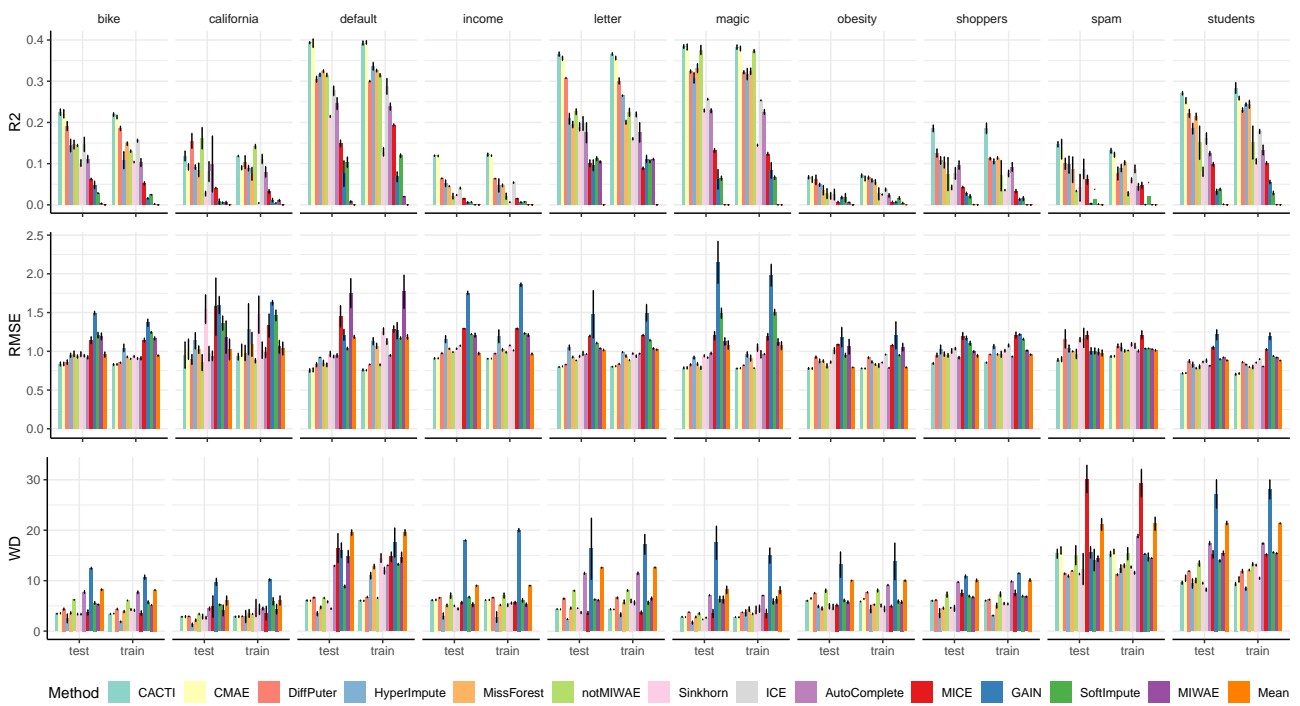

(a) MCAR at 70% missingness

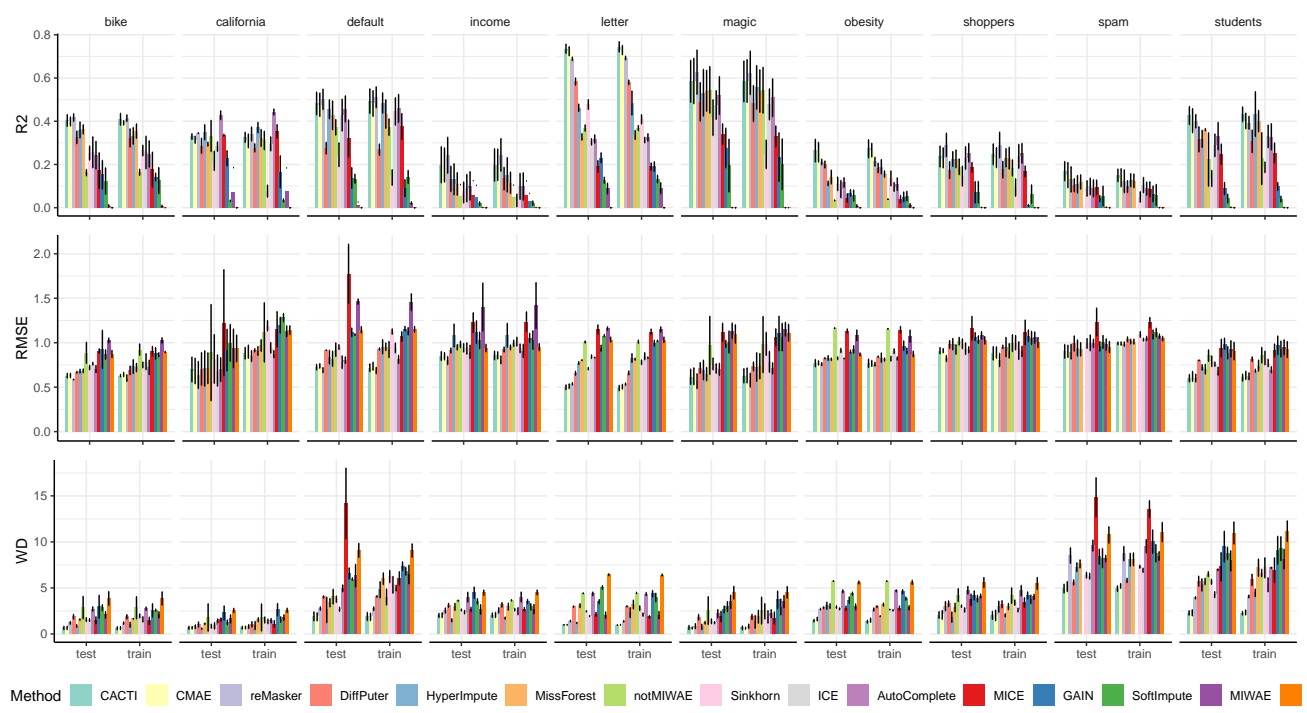

(b) MAR at 70% missingness

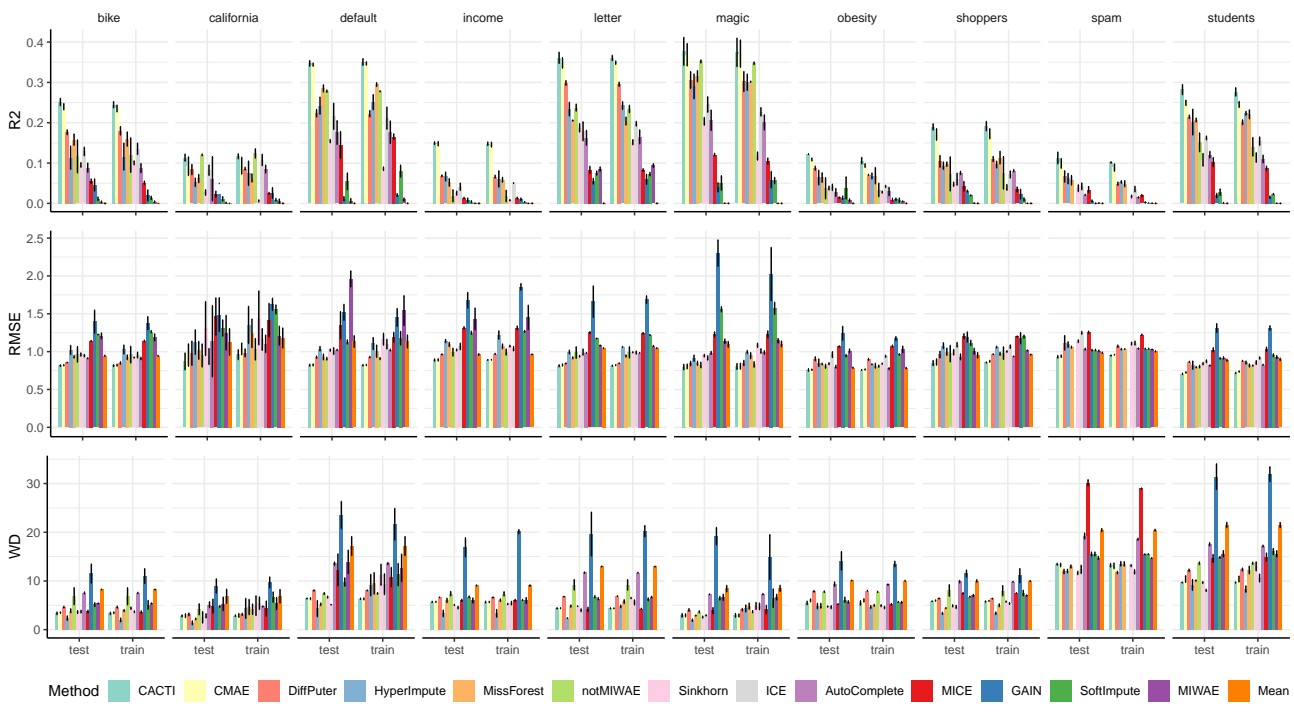

(c) MNAR at 70% missingness

*Figure A8.* Performance comparison of CACTI against 13 baseline methods. Experiments were performed on 10 datasets under MCAR, MAR, and MNAR at 70% missingness. Results shown as mean ± 95% CI.

*Table A14.* Comparison of runtime and memory statistics of CACTI with ReMasker on 10 datasets. Experiments were performed under MAR scenario at 30% missingness. The runtime is measured for the training and inference (on train/test split) stages in seconds (s) and peak GPU memory consumed in gigabytes (GB).

| METHOD | DATA | PER EPOCH (S) | INFER TRAIN SPLIT (S) | INFER TEST SPLIT (S) | PEAK GPU MEM (GB) |
|---|---|---|---|---|---|
| CACTI | OBESITY | 0.42 | 6.15 | 1.52 | 0.16 |
| | STUDENTS | 0.64 | 15.53 | 3.84 | 0.18 |
| | SPAM | 0.71 | 16.36 | 4.11 | 0.26 |
| | BIKE | 1.24 | 25.82 | 6.38 | 0.16 |
| | SHOPPERS | 1.68 | 34.85 | 8.67 | 0.16 |
| | MAGIC | 2.39 | 55.37 | 13.70 | 0.16 |
| | LETTER | 2.91 | 82.06 | 21.01 | 0.16 |
| | CALIFORNIA | 2.95 | 59.93 | 14.92 | 0.08 |
| | DEFAULT | 4.61 | 101.91 | 24.84 | 0.16 |
| | INCOME | 5.76 | 165.59 | 42.17 | 0.16 |
| REMASKER | OBESITY | 0.28 | 4.28 | 1.07 | 0.14 |
| | STUDENTS | 0.49 | 9.01 | 2.24 | 0.14 |
| | SPAM | 0.58 | 9.30 | 2.34 | 0.15 |
| | BIKE | 1.06 | 17.74 | 4.41 | 0.14 |
| | SHOPPERS | 1.55 | 25.46 | 6.30 | 0.14 |
| | MAGIC | 2.09 | 38.23 | 9.53 | 0.14 |
| | LETTER | 2.10 | 40.67 | 10.09 | 0.15 |
| | CALIFORNIA | 2.37 | 41.89 | 10.44 | 0.04 |
| | DEFAULT | 3.18 | 61.21 | 15.18 | 0.15 |
| | INCOME | 4.80 | 95.45 | 23.94 | 0.15 |

# E. Extended Ablation Analysis Results

In this section, we present additional ablation analysis results. First, Table A15 summarizes the results of a 3-way comparison (over all 10 datasets) between CACTI, CMAE (CACTI without context) and ReMasker (the strongest MAE with random masking) to quantify the magnitude and the statistical significances of the improvements driven by MT-CM, context awareness and the combination of the two. A one-sided paired t-test is used to evaluate the statistical significance of improvement of the target method over the baseline. Next, Figure A9 shows the per-dataset difference in $R^2$ performance between CACTI and CMAE across all datasets, missingness situations and simulated missingness percentages. We also perform additional ablation comparing the effects of calculating the loss over observed only values, masked only values and a combination of the two in Table A16.

These results are followed by the complete (per-dataset) results of all our ablation analysis, demonstrating the effects of (a) MT-CM, RM, and/or CTX and (b) loss function on model performance. For this experiment we use 4 UCI datasets (bike, default, spam, and students), each under three missingness scenarios (MCAR, MAR, and MNAR), with a simulated missingness ratio of 0.3. We measure performance using $R^2$, RMSE, and WD, on both the train and test split.

*Table A15.* Paired t-test to evaluate statistical significance of gain in performance between CACTI, CMAE (CACTI w/o context) and ReMasker.

| MISSINGNESS | TARGET METHOD | BASELINE METHOD | AVG. $R^2$ GAIN | P-VALUE |
|---|---|---|---|---|
| ALL | CACTI | REMASKER | 0.034 | 4.4E-7 |
| | CACTI | CMAE | 0.013 | 1.1E-5 |
| | CMAE | REMASKER | 0.021 | 4.2E-5 |
| MCAR | CACTI | REMASKER | 0.023 | 5.5E-4 |
| | CACTI | CMAE | 0.014 | 3.E-3 |
| | CMAE | REMASKER | 0.017 | 8.9E-3 |
| MAR | CACTI | REMASKER | 0.025 | 2.3E-2 |
| | CACTI | CMAE | 0.007 | 9.4E-2 |
| | CMAE | REMASKER | 0.018 | 4.8E-2 |
| MNAR | CACTI | REMASKER | 0.054 | 1.1E-4 |
| | CACTI | CMAE | 0.017 | 8.7E-4 |
| | CMAE | REMASKER | 0.037 | 4.6E-4 |

*Table A16.* **Loss ablations.** Effect of the loss function on accuracy. Metrics represent the average across four datasets (30% missingness).

| LOSS TYPE | $R^2$ ($\uparrow$) | | RMSE ($\downarrow$) | |
|---|---|---|---|---|
| | MAR | MNAR | MAR | MNAR |
| $\mathcal{L}_{\mathbb{O}} + \mathcal{L}_{\mathbb{M}}$ | **0.46** | **0.46** | **0.68** | **0.67** |
| $\mathcal{L}_{\mathbb{M}}$ | 0.41 | 0.43 | 0.71 | 0.70 |
| $\mathcal{L}_{\mathbb{O}}$ | 0.03 | 0.04 | 2.67 | 2.93 |

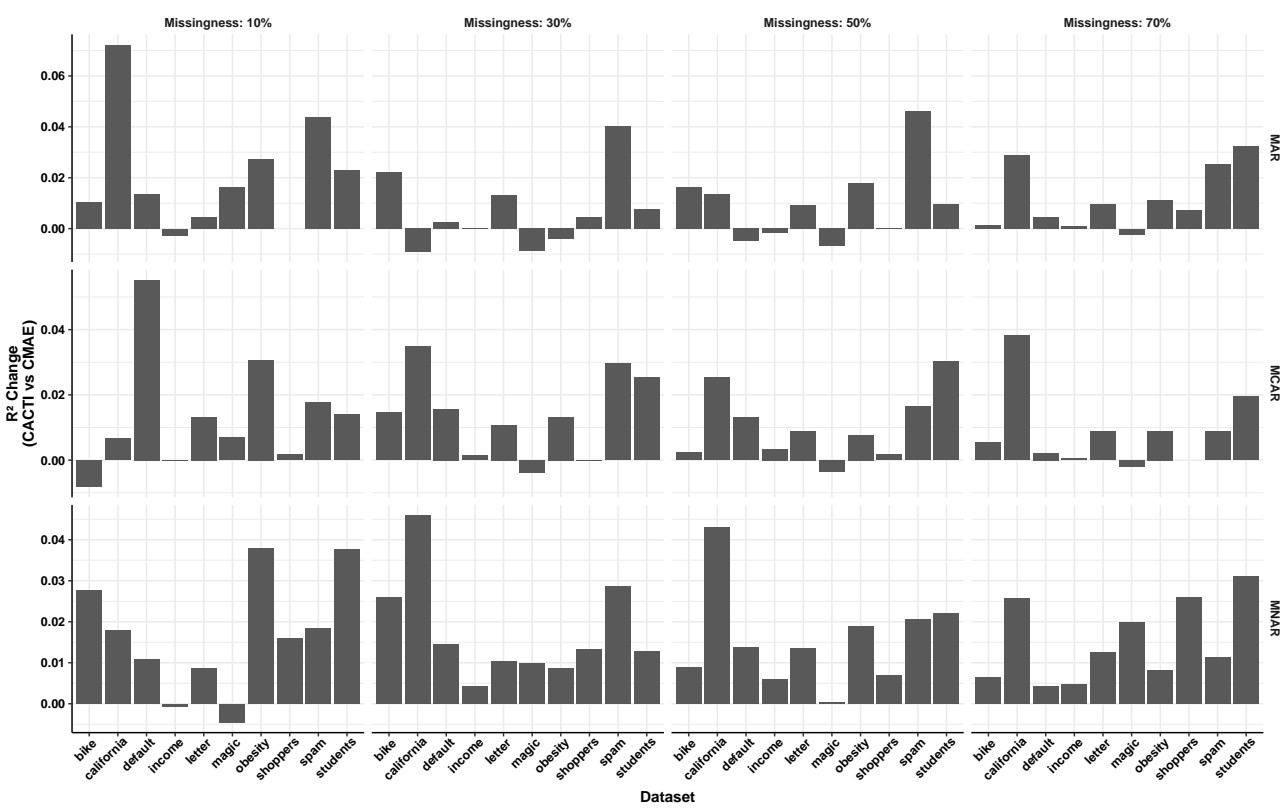

*Figure A9.* Difference in $R^2$ between CACTI and CMAE across all 10 datasets, missingness conditions and simulated missingness percentages. $R^2$ change $> 0$ indicates CACTI is better than CMAE under the respective setting.

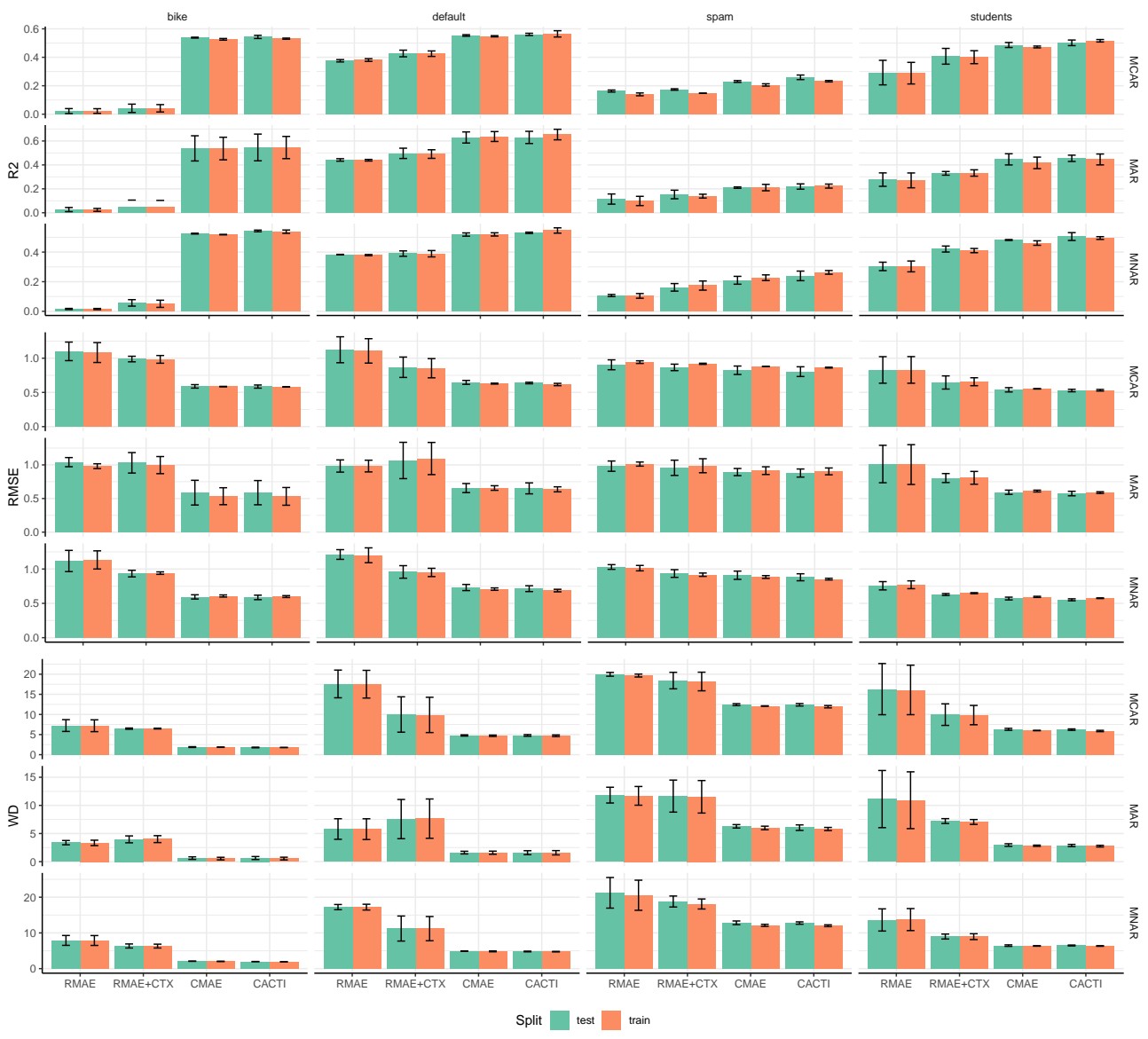

*Figure A10.* Experiments performed across four datasets split into train/test, under MCAR, MAR, and MNAR, at 30% missingness. Metrics ($R^2$, RMSE, WD) are reported as mean $\pm$ 95% CI. Ablations demonstrate how model performance is affected by MT-CM, RM, and/or CTX.

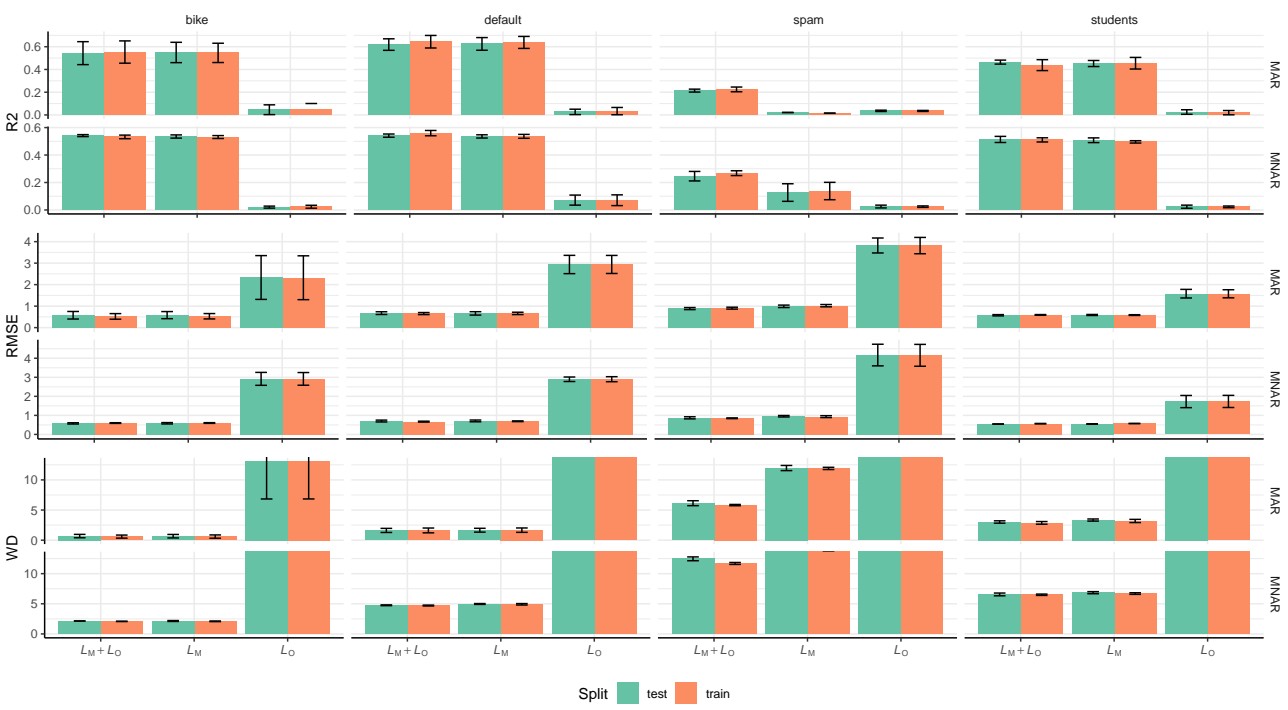

*Figure A11.* Experiments performed across four datasets split into train/test, under MCAR, MAR, and MNAR, at 30% missingness. Metrics ($R^2$, RMSE, WD) are reported as mean ± 95% CI. Ablations demonstrate how model performance is affected by the loss function.

## F. Extended sensitivity analysis results

In this section, we present the present the average sensitivity analysis results for context embedding proportion (Table A17) and masking rate (Figure A12). All sensitivity analysis experiments are performed on 4 UCI datasets (bike, default, spams, students), under 2 missingness scenarios (MAR and MNAR), and 0.3 missingness ratio, split into train and test splits. Performance is measured using $R^2$, RMSE, and WD.

These results are followed by the complete (per-dataset) results of all our sensitivity analysis. Figure A13, Figure A14, Figure A15, Figure A16, Figure A17, and Figure A18 show the effects of independently varying MT-CM rate, encoder depth, decoder depth, embedding size, context embedding model, and context embedding proportion respectively.

*Table A17.* **Context embedding sensitivity.** Average performance effect of context (CTX) proportions (30% missing).

| CTX PROPORTION | $R^2$ (↑) | | RMSE (↓) | |
|---|---|---|---|---|
| | MAR | MNAR | MAR | MNAR |
| 0.25 | 0.47 | **0.46** | **0.66** | **0.67** |
| 0.50 | **0.48** | **0.46** | **0.66** | 0.68 |
| 0.75 | 0.46 | **0.46** | 0.67 | 0.68 |

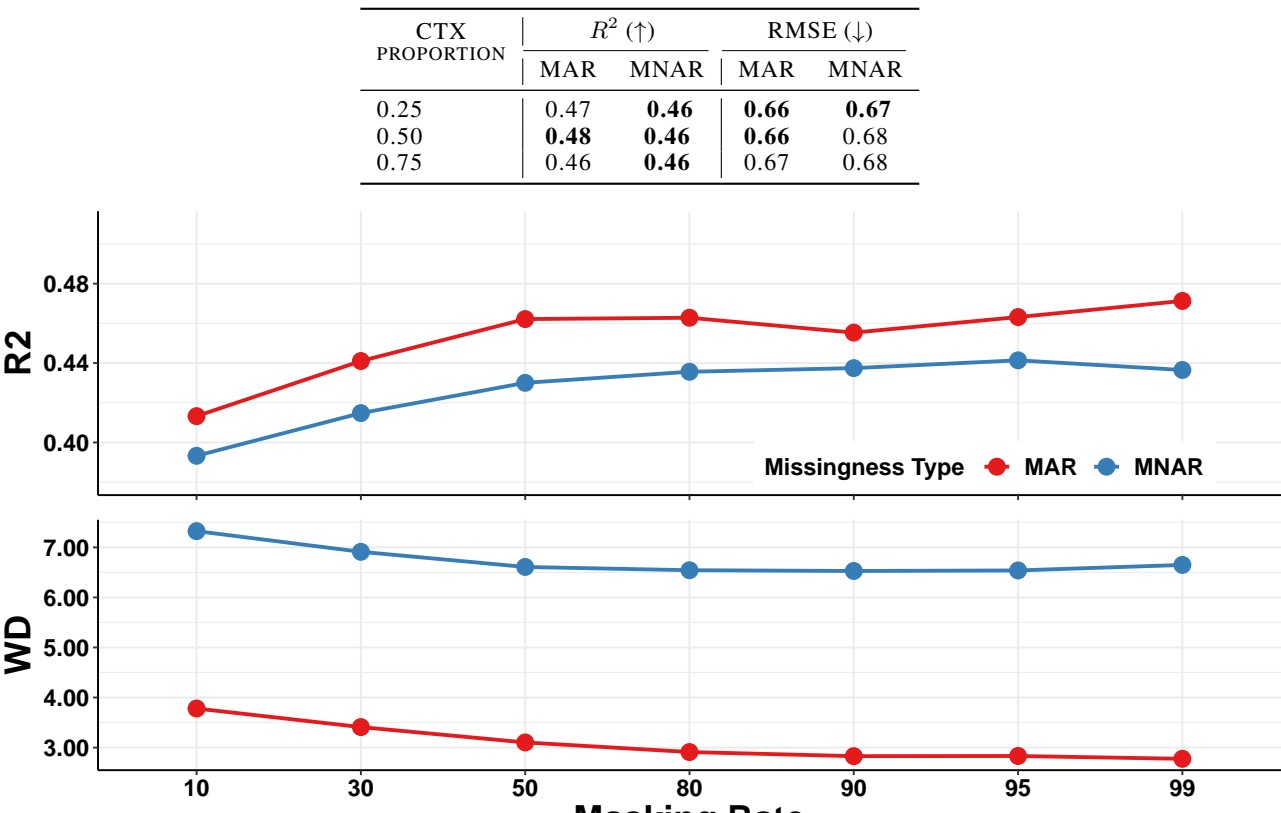

*Figure A12.* **Masking rate sensitivity.** Average performance metrics over a range of MT-CM masking rate choices. Evaluated under MAR and MNAR with at 30% missingness.

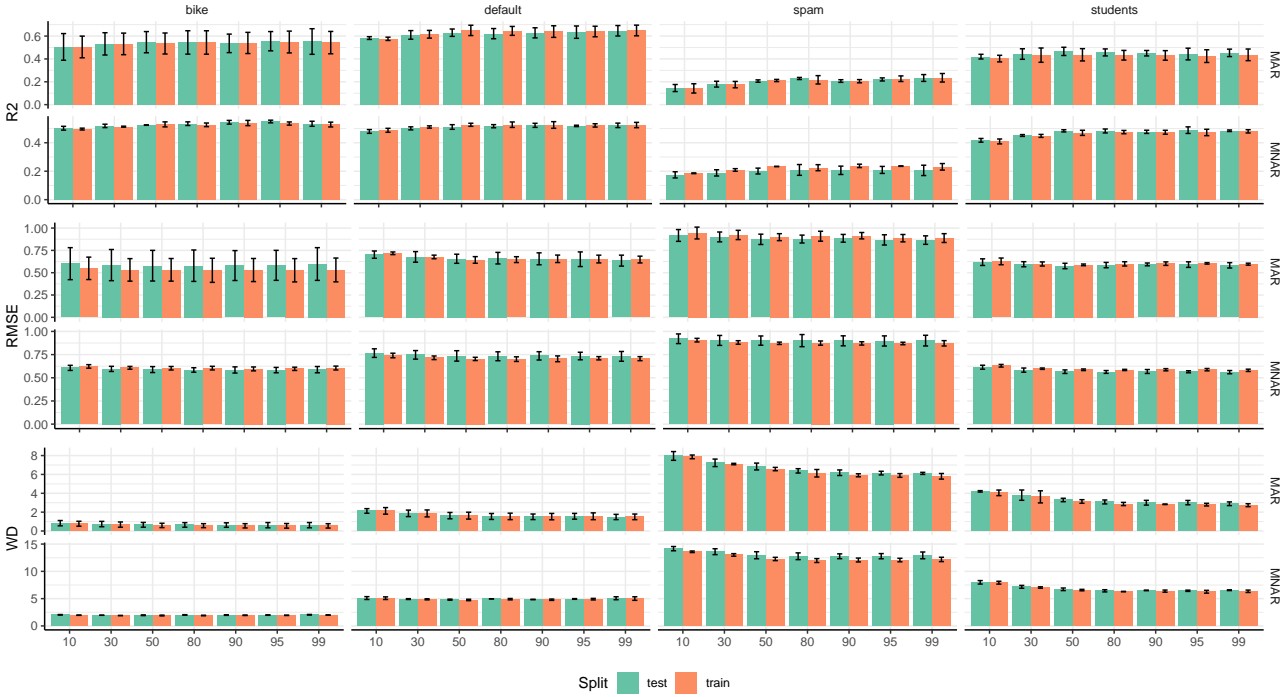

*Figure A13.* Experiments performed across four datasets split into train/test, under MAR and MNAR, at 30% missingness. Metrics ($R^2$, RMSE, WD) are reported as mean $\pm$ 95% CI and show model sensitivity to MT-CM.

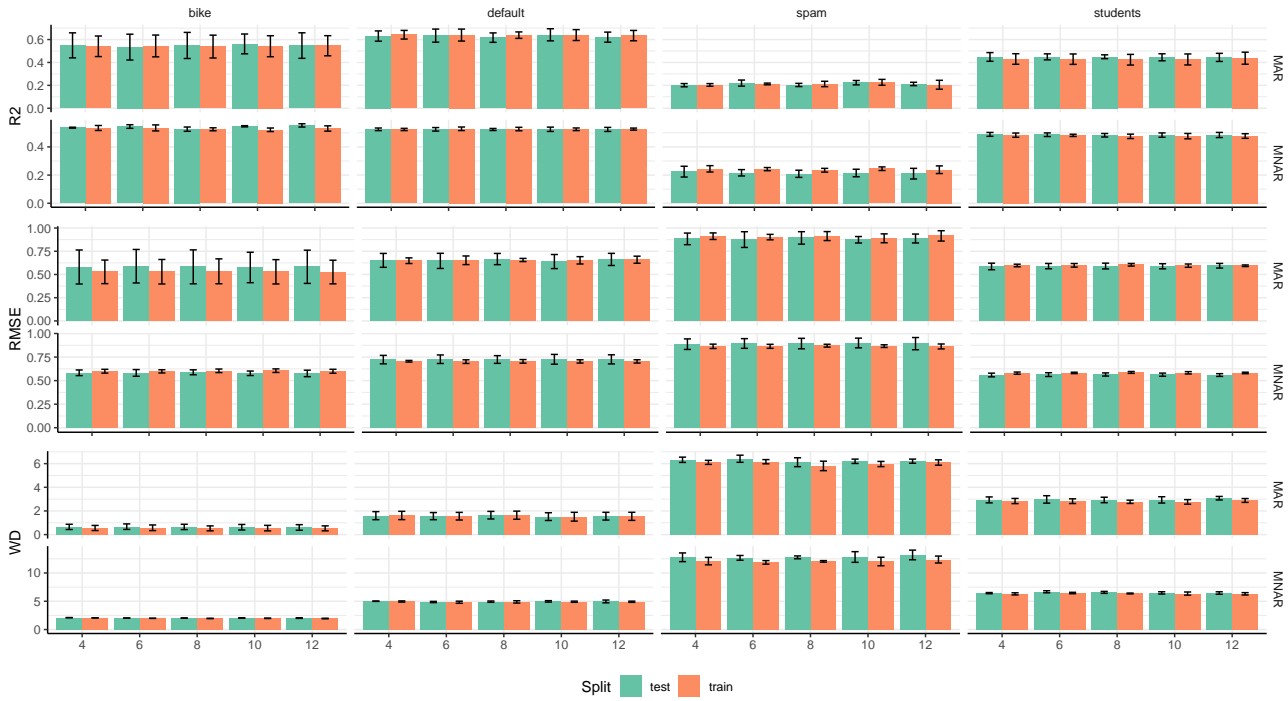

*Figure A14.* Experiments performed across four datasets split into train/test, under MAR and MNAR, at 30% missingness. Metrics ($R^2$, RMSE, WD) are reported as mean $\pm$ 95% CI and show model sensitivity to encoder depth.

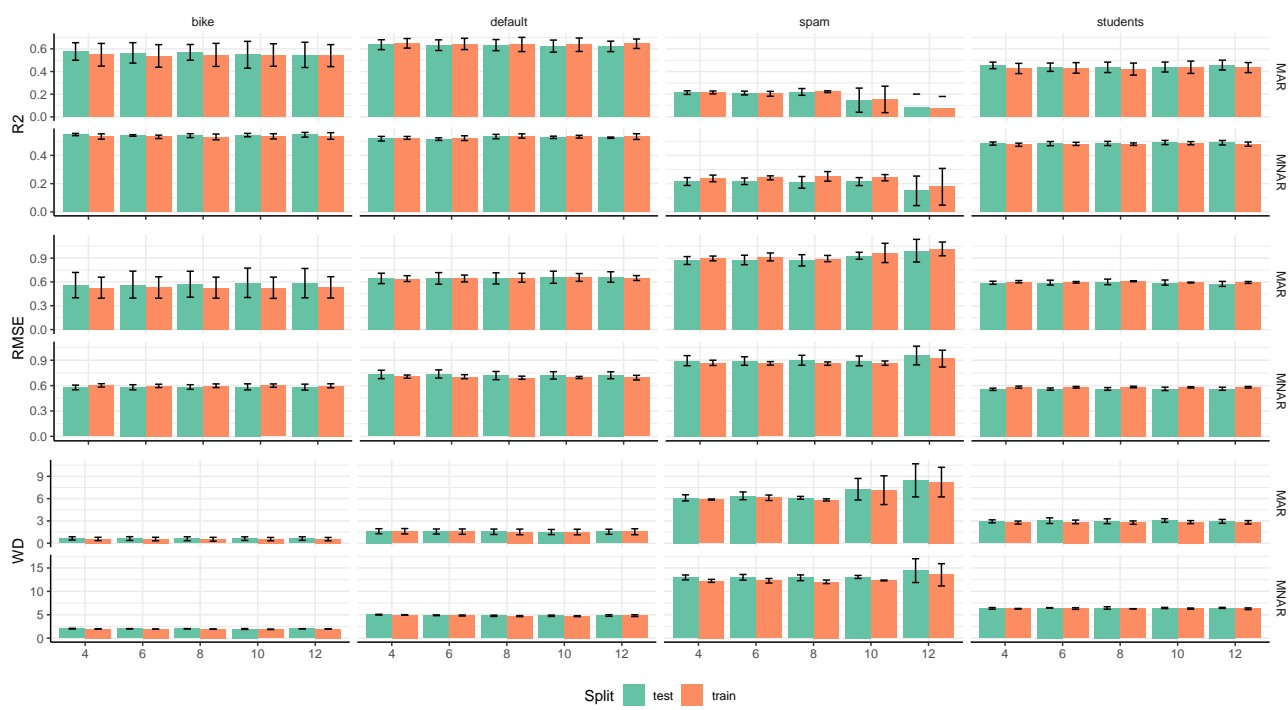

*Figure A15.* Experiments performed across four datasets split into train/test, under MAR and MNAR, at 30% missingness. Metrics ($R^2$, RMSE, WD) are reported as mean $\pm$ 95% CI and show model sensitivity to decoder depth.

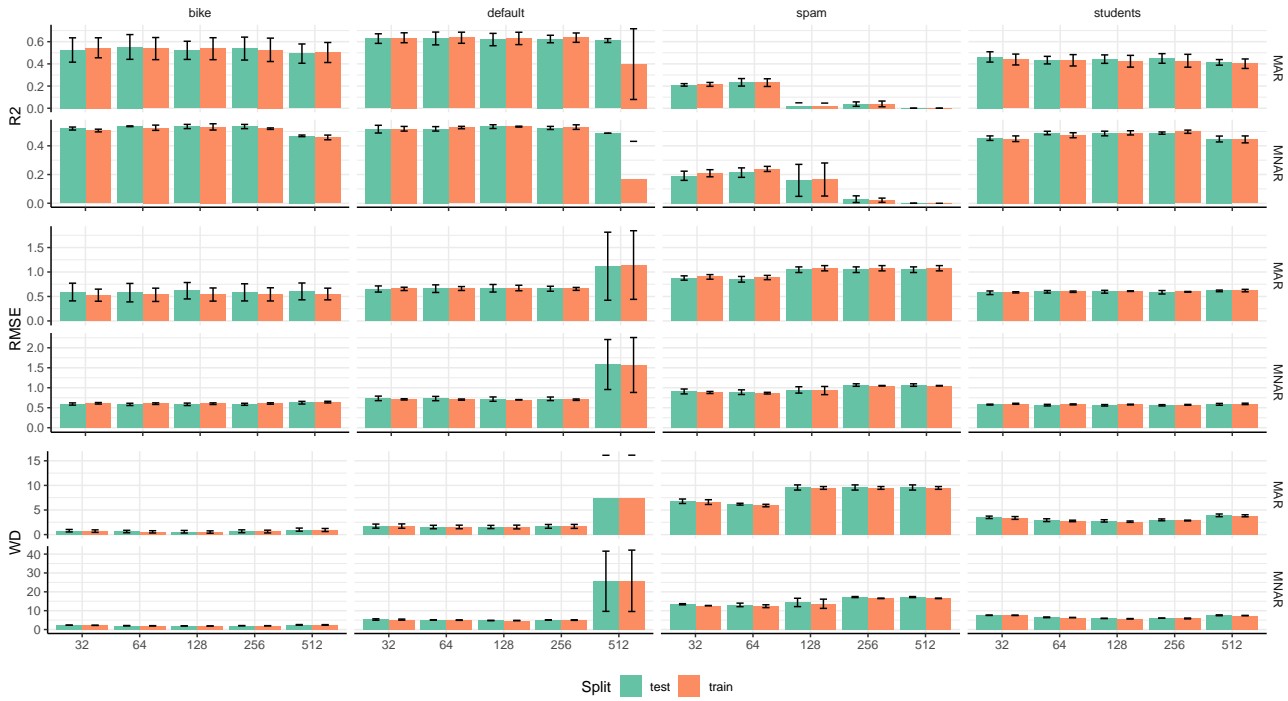

*Figure A16.* Experiments performed across four datasets split into train/test, under MAR and MNAR, at 30% missingness. Metrics ($R^2$, RMSE, WD) are reported as mean $\pm$ 95% CI and show model sensitivity to embedding size.

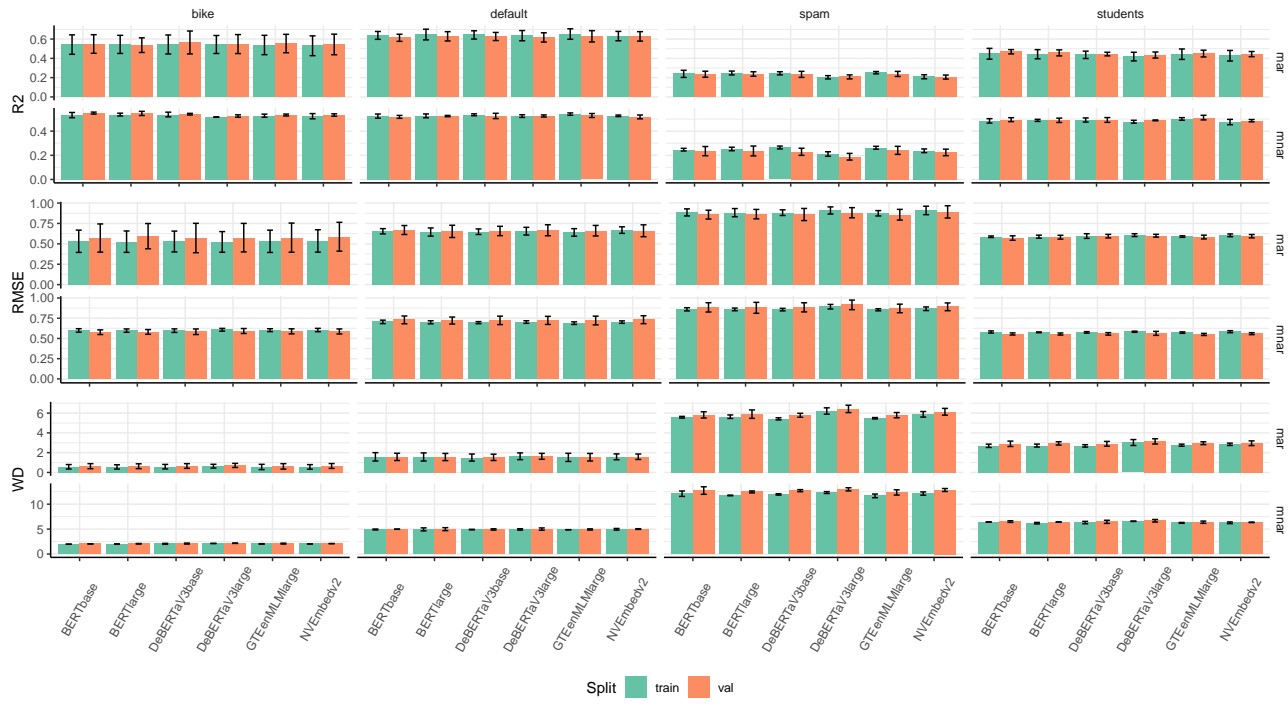

*Figure A17.* Experiments performed across four datasets split into train/test, under MAR and MNAR, at 30% missingness. Metrics ($R^2$, RMSE, WD) are reported as mean $\pm$ 95% CI and show model sensitivity to context embedding model.

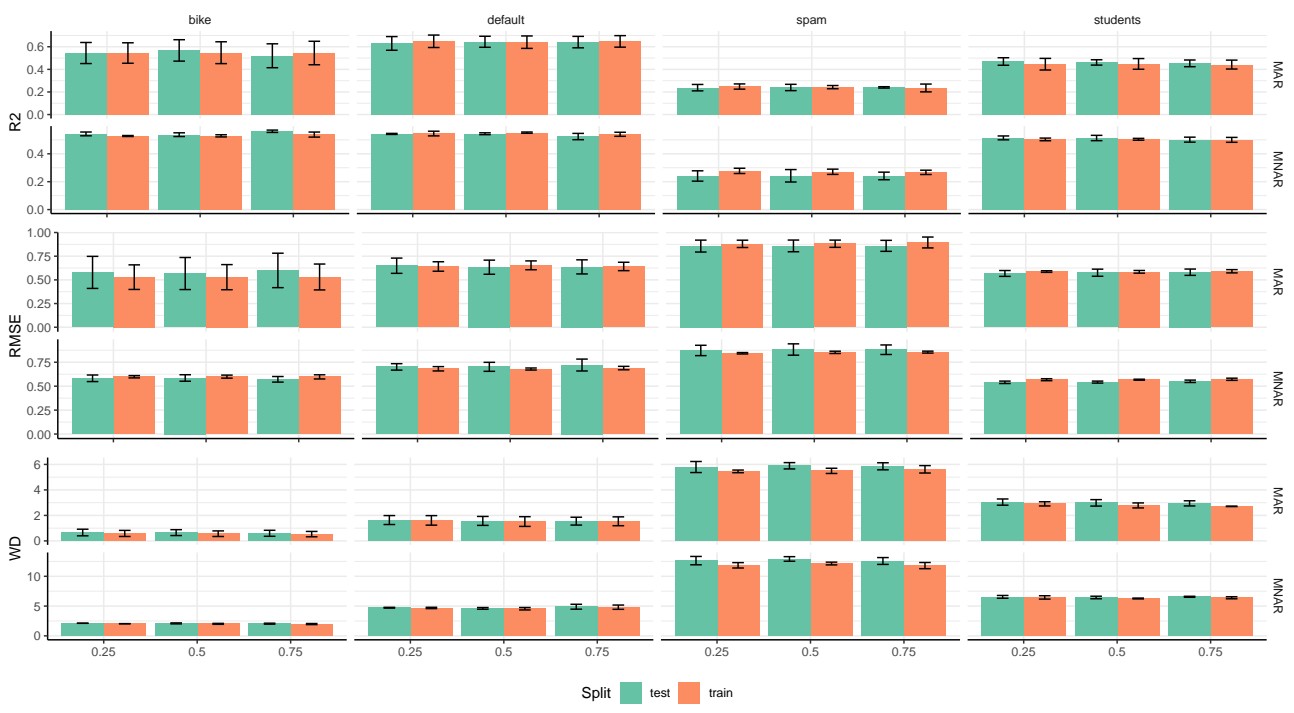

*Figure A18.* Experiments performed across four datasets split into train/test, under MAR and MNAR, at 30% missingness. Metrics ($R^2$, RMSE, WD) are reported as mean $\pm$ 95% CI and show model sensitivity to context embedding proportion.

