# OpenReview forum: "CACTI: Leveraging Copy Masking and Contextual Information to Improve Tabular Data Imputation"
_ICML.cc/2025/Conference — ICML 2025 spotlightposter_

### Official Review · Reviewer_eZeh · 2025-02-21

**Overall Recommendation:** 4

**Summary:**

The authors propose a new method to impute missing data in tabular data sets. This method, named CACTI, incorporate three components : a mask autoencoding approach, a median truncated copy-masking training strategy and the use of semantic relations between features.
The proposed method is evaluated across 10 real-world data sets with different missingness scenario (MCAR, MAR and MNAR) and different metrics related to pointwise imputation, distributional imputation or predictive performances.

**Claims And Evidence:**

The method is well described and the experimental protocol is well explained.

Regarding the way masks are generated : since they are permuted across observations, I agree that the dependence between missingness components is preserved. However, if masks are MAR, that is the probability of missingness depends on the observed values, permuting masks will break this dependence. Thus, the proposed strategy seems valid only in MCAR setting, with possible dependencies between missingness components. Thus, it is not valid for any type of missingness. Please correct me if this is not right.

**Essential References Not Discussed:**

NA

**Experimental Designs Or Analyses:**

The experimental protocol is sound. Many baselines from classic imputation techniques to deep learning strategies are compared.

**Methods And Evaluation Criteria:**

The experimental protocol is sound. In table 1, it would be interesting to add CACTI without the context awareness. Such a method is more comparable to classic method such as MissForest, and it would be interesting to see how they compare to each other (as I understand Table 2 does that but only on few data sets).

It could also be interesting to test the robustess to mislabeled input variables : what happens in terms of predictive performances, when the description of columns are permuted ?

**Other Comments Or Suggestions:**

Typos :
- page 1, second column : ‘can arise due due to’
- page 2 line 81, where $k \in [K]$ a verb is missing here
- page 2, line 66 : ‘a direct way to to incorporate’
- page 3 line 134, ‘contextual awareness to the to improve’
- page 3 line 158 since $m=1$ corresponds to an observed entry, I would say that feature $i$ is observed any time $j$ is observed
- page 5, line 273, so can we simulate
- page 7 line 336 just using the the feature context

**Other Strengths And Weaknesses:**

This paper propose to combine different existing ingredient to impute missing data. While there is no theory, and while the techniques are not totally new, the empirical protocol is exhaustive, testing for different missing data mechanisms, different datasets, and comparing several baselines. The empirical evaluation is thorough and is interesting on its own, in my opinion.

**Questions For Authors:**

-

**Relation To Broader Scientific Literature:**

The literature is well cited with respect to missing data imputation.

**Theoretical Claims:**

NA

---

> ### Author Rebuttal · Authors · 2025-04-01
>
> We would like to thank the reviewer for the helpful feedback and questions.
>
> ## Comments Addressed
> We've added CMAE (CACTI without context)  as an additional benchmark in the main results table (qrHN rebuttal Table. R1). We’ve also added a comparison between CACTI, CMAE to quantify the statistical significance of the improvements driven by context awareness (qrHN rebuttal Table. R2). We hope these results further clarify that context yields statistically significant (p<0.05) improvement over masked learning alone and is a useful source of inductive bias to improve imputation. We really appreciate the reviewer's attention to detail and highlighting the typos. They have now been addressed, and we apologize for the oversight.
>
> Next, our theoretical results in Appendix G show that copy masking can be *more* effective than random masking under MNAR. For MAR, consider a structured MAR setting [1] where the missing variables are dependent on some observed variable but also have additional dependencies, e.g., feature i is missing if feature j is missing. Such settings are quite common in questionnaire and other datasets. Under this setting, copy masking is still able to maintain the *structured* missingness between the variables whereas random masking completely diverges from the missingness process. The crux of our argument in Appendix G is that copy masking is better suited to modeling MAR and MNAR than random masking. For example, unlike random masking, applying copy masking to MAR results in masking only the missing features. This encourages the MAE to learn a conditional model that maximizes the probability of the masked feature given the observed features.
>
> ## Contributions
> While MAEs with random masking (ReMasker) and MLPs with naive copy masking (AutoComplete) have both been used to tackle imputation in prior works, we believe our work makes novel, concrete contributions.
> 1. MT-CM, an upgrade to copy masking, is a novel, key contribution. As highlighted by our results in Table A7 the application of naive copy masking to transformer-based MAE architecture leads to suboptimal performance and reduced learning efficiency. This is a significant gap in the field, as transformers are arguably the most popular architecture in current literature. MT-CM enables the application of copy masking effectively to such architectures and creates an upper bound on the worst-case scenario in batch-based training, i.e. the proportion of null tokens in any batch is upper bounded by 50%. Our MT-CM strategy unlocks the effective use of copy masking for any transformer-based masked learning model and can have applications beyond just tabular imputation.
> 2. CACTI is the first approach (to the best of our knowledge) to effectively leverage context as a useful inductive bias and show statistically significant improvements in various imputation settings. Our results highlight the importance of investigating new inductive biases as a promising direction for future tabular imputation research, particularly in domains with MNAR missingness.
> 3. Our results establish CACTI as a strong baseline for evaluating and developing future approaches in tabular imputation.
> 4. Although our theoretical section (Appendix G) is not highlighted in the main text (due to space restrictions), we believe we do provide a new empirical risk framework for evaluating MAE training without fully observed data (previous works don’t consider the probabilistic nature of the missingness process and only view the objective as representation loss minimization), provide the first formal motivation for copy masking and show how random masking is suboptimal compared to copy masking in non-MCAR settings. If given the chance we would like to introduce this theoretical framework in the main text in the final draft.
>
> We hope this addresses the reviewer's concerns and find this paper to be a valuable contribution to the current literature. We would appreciate their reconsideration of their overall recommendation. And as always we're happy to address any additional questions or concerns!
>
> [1] Jackson, J., et. al. A complete characterisation of structured missingness, 2023.

---

### Official Review · Reviewer_qrHN · 2025-03-08

**Overall Recommendation:** 3

**Summary:**

This paper addresses the missing data imputation problem by using a transformer-based architecture that leverages the missing patterns and textual information about features as inductive biases to improve imputation accuracy. Specifically, the paper proposes a median truncated copy masking training strategy that captures the empirical missing patterns of data during training. Then, the method exploits the feature name information by extracting language model embedding of the feature name and taking it as the input of the imputation model to capture the context information. The experiments show that these two inductive biases improve the accuracy of imputation.

**Claims And Evidence:**

The claims in the paper are supported by clear and convincing evidence.

**Essential References Not Discussed:**

No

**Experimental Designs Or Analyses:**

The experimental designs is comprehensive with different data set and methods are included. Also, the hyperparameter and ablation study are included.

**Methods And Evaluation Criteria:**

The proposed methods and evaluation criteria make sense for the target problem.

**Other Comments Or Suggestions:**

No

**Other Strengths And Weaknesses:**

Strengths:
 The writing and organization of the paper are clear. The experiment is also comprehensive and includes many data sets and methods. Also, ablation studies and the effect of hyperparameters are also included. The paper also theoretically proves the rationale of why the proposed masking strategy is better than the naïve random masking.

Weakness

The results tables could contain standard deviation to show the statistical significance.

**Questions For Authors:**

No

**Relation To Broader Scientific Literature:**

The study is focused on missing data Imputation with inductive biases of missing patterns and feature name description which is applicable to any domain that has MNAR data and whose feature description conveys important information, for example, healthcare.

**Theoretical Claims:**

The overall proofs in the supplementary are sound.

---

> ### Author Rebuttal · Authors · 2025-04-01
>
> We thank the reviewer for their thoughtful review and suggestions. We would like to highlight that all the individual per dataset/missingness condition results for all analyses performed in this paper were previously reported with 95% confidence intervals in the Appendix. We choose to exclude standard errors (SEs) in the main text to ensure visual parsimony and due to space constraints. Please see the silhouette for the main benchmarking results table with SEs reported in Table. R1 which we will include in the final version. Finally, we have a new set of results in Table. R2, which shows that CACTI demonstrates, via a paired t-test, statistically significant (p < 0.05) improvement in R2 and RMSE over all missingness conditions when compared to the next best method (ReMasker). This relative ranking improvement holds even after we exclude the 4 datasets used for the sensitivity and ablation analysis (Table. R3). This provides further evidence to show that leveraging inductive biases from dataset-specific missingness patterns and context information can help improve imputation accuracy. Finally, if the reviewer believes we have sufficiently addressed their concerns and finds this paper to be a valuable contribution to the current literature, we would appreciate their consideration in increasing their score. As always, we're happy to address any additional questions or concerns!
>
> **[Please note that we’re only reporting the top few methods since we’re limited to 5000 characters per rebuttals, final version of the paper will contain all methods when appropriate]**
>
> **Table. R1**: Main benchmarking results with standard errors in parentheses and two sets of results indicate train\|test.
> | Method |  |R2 | | |RMSE | |
> |--------|--------|--------|--------|--------|--------|--------|
> | | MCAR | MAR | MNAR | MCAR | MAR | MNAR |
> | CACTI | 0.455\|0.461  (0.002)\|(0.003) | 0.469\|0.467 (0.010)\|(0.011) | 0.458\|0.456 (0.003)\|(0.004) | 0.663\|0.640 (0.008)\|(0.005) | 0.675\|0.694 (0.016)\|(0.016) | 0.683\|0.666 (0.004)\|(0.006) |
> | CMAE | 0.441\|0.447 (0.002)\|(0.002) | 0.459\|0.460 (0.009)\|(0.010) | 0.440\|0.439 (0.002)\|(0.003) | 0.673\|0.653 (0.008)\|(0.007) | 0.685\|0.696 (0.017)\|(0.016) | 0.699\|0.691 (0.005)\|(0.008) |
> | ReMasker | 0.437\|0.438 (0.002)\|(0.002) | 0.445\|0.443 (0.010)\|(0.010) | 0.402\|0.402 (0.003)\|(0.004) | 0.681\|0.665 (0.008)\|(0.006) | 0.691\|0.712 (0.017)\|(0.014) | 0.729\|0.709 (0.005)\|(0.006) |
> | DiffPuter | 0.400\|0.415 (0.003)\|(0.004) | 0.386\|0.430 (0.010)\|(0.012) | 0.363\|0.372 (0.004)\|(0.004) | 0.731\|0.704 (0.009)\|(0.005) | 0.770\|0.752 (0.020)\|(0.021) | 0.794\|0.767 (0.024)\|(0.024) |
> | HyperImpute | 0.406\|0.382 (0.003)\|(0.004) | 0.439\|0.391 (0.010)\|(0.011) | 0.393\|0.347 (0.005)\|(0.005) | 0.722\|0.734 (0.007)\|(0.011) | 0.727\|0.774 (0.017)\|(0.016) | 0.757\|0.776 (0.006)\|(0.007) |
>
> **Table. R2**: T-test to quantify the statistical significance of the contributions of MT-CM and context awareness (CMAE = CACTI w/o context).
> | Metric | Missingness | Target Method | Baseline Method | Diff. Est. | P Value |
> |---|---|---|---|---|---|
> |  | All | CACTI | ReMasker | 0.034 | 4.4e-7 |
> |  | All | CACTI | CMAE | 0.013 | 1.1e-5 |
> |  | All | CMAE | ReMasker | 0.021 | 4.2e-5 |
> |  | MCAR | CACTI | ReMasker | 0.023 | 5.5e-4 |
> |  | MCAR | CACTI | CMAE | 0.014 | 3.e-3 |
> |  | MCAR | CMAE | ReMasker | 0.017 | 8.9e-3 |
> | R2 | MAR | CACTI | ReMasker | 0.025 | 2.3e-2 |
> |  | MAR | CACTI | CMAE | 0.007 | 9.4e-2 |
> |  | MAR | CMAE | ReMasker | 0.018 | 4.8e-2 |
> |  | MNAR | CACTI | ReMasker | 0.054 | 1.1e-4 |
> |  | MNAR | CACTI | CMAE | 0.017 | 8.7e-4 |
> |  | MNAR | CMAE | ReMasker | 0.037 | 4.6e-4 |
>
> **Table. R3**: Results excluding the 4 datasets used for sensitivity analysis.
> | Method    |           | R2        |           |            | RMSE       |           |        |    WD       |           |
> |-----------|-----------|-----------|-----------|------------|------------|-----------|-----------|-----------|-----------|
> |           | MCAR      | MAR       | MNAR      | MCAR       | MAR        | MNAR      | MCAR      | MAR       | MNAR      |
> | CACTI     | 0.45\|0.45 | 0.47\|0.46 | 0.46\|0.46 | 0.67\|0.64  |  0.68\|0.71 | 0.69\|0.66 | 3.19\|3.23 | 1.36\|1.41 | 3.31\|3.36 |
> | CMAE      | 0.44\|0.44 | 0.47\|0.46 | 0.45\|0.44 | 0.68\|0.66  | 0.69\|0.70  | 0.70\|0.68 | 3.22\|3.27 | 1.41\|1.45 | 3.39\|3.44 |
> | ReMasker  | 0.44\|0.44 | 0.45\|0.45 | 0.42\|0.42 | 0.68\|0.66  | 0.69\|0.72  | 0.72\|0.68 | 3.24\|3.28 | 1.69\|1.72 | 3.33\|3.38 |
> | DiffPuter | 0.41\|0.43 | 0.41\|0.46 | 0.38\|0.39 | 0.73\|0.69  | 0.74\|0.73  | 0.80\|0.76 | 3.33\|3.32 | 1.89\|1.53 | 3.43\|3.41 |
> | HyperImpute | 0.40\|0.37 | 0.43\|0.38 | 0.39\|0.34 | 0.74\|0.75  | 0.73\|0.79  | 0.77\|0.76 | 2.75\|3.27 | 1.46\|1.86 | 2.76\|3.12 |

---

### Official Review · Reviewer_PQVE · 2025-03-13

**Overall Recommendation:** 3

**Summary:**

The authors introduce CACTI (Context Aware Copy masked Tabular Imputation) for imputing missing values in tabular data. CACTI’s backbone is a Masked Autoencoder based on Transformers. It brings the following key modifications to this architecture:

- Instead of randomly masking observed values as in ReMasker, it uses copy-masking: additional missing values are introduced in sample i by applying the missingness pattern of another sample j. These additional missing values will then serve to create the reconstruction loss. Copy-masking allows to exploit the missing patterns present in the dataset.

- Copy-masking is further enhanced with Median Truncated Copy Masking (MT-CM), which truncates all samples in a batch to have a maximum number of observed features. This max. number is defined by the median number of observed features in a batch. This prevents from having too many missingness tokens to process.

- Context-awarness is achieved by concatenating column description embeddings (obtained with GTE-en-MLM-large) with the numerical embeddings of scalar values.

This architecture is evaluated on 10 datasets, across varying missingness rates and the 3 missingness mechanisms (MCAR, MAR, MNAR). The imputation is evaluated according to the reconstruction R2, RMSE, and Wasserstein distance. CACTI is evaluated against 13 baselines and shows state-of-the-art performance across the 3 missingness mechanisms.

**Claims And Evidence:**

The claims are supported by convincing evidence.

**Essential References Not Discussed:**

I think that ReMasker and [3], which appear to be the cornerstones of this work (respectively for using a MAE with random masking for imputation, and for using copy-masking) could be introduced with more details in the related work.

For example, the authors say: “Our first contribution is in extending an approach introduced by [3] of using “… To clearly identify contributions, it would be easier to first introduce copy-masking as a related work, and then present the proposed extension as a contribution.

[3] An et al 2023, Deep learning-based phenotype imputation on population-scale biobank data increases genetic discoveries.

**Experimental Designs Or Analyses:**

yes

**Methods And Evaluation Criteria:**

Methods and evaluation criteria make sense.

**Other Comments Or Suggestions:**

* On Fig 2., it may be more informative to plot the R2 relative to the mean R2 across methods for a given dataset and missingness mechanism.

* The results in Table 4 on model architecture (encoder and decoder depth and embedding size) show that the smallest options considered are almost the best. It is not clear that an encoder depth of 10 provides significantly better results that a depth of 4. Exploring smaller depths and embedding size would be useful, as the smaller the model the better (given a fixed performance).

* Given the results presented in Figure 3 on Masking rate sensitivity, where increasing the copy masking probability improves performances up to 99% (the maximum probability tested), it could make sense to remove this hyperparameter and apply copy-masking 100% of the time.

* The default hyperparameters of competing method should be specified in Appendix to ensure reproducibility.

* Is the [MASK] token the same for all features?


**Typos:**

l.11: due due

l.66: to to

l.134: to the to

l.214: with a detail description

l.239: we conduct a through ablation

l.271: CACTI’s out performs

l.313: of the of the top 5

**Other Strengths And Weaknesses:**

**Strengths**

- The improvements in the imputation performance, notably under MNAR, are large. +0.02 (MCAR) / +0.03 (MAR) / +0.06 (MNAR) R2 pts compared to the next best model ReMasker.

- Many ablation studies help identify the most critical parts of a model, providing valuable insights for guiding future research.


**Weaknesses**

- I found the paper difficult to follow. The explanations rely heavily on notations, which I believe unnecessarily complicates the presentation. Some concepts require significant effort to grasp. The writing could be improved by reducing unnecessary notations and incorporating figures to clarify key ideas, notably copy-masking.

- The results do not convincingly show that using the context improves imputation. We see on Table 2 that using the context on top of random masking improves the R2 from 0.20 to 0.26. However, the result is obtained over 4 datasets only. Moreover, when copy-masking is used instead of random masking, the effect of the context disappears in MAR and becomes very small in MNAR, potentially not significant. From these experiments, it is hard to tell whether using context can help imputations.

- Are the 4 datasets of the ablation studies part of the 10 evaluation datasets? If yes, as it seems that CACTI’s hyperparameters were chosen based on these 4 datasets, it would mean that hyperparameters were chosen on part of the test set, and that the performances may be inflated for these datasets.

**Questions For Authors:**

* L. 135, the authors state: “we perform a row-wise permutation to create a mask”. Row-wise permutation would mean shuffling elements within each row. Based on other elements in the paper, notably the permutation matrix of size N by N in Algorithm 1, I think that the authors do row-shuffling (eg swapping row I with row j). What do the authors do?

* CACTI outperforms ReMasker by 0.03 R2 pts in MAR (0.06 in MNAR) on average over 10 datasets, but CACTI outperforms RMAE, which uses ReMasker’s random masking, by a much larger margin (+0.25 R2 pts in MAR and MNAR) on 4 datasets. Why such difference in the size of the improvement given that both RMAE and ReMasker use random masking?

* To leverage the missingness structure, a straightforward competing approach would be to include the mask as additional input features for the imputation model. This may not be straightforward for all methods, but it is for methods such as missforest.

* Do the 10 evaluation datasets include the 4 datasets used for exploring the architecture hyperparameters?

* How were the evaluation datasets chosen?

* What do the column description look like in the 10 evaluation datasets? Were they chosen so that these descriptors are meaningful?

**Relation To Broader Scientific Literature:**

This paper provides an imputation model for tabular data. Important references for this task are cited in the introduction. A particularity of this approach is that it is meant to also be effective in MNAR, while many methods are only valid under M(C)AR (although the theoretical justification for this is not clearly stated).

**Theoretical Claims:**

A “theoretical framework to motivate the need for copy masking and why random masking might not be optimal in non-MCAR settings” is provided in Appendix G. Yet it is not exposed in the main paper, and does not contain a clear proposition or theorem, so I did not consider it as a contribution.

---

> ### Author Rebuttal · Authors · 2025-04-01
>
> We thank the reviewer for their valuable feedback.
>
> ## Does context help?
> Our ablation analysis focused on the *relative* contributions of median truncated copy masking (MT-CM) and context awareness with fixed architecture settings and hyperparameters (Appendix C.2). Random masking autoencoder (RMAE) and ReMasker perform nearly identically when hyperparameters match (internally verified). Performance differences primarily stem from different masking rates: 30% for ReMasker (optimal per original paper) versus 90% for CACTI/RMAE.
>
> To address potential ambiguity, we extended our analysis of CMAE (CACTI without context but with MT-CM) from 4 to all 10 datasets (qrHN rebuttal Table. R1). CMAE consistently improves or matches ReMasker across R2, RMSE, and WD metrics, while CACTI strictly dominates CMAE in all conditions. This confirms that context can improve imputation accuracy. We also performed a comparison using one-sided paired t-tests (qrHN rebuttal Table. R2). CACTI shows statistically significant (p<0.05) improvement over CMAE for R2 (and RMSE) in most settings, reinforcing that context enhances imputation. MT-CM (via CMAE) delivers statistically significant R2 improvements over random masking (via ReMasker) across all settings. CACTI, combining both approaches, shows significant improvement across nearly all metrics and conditions. While MT-CM provides larger improvements, context contributes meaningfully, demonstrating that both MT-CM and context awareness provide useful inductive biases.
>
> ## Datasets
> We selected 10 open datasets from those used by Hyperimpute, ReMasker, and Diffputer authors for fair comparison. Our selection criteria were: 1) 6 with mixed features and 4 with continuous-only features datasets to ensure type diversity, and 2) half the datasets having feature descriptions. Table A9 will be expanded with information about which datasets have column descriptions. Our anonymous repo provides code to obtain all datasets and generate colinfo.txt files (see "Generating UCI ML Datasets" in README).
>
> The 4 datasets used in ablation and sensitivity analysis were randomly chosen from the 10 used in main benchmarking. While this approach is common in prior works, we acknowledge the potential risk of inflating overall results. To maintain fairness, we used similar/same datasets and optimal parameters from previous works. To directly address this concern, we re-analyzed our main benchmarking results excluding these 4 datasets (qrHN rebuttal Table. R3). Results confirm CACTI still strictly outperforms existing state-of-the-art methods in R2 and RMSE, addressing concerns about fairness and result inflation.
>
> ## Writing
> We appreciate the reviewer highlighting these typos. While our notation was chosen for precision, we will move significant portions to the appendix, replacing them with more intuitive text descriptions and figures. ReMasker and [3] are indeed key previous works and we’ll better highlight ReMasker and move the discussion on copy masking from methods to related works. The [MASK] token is indeed fixed and identical for all features, following standard practice in MAEs and ViTs. The hyperparameters of baseline method will be specified in an appendix table. Finally, we apologize for the ambiguity - we mean row-wise shuffling (permuting rows of the observed missingness mask matrix). All these will be fixed in the final draft.
>
> ## Other Comments
> Regarding the comment about including  “the mask as additional input features for the imputation model”, we included notMIWAE, which leverages this idea. It maximizes the ELBO of the joint likelihood of the observed features and missing pattern via IWVAE, this ensured we compared against methods that directly factor in missingness structure. As seen in Table 2, we outperform notMIWAE.
>
> Next, while higher copy-masking rates sometimes improve performance, keeping this as a tunable hyperparameter is justified because: 1) performance decreases at rates >95% in MNAR settings for R² and WD; 2) our conservative 90% rate maximizes average performance across sensitivity analysis datasets but isn't universally optimal; 3) practitioners often prioritize specific features based on domain needs, making flexible masking valuable; and 4) optimal masking rates vary significantly across datasets. Next, the embedding size and depths chosen here perform consistently well with reasonable resource usage (<200MB GPU RAM). Since imputation is rarely the end application, we maintain these as tunable parameters for users to optimize their specific use cases.
>
> Finally, we refer the reviewer to point 4 in the contributions section of rebuttal to reviewer eZeh, which addresses concerns about the theoretical contributions.
>
> We hope our responses have addressed the reviewer's primary concerns and demonstrated the paper's contribution to current literature. If so, we would appreciate a reconsideration of the overall recommendation. We're happy to address any additional questions!

---

> > ### Comment · Reviewer_PQVE · 2025-04-04
> >
> > Thank you for the responses.
> >
> > According to the new tables provided, it seems that the context (i.e. using CACTI rather than CMAE) improves performances slightly but significantly in MCAR and MNAR, but not in MAR (all for a missingness proportion of 30%).
> >
> > These results do not entirely convince me that the context significantly improves performances (ie CACTI relative to CMAE), because they are based on a limited number of datasets, at a single missing rate, and are significant for 2/3 missingness mechanisms.
> >
> > To see results per dataset rather than on average, could the authors provide a scatterplot (or a table if easier) showing the performance of CMAE vs CACTI for each dataset, identifying which dataset has feature descriptions?
> >
> > For future work, I think it would be an interesting sanity check to permute feature names and descriptions in a dataset, and check whether this affects performances.

---

> > > ### Author Response · Authors · 2025-04-06
> > >
> > > We thank the reviewer for engaging in this discussion! We extended our analysis to all missingness percentages and settings: context provides a statistically significant improvement (p<0.05) in all settings except for MAR@30% (Table R4). CACTI also outperforms CMAE (win rate) in a majority of the datasets. Please see the requested per dataset results for 30% missingness in Table R5.
> > >
> > > Additionally, all datasets have column names but some also have extended column descriptions. Column relatedness will vary across datasets, and column names alone may contain sufficient semantic information. The following datasets have column descriptions for all features: california, magic, letter, obesity, students. We also humbly note that the number and diversity of evaluation datasets are on par with the evaluation procedure used for the most recently published imputation approaches such as DiffPuter (ICLR’25) and ReMasker (ICLR’24) to allow for a fair comparison with current literature. And indeed, it would be informative to assess the sensitivity of permuting the column description assignments and/or replace them with random information in future work.
> > >
> > >
> > > Finally, while we claim that context *can* be a useful source of inductive bias, it’s contribution can vary. In settings where context is not helpful, the model can choose to ignore it. We hypothesize that MAR is probably the most straightforward setting for MAEs since only a subset of the features are masked (as opposed to all features in MCAR and MNAR). Here CACTI would simply learn a model that maximizes the probability of the masked feature given the observed. We expect context to be most useful in the most difficult MNAR cases where the imputation task is more complex.
> > >
> > > **Due to the 5K character limit, we are unable to include the per datasets results for 10, 50 and 70 percent missingness here. If the reviewer would like to see these results we kindly ask them to reply to this rebuttal to give us a chance to add these results.**
> > >
> > > **Table. R4**: T-test to quantify the statistical significance of the contributions of context awareness (CACTI v CMAE) in all missingness percentages. Win rate = % of datasets where CACTI > CMAE.
> > > | Missingness | Miss % |  P Value | Diff. Est. |Win Rate % |
> > > |-----------|-----------|----------|--------| --------|
> > > | MAR       | 10        | 7.9e-03  | 0.023  | 89 |
> > > | MAR       | 30        | 9.4e-02  | 0.007  | 70|
> > > | MAR       | 50        | 3.7e-02  | 0.010  | 70|
> > > | MAR       | 70        | 7.2e-03  | 0.012  | 90|
> > > | MCAR      | 10        | 2.0e-02  | 0.014  | 80|
> > > | MCAR      | 30        | 3.5e-03  | 0.014  | 80|
> > > | MCAR      | 50        | 6.5e-03  | 0.011  | 90|
> > > | MCAR      | 70        | 2.0e-02  | 0.010  | 89|
> > > | MNAR      | 10        | 2.3e-03  | 0.017  | 80|
> > > | MNAR      | 30        | 8.7e-04  | 0.017  | 100|
> > > | MNAR      | 50        | 1.3e-03  | 0.016  | 100|
> > > | MNAR      | 70        | 4.6e-04  | 0.015  | 100|
> > >
> > > **Table. R5**: R2 performance estimates for all datasets, missingness types, at 30% missingness. Delta = the increase in CACTI R2 relative to CMAE.
> > > | Dataset    | Missingness | CACTI | CMAE  | Delta    |
> > > |------------|-----------|-------|-------|----------|
> > > | bike       | MAR       | 0.560 | 0.538 | 0.0222   |
> > > | bike       | MCAR      | 0.553 | 0.538 | 0.0147   |
> > > | bike       | MNAR      | 0.551 | 0.525 | 0.0261   |
> > > | california | MAR       | 0.435 | 0.444 | -0.00907 |
> > > | california | MCAR      | 0.412 | 0.377 | 0.0348   |
> > > | california | MNAR      | 0.390 | 0.344 | 0.0460   |
> > > | default    | MAR       | 0.632 | 0.629 | 0.00261  |
> > > | default    | MCAR      | 0.569 | 0.553 | 0.0157   |
> > > | default    | MNAR      | 0.533 | 0.518 | 0.0146   |
> > > | income     | MAR       | 0.246 | 0.246 | 0.000169 |
> > > | income     | MCAR      | 0.286 | 0.284 | 0.00152  |
> > > | income     | MNAR      | 0.306 | 0.301 | 0.00435  |
> > > | letter     | MAR       | 0.818 | 0.804 | 0.0131   |
> > > | letter     | MCAR      | 0.804 | 0.793 | 0.0107   |
> > > | letter     | MNAR      | 0.779 | 0.768 | 0.0104   |
> > > | magic      | MAR       | 0.517 | 0.525 | -0.00879 |
> > > | magic      | MCAR      | 0.554 | 0.558 | -0.00373 |
> > > | magic      | MNAR      | 0.582 | 0.572 | 0.00987  |
> > > | obesity    | MAR       | 0.358 | 0.362 | -0.00419 |
> > > | obesity    | MCAR      | 0.293 | 0.280 | 0.0132   |
> > > | obesity    | MNAR      | 0.292 | 0.284 | 0.00857  |
> > > | shoppers   | MAR       | 0.402 | 0.398 | 0.00459  |
> > > | shoppers   | MCAR      | 0.370 | 0.370 | -0.00022 |
> > > | shoppers   | MNAR      | 0.394 | 0.381 | 0.0133   |
> > > | spam       | MAR       | 0.251 | 0.210 | 0.0403   |
> > > | spam       | MCAR      | 0.260 | 0.230 | 0.0297   |
> > > | spam       | MNAR      | 0.238 | 0.210 | 0.0286   |
> > > | students   | MAR       | 0.454 | 0.446 | 0.00750  |
> > > | students   | MCAR      | 0.512 | 0.486 | 0.0253   |
> > > | students   | MNAR      | 0.494 | 0.482 | 0.0127   |

---

### Decision · Program_Chairs · 2025-05-01

**Decision:**

Accept (spotlight poster)

**Comment:**

This submission present a missing-value imputation with a approach following the line of masked auto-encoders, but recycling missing-value patterns actually present in the data as well as exploiting the column names. The submission generated interest from the reviewers and discussion. The reviewers appreciated the gain in imputation performance.

It was noted that the readability could be improved. Should the paper be accepted, the authors should work on this.